# Presynaptic NMDARs on spinal nociceptor terminals state-dependently modulate synaptic transmission and pain

Rou-Gang Xie [1,11], Wen-Guang Chu[1,11], Da-Lu Liu[1,2,11], Xu Wang[1,3,11], Sui-Bin Ma[1], Fei Wang[1], Fu-Dong Wang[4], Zhen Lin[5], Wen-Bin Wu[4], Na Lu[1], Ying-Ying Liu[1], Wen-Juan Han[1], Hui Zhang[6], Zhan-Tao Bai[3], San-Jue Hu[1], Hui-Ren Tao[7], Thomas Kuner [8], Xu Zhang[9], Rohini Kuner [10], Sheng-Xi Wu [1✉] & Ceng Luo [1✉]

Postsynaptic NMDARs at spinal synapses are required for postsynaptic long-term potentiation and chronic pain. However, how presynaptic NMDARs (PreNMDARs) in spinal nociceptor terminals control presynaptic plasticity and pain hypersensitivity has remained unclear. Here we report that PreNMDARs in spinal nociceptor terminals modulate synaptic transmission in a nociceptive tone-dependent manner. PreNMDARs depresses presynaptic transmission in basal state, while paradoxically causing presynaptic potentiation upon injury. This state-dependent modulation is dependent on $Ca^{2+}$ influx via PreNMDARs. Small conductance $Ca^{2+}$-activated $K^+$ (SK) channels are responsible for PreNMDARs-mediated synaptic depression. Rather, tissue inflammation induces PreNMDARs-PKG-I-dependent BDNF secretion from spinal nociceptor terminals, leading to SK channels downregulation, which in turn converts presynaptic depression to potentiation. Our findings shed light on the state-dependent characteristics of PreNMDARs in spinal nociceptor terminals on modulating nociceptive transmission and revealed a mechanism underlying state-dependent transition. Moreover, we identify PreNMDARs in spinal nociceptor terminals as key constituents of activity-dependent pain sensitization.

[1] Department of Neurobiology, School of Basic Medicine, Fourth Military Medical University, 710032 Xi'an, China. [2] Department of Radiation Medicine and Protection, Ministry of Education Key Lab of Hazard Assessment and Control in Special Operational Environment, School of Public Health, Fourth Military Medical University, 710032 Xi'an, China. [3] Research Center for Resource Polypeptide Drugs & College of Life Sciences, Yanan University, 716000 Yanan, China. [4] Class 2015, The Fourth Squadron of the Fourth Regiment, School of Basic Medicine, Fourth Military Medical University, 710032 Xi'an, China. [5] Class 2013, The Fourth Squadron of the Second Regiment, School of Basic Medicine, Fourth Military Medical University, 710032 Xi'an, China. [6] Department of Health Statistics, School of Preventive Medicine, Fourth Military Medical University, 710032 Xi'an, China. [7] Department of Spine Surgery, Shenzhen University General Hospital, 518055 Shenzhen, Guangdong, China. [8] Institute for Anatomy and Cell Biology, University of Heidelberg, Im Neuenheimer Feld 307, Heidelberg 69120, Germany. [9] Institute of Neuroscience and State Key Laboratory of Neuroscience, CAS Center for Excellence in Brain Science, Shanghai Institutes for Biological Sciences, Chinese Academy of Sciences, 200031 Shanghai, China. [10] Pharmacology Institute, University of Heidelberg, Im Neuenheimer Feld 366, Heidelberg 69120, Germany. [11]These authors contributed equally: Rou-Gang Xie, Wen-Guang Chu, Da-Lu Liu, Xu Wang. ✉email: shengxi@fmmu.edu.cn; luoceng@fmmu.edu.cn

M echanisms underlying chronic, pathological pain are not well-understood. Long-term potentiation (LTP) at spinal synapses between nociceptor terminals and spinal projection neurons has been proposed to be a key cellular basis for pain hypersensitivity caused by inflammation or nerve injury[1–4]. This spinal LTP is assumed to be post-synaptically induced (post-LTP) and depends on activation of postsynaptic N-methyl-D-aspartate subtype of glutamate receptors (NMDARs)[1,3,4]. Mounting preclinical evidence indicates that centrally expressed NMDARs are crucial to pain hypersensitivity[5–7]. However, pharmacological blockade of these receptors in human is associated with unwanted central side effects, e.g. cognitive impairment, psychiatric disorders. Thus, for clinical exploitation of the analgesic properties at targeting NMDA receptors, a major challenge is to devise strategies that reduce or abolish their adverse effects without attenuating the analgesic effects.

Apart from post-LTP, presynaptic LTP (pre-LTP) has also been observed in many brain regions, i.e. hippocampus, anterior cingulate cortex (ACC), amygdala[8–10]. This form of pre-LTP has been shown to be additive to post-LTP and adds salience to information encoded by post-LTP. For example, pre-LTP and post-LTP reported in ACC converge to mediate the interaction between chronic pain and anxiety[8]. Previous studies have demonstrated that peripheral inflammation or injury leads to repetitive firing of nociceptive afferents and increased glutamate release from presynaptic spinal nociceptor terminals[11,12]. But would this presynaptic activity alone be enough to trigger LTP (pre-LTP)? What presynaptic candidates located in nociceptors could underlie this spinal pre-LTP? Would these presynaptic changes and mechanisms be related to pain hypersensitivity associated with tissue injury?

In the classic view, postsynaptic NMDARs act via $Ca^{2+}$ to signal coincidence detection in Hebbian plasticity in the brain including at C-fiber synapses in the spinal cord[3,4,13]. However, additional and unconventional modes of NMDARs function have also consistently been reported[14–17]. For example, NMDARs are also found to be presynaptically localized in specific regions, i.e. somatosensory cortex[18], hippocampus[19], visual cortex[20], cerebellum[21], amygdala[22], corticostriatal synapses[23], and spinal primary afferent terminals[24]. Emerging anatomical and physiological reports have documented the key significance of presynaptic NMDARs (PreNMDARs) in shaping synapse-specific effects on neurotransmitter release and plasticity, i.e. LTP or LTD in cortical synapses[14,15,22,23,25]. However, despite these long-standing insights and advances, whether and how PreNMDARs in spinal nociceptor terminals contribute to spinal presynaptic plasticity and further to pain hypersensitivity remains to be thoroughly examined. This possibility was first suggested by the finding that activation of PreNMDARs modulates substance P and glutamate release in the spinal cord. However, these reports are contradictory and debated so far. For example, activation of PreNMDARs by intrathecal NMDA injection evoked substance P release to elicit pain in one study[26], but not another[27]. Acute exogenous NMDA decreased[28] or had no effect[29] on glutamate release in normal rats, but increased them in opiate-tolerant[30] and neuropathic rats[29,31,32]. Thus, we aimed to elucidate exactly how PreNMDARs modulate spinal presynaptic plasticity and pain hypersensitivity and which cellular processes underlie these actions.

Utilizing transgenic mice with specific deletion of functional NMDARs from presynaptic nociceptor terminals, we report here that PreNMDARs modulate spinal presynaptic plasticity in a nociceptive tone-dependent manner. In the basal state, activation of PreNMDARs leads to presynaptic depression, but converts to evoking presynaptic potentiation upon peripheral inflammation. This reveals the functional switch of PreNMDARs in the pain system. Further mechanistic analysis revealed that this nociceptive tone-dependent transition is dependent on the $Ca^{2+}$ influx through functional PreNMDARs in spinal nociceptor terminals. Activation of small-conductance $Ca^{2+}$-activated $K^+$ channels (SK channels) via $Ca^{2+}$ influx through PreNMDARs is responsible for presynaptic depression in the basal state. Rather, tissue inflammation induces PreNMDARs-PKG-I (cGMP-dependent protein kinase G I)-dependent BDNF (brain-derived neurotrophic factor) secretion, leading to downregulation of SK channels, which in turn serves to convert presynaptic depression to potentiation. Our findings shed light on the state-dependent characteristics of PreNMDARs in spinal nociceptor terminals on modulating pain signals transmission from the periphery to the central nervous system and revealed a mechanism underlying this state-dependent switch. Moreover, we identify PreNMDARs in spinal nociceptor terminals as key constituents of activity-dependent secondary pain hypersensitivity associated with tissue injury.

## Results

**SNS-Cre-mediated conditional deletion of NR1 subunit of NMDARs in presynaptic nociceptive dorsal root ganglion (DRG) neurons.** We generated mice lacking NR1 subunit of NMDARs in a primary nociceptor-specific manner (SNS-NR1$^{-/-}$) via Cre/loxP-mediated recombination by mating SNS-Cre mice under the control of the Nav1.8 promoter[33,34] with mice carrying the floxed *nr1* allele (NR1$^{fl/fl}$). An anti-NR1 antibody showed immunoreactivity in small-, medium- as well as large-diameter DRG neurons in NR1$^{fl/fl}$ mice (Supplementary Fig. 1a, b). In contrast, in SNS-NR1$^{-/-}$ mice, only large-diameter DRG neurons continued to show NR1 expression whereas small- to medium-diameter neurons were largely devoid of anti-NR1 immunoreactivity (Supplementary Fig. 1a, b). Confocal analysis revealed a colocalization of NR1 in isolectin B4 (IB4)-labeled non-peptidergic nociceptors and CGRP-expressing peptidergic nociceptors in NR1$^{fl/fl}$ mice, both of which are selectively lost in SNS-NR1$^{-/-}$ mice (Supplementary Fig. 1a, c). In contrast, large-diameter neurofilament-200-immunoreactive neurons entirely retained NR1 expression in the SNS-NR1$^{-/-}$ mice (Supplementary Fig. 1a, c). In the spinal cord, anti-NR1 immunoreactivity was largely decreased in the superficial dorsal horn in SNS-NR1$^{-/-}$ mice, as would be expected from SNS-Cre-mediated gene deletion in nociceptor terminals (Supplementary Fig. 1d). In contrast, intrinsic neurons in the spinal cord entirely maintained NR1 immunoreactivity (Supplementary Fig. 1d). NR1 expression was entirely unaltered in the brains of SNS-NR1$^{-/-}$ mice (Supplementary Fig. 1d). Western blot analysis on DRG, spinal cord, and brains of NR1$^{fl/fl}$ mice and SNS-NR1$^{-/-}$ mice confirmed a DRG-specific loss of NR1 in SNS-NR1$^{-/-}$ mice (Supplementary Fig. 1e–g, uncropped gels are not available, see "Methods" for details).

NR1 is the essential subunit of NMDARs, deletion of NR1 subunit in nociceptive DRG neurons would render them without functional NMDARs. To test this, we performed whole-cell patch-clamp recordings on whole-mount DRG preparations derived from NR1$^{fl/fl}$ and SNS-NR1$^{-/-}$ mice using Fluro488-conjugated Isolectin B4 (IB4-Fluro488) for live identification of small nociceptive DRG neurons (Supplementary Fig. 2a). Bath application of NMDA (250 μM) elicited a robust inward current in IB4-labeled DRG neurons from NR1$^{fl/fl}$ mice, which was blocked by pretreatment of NMDAR antagonist, AP-5 (50 μM), confirming the involvement of NMDARs (Supplementary Fig. 2b, c). In contrast, no obvious current was induced in IB4-labeled DRG neurons from SNS-NR1$^{-/-}$ mice (Supplementary Fig. 2b, c). In parallel, the deletion of functional NMDARs in SNS-NR1$^{-/-}$ mice was also confirmed in CGRP-expressing peptidergic neurons (Supplementary Fig. 2d). Furthermore, the sensitivity of this inward current to open channel blocker MK801, but not to $^{10}$panx, a Pannexin 1-blocking peptide suggests

that this inward current is mediated by open ionotropic NMDARs, but not by pannexin channels through a metabotropic signaling mode of NMDARs[35] (Supplementary Fig. 2e, f). In support of specific deletion of functional NMDARs in nociceptors, we showed that NMDA-induced inward current was identical in spinal neurons derived from NR1$^{fl/fl}$ and SNS-NR1$^{-/-}$ mice (Supplementary Fig. 2g). Other receptors' function such as kainate receptors remained intact as well after the deletion of NR1 in nociceptors (Supplementary Fig. 2h). Taken together, SNS-NR1$^{-/-}$ mice showed a nociceptor-specific loss of NR1 subunit and functional NMDARs while retaining expression and function in neurons of the central nervous system.

We next addressed whether SNS-Cre-mediated NR1 deletion induces aberrant developmental defects in the nociceptive DRG neurons. Double immunofluorescence results revealed that the number of IB4-labeled non-peptidergic DRG neurons and CGRP-expressing peptidergic DRG neurons were not altered by deletion of NR1 subunit in nociceptors (Supplementary Fig. 2i). Furthermore, central and peripheral patterning of peptidergic or non-peptidergic nociceptors was normal in adult SNS-NR1$^{-/-}$ mice (Supplementary Fig. 2j). We conclude therefore that SNS-NR1$^{-/-}$ mice show intact development of spinal and peripheral sensory circuits and went on to assess functional changes in these mice.

**Requirement of PreNMDARs in spinal terminals of nociceptors for pre-LTP at synapses between C-nociceptors and spinal-PAG projection neurons.** Previous reports demonstrated that LTP evoked by natural, asynchronous low-rate discharge at synapses between C-nociceptors and spinal lamina I neurons projecting to the periaqueductal gray (PAG) is post-synaptically induced and depends on NMDARs[3,4]. To investigate whether pre-LTP may be induced at spinal synapses, we employed a stimulating protocol for inducing pre-LTP in the amygdala and ACC[8,10,34]. We recorded evoked excitatory postsynaptic currents (eEPSCs) in spinal-PAG projection neurons at a holding potential of −70 mV by stimulation of the dorsal root. Spinal-PAG projection neurons were retrogradely labeled upon stereotactic injection of retrograde fluorescent marker DiI (1,1′-dioctadecyl-3,3,3′,3′-tetramethylindocarbocyanine perchlorate) in the PAG (Fig. 1a). In spinal-PAG projection neurons of wild-type mice, a conditioning low-frequency stimulation (LFS) of 2 Hz for 2 min at a holding potential of −70 mV produced pre-LTP of mono-synaptic C-fiber evoked EPSCs (C-eEPSCs) by more than 150% at 30 min (Fig. 1b). To test whether pre-LTP requires a postsynaptic function of NMDARs, we dialyzed NMDARs blocker, MK-801 (1 mM), into spinal neurons via the patch pipette, which has been verified to be able to block postsynaptic NMDARs-induced inward current (Supplementary Fig. 3a, b). These manipulations did not significantly affect the magnitude or duration of C-fiber-evoked pre-LTP at spinal-PAG synapses (Fig. 1b), suggesting that NMDARs localized post-synaptically in spinal-PAG projection neurons does not play a role in pre-LTP at this synapse.

To assess the role of PreNMDARs localized in spinal (central) terminals of nociceptors, we then analyzed NR1$^{fl/fl}$ mice and SNS-NR1$^{-/-}$ mice. In spinal-PAG projection neurons of NR1$^{fl/fl}$ mice, a conditioning LFS produced pre-LTP with a magnitude of 147.1 ± 20.6% at 30 min (Fig. 1c, d). Prior to LFS, baseline values of C-eEPSCs stayed constant and comparable between both genotypes (Fig. 1c, e). In striking contrast, the same LFS did not evoke pre-LTP in SNS-NR1$^{-/-}$ mice (Fig. 1c–f). In further support, bath application of AP5 (50 μM), an NMDARs antagonist, completely blocked the occurrence of pre-LTP in NR1$^{fl/fl}$ mice (Supplementary Fig. 3c, d). It also holds true for an NR1-selective antagonist, CGP78608 (1 μM) (Supplementary Fig. 3e, f). In contrast, this pre-LTP remained intact in the presence of kainite receptors antagonist, UBP310 (10 μM)

(Supplementary Fig. 3g, h). These results indicate that a loss of PreNMDARs was linked to a failure of activity-dependent pre-LTP at synapses between nociceptors and spinal-PAG projection neurons.

To further consolidate the presynaptic mechanisms of spinal pre-LTP and a role for PreNMDARs, we employed two protocols which are commonly used as a measure for presynaptic function[36]. First, a protocol of minimal stimulation was performed to study synaptic events which could be clearly assigned to activation of presynaptic primary afferent fibers alone, by setting the dorsal root stimulation parameters at which a synaptic failure rate of approximately 60% was achieved in recording solution containing 1 mM Ca$^{2+}$ and 5 mM Mg$^{2+}$. In the absence of conditioning LFS, the failure rate remained stable upon repetitive test stimulation. However, upon application of conditioning LFS, minimal stimulation induced a dramatic reduction in the failure rate in slices derived from NR1$^{fl/fl}$ mice, suggesting a change in the probability of neurotransmitter release (Fig. 1g, h). In striking contrast, the synaptic failure rate in SNS-NR1$^{-/-}$ mice was not altered significantly following conditioning LFS (Fig. 1g, h).

Second, we performed the paired-pulse ratio (PPR) analysis, which represents a short-term plasticity and is well accepted as an indication of presynaptic mechanisms of LTP in the hippocampus[37]. In spinal slices derived from NR1$^{fl/fl}$ and SNS-NR1$^{-/-}$ mice, we recorded paired-pulse facilitation (PPF) as well as paired-pulse depression (PPD) prior to conditioning LFS. Previous work has shown that PPR (either PPF or PPD) can decrease as well as increase in conjunction with LTP in a manner inversely proportional to the PPR prior to the conditioning stimulus in brain regions, e.g. hippocampus[37], spinal cord[34,38]. Upon application of LFS, a majority of neurons from NR1$^{fl/fl}$ mice demonstrated a clear change in PPF or PPD (Fig. 1i, upper panels). In contrast, neurons from SNS-NR1$^{-/-}$ mice did not (Fig. 1i, lower panels). We plotted the paired-pulse ratio (PPR) of the entire cohort of recorded neurons at 30 min after LFS as a function of the basal PPR recorded prior to LFS (Fig. 1j). Quantification analysis revealed that 92% (12 out of 13) of PPF-expressing neurons and 39% (7 out of 18) of PPD-expressing neurons in NR1$^{fl/fl}$ mice displayed a marked reduction in PPF or PPD after conditioning stimulus, which is indicative of an increase in the probability of release (Supplementary Fig. 4a, left and middle panels). The other 61% (11 out of 18) of PPD-expressing neurons exhibited an increase of PPD after conditioning stimulus (Supplementary Fig. 4a, right panels). This increase of PPD was also reported previously in opioid withdrawal-induced spinal LTP[38] as well as other systems[39,40]. This increase in PPD after LFS may suggest that LFS induced an increase in readily releasable pool of synaptic vesicles, which in turn results in more synaptic vesicles that can be released from nerve terminals in response to the second stimulus. Furthermore, we plotted the magnitude of LTP to the change of PPR and observed that in NR1$^{fl/fl}$ mice, higher magnitudes of LTP were consistently linked with a big change in PPR (decrease/increase), which is indicative of presynaptic mechanisms (Supplementary Fig. 4b). In striking contrast, these changes in PPF or PPD were not observed or greatly impaired in neurons from SNS-NR1$^{-/-}$ mice (Supplementary Fig. 4c, d). To evaluate the change of PPF and PPD altogether, we established a parameter which can reflect the magnitude of PPR change following LFS, that is the vertical distance of each spot shown in Fig. 1j to the diagonal line (Fig. 1j), as reported in a previous study[41]. The spots falling in the diagonal line represent no significant change of PPR. Quantification analysis revealed that NR1$^{fl/fl}$ mice showed much stronger changes of PPR after LFS, as compared to SNS-NR1$^{-/-}$ mice (Fig. 1k), indicating again of the presynaptic mechanisms involved. Taken together, the failure rate and PPR analysis collectively suggest that activity-dependent pre-LTP between

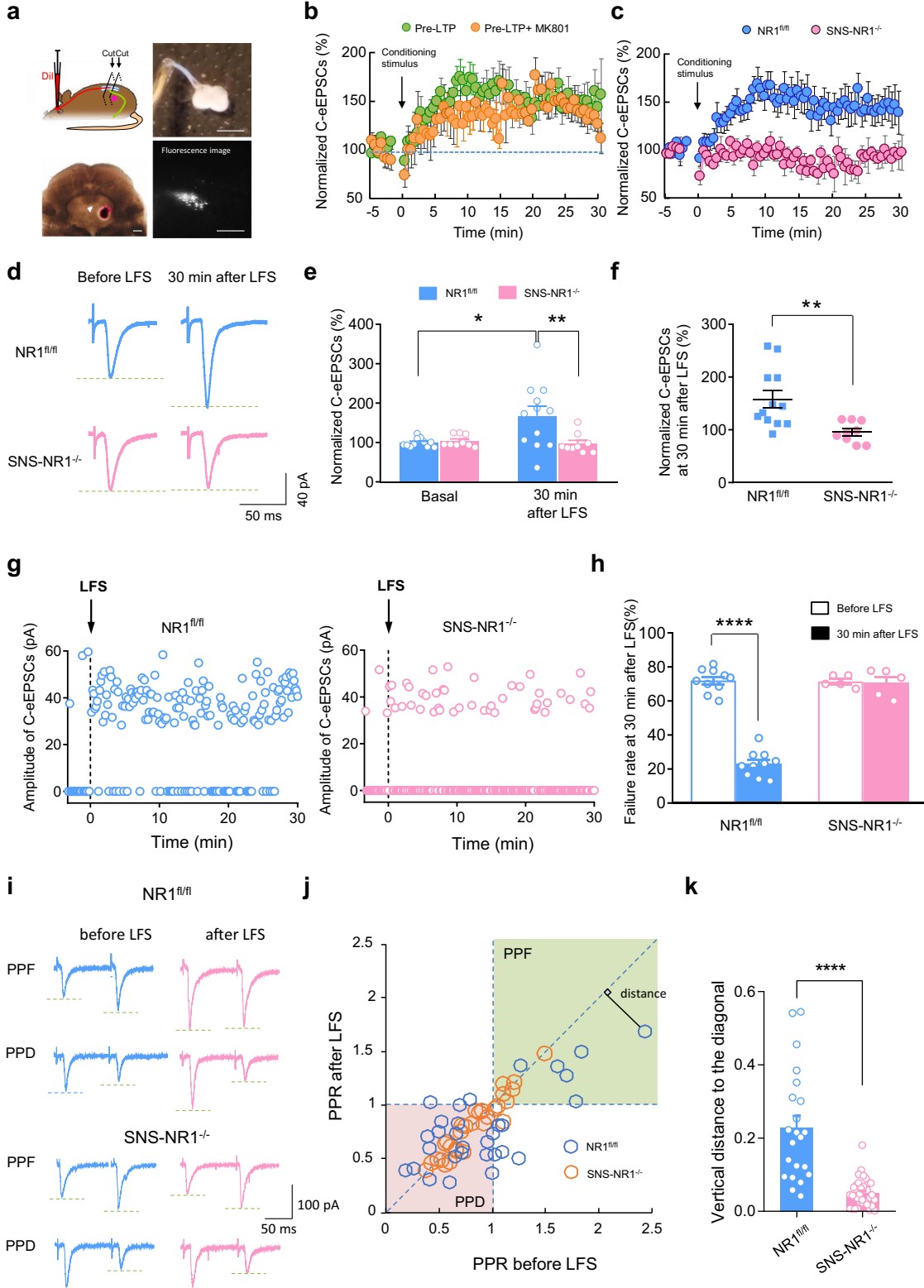

nociceptors and spinal-PAG projection neurons comes about via presynaptic mechanisms largely involving an increase of release probability via PreNMDARs.

**Evidence for a role of PreNMDARs on exogenous NMDA-induced modulation of synaptic transmission.** To further confirm the key role of PreNMDARs on spinal synaptic plasticity, we went on to test the effect of exogenous NMDA. Given a crucial role of PreNMDARs in spinal pre-LTP, we expect that activation of PreNMDARs by exogenous NMDA would potentiate synaptic transmission between C nociceptors and spinal-PAG projection neurons. However, acute bath application of NMDA (50 μM, 30 s) depressed but not potentiated C-eEPSCs in spinal slices from naive NR1^fl/fl mice (Fig. 2a, b, 65.5 ± 7.1% of a pre-drug

**Fig. 1 Presynaptic long-term potentiation (pre-LTP) in the spinal lamina I neurons of SNS-NR1$^{-/-}$ mice and NR1$^{fl/fl}$ mice, and analysis of its presynaptic mechanism. a** Schematic diagram of the experimental approach for dorsal root stimulation and whole-cell patch-clamp recordings from spinal-PAG projection neurons in lamina I retrogradely labeled by DiI injection into contralateral ventrolateral PAG. **b** Spinal pre-LTP was observed following low-frequency stimulation (LFS) (2 Hz) of dorsal roots at a holding potential of −70 mV in wild-type mice (green filled circles), which was preserved in the blockade of postsynaptic NMDARs via application of MK-801 in the patch pipette (yellow filled circle) ($n = 12$ for Pre-LTP, $n = 9$ for Pre-LTP + MK801). **c, d** Time course (**c**) and representative traces (**d**) and quantitative summary (**e, f**) of spinal pre-LTP evoked by conditioning LFS in NR1$^{fl/fl}$ mice, but not in SNS-NR1$^{-/-}$ mice. $n = 12$ for NR1$^{fl/fl}$, $n = 9$ for SNS-NR1$^{-/-}$. *$P < 0.05$, **$P < 0.01$ by Kruskal–Wallis $H$ test in (**e**), **$P < 0.01$ by unpaired $t$ test in (**f**). **g** Persistent activation of C nociceptors potentiates synaptic transmission by decreasing synaptic failure rate via activation of PreNMDARs in the spinal nociceptor terminal. Representative examples of the frequency of synaptic failures (0 pA) and synaptic successes (C-eEPSCs) evoked by minimal stimulation prior to and after delivery of the conditioning LFS in NR1$^{fl/fl}$ mice and SNS-NR1$^{-/-}$ mice. **h** Summary of averaged failure rates upon minimal stimulation of dorsal roots prior to and 30 min following LFS ($n = 10$ slices for NR1$^{fl/fl}$ mice and 5 slices for SNS-NR1$^{-/-}$ mice). ****$P < 0.0001$ by Uncorrected Fisher's LSD one-way ANOVA. **i** Traces of typical recordings showing PPF or PPD of C-eEPSCs induced by pairs of pulses with an interval of 110 ms prior to (blue traces) and at 30 min following LFS (pink traces). **j** Paired-pulse ratio (PPR) prior to LFS is plotted against PPR at 30 min after LFS in NR1$^{fl/fl}$ mice and SNS-NR1$^{-/-}$ mice. **k** C-eEPSCs recorded in NR1$^{fl/fl}$ mice showed clear change of PPR following LFS, which is significantly different from SNS-NR1$^{-/-}$ mice. $n = 22$-35. ****$P < 0.0001$ by Mann–Whitney $U$ test. Scale bars represent 100 μm in left lower panel, 2 mm in right upper panel and 10 μm in right lower panel in (**a**). Data are represented as mean ± S.E.M. See Supplemental Table 2 for detailed statistical information. LFS, low-frequency stimulation; PPR, paired-pulse ratio; PPF, paired-pulse facilitation; PPD, paired-pulse depression.

level). This NMDA effect was presynaptic in origin since MK-801 (1 mM) was present in the patch pipette for blockade of postsynaptic NMDARs. Meanwhile, obvious change in PPR of C-eEPSCs upon acute NMDA application supports a presynaptic mechanism involving a decrease of release probability (Supplementary Fig. 5a-c). Furthermore, long-term perfusion of NMDA (50 μM, 10 min) produced a long-term depression (LTD) of C-eEPSCs in NR1$^{fl/fl}$ mice, with 58.3 ± 7.9% of inhibition at 30 min after washout (Fig. 2c, d). This NMDA-induced LTD was associated with the obvious change of PPR as well (Supplementary Fig. 5d–f). In striking contrast, neither acute nor long-term treatment with NMDA influenced either the magnitude or PPR of C-eEPSCs in SNS-NR1$^{-/-}$ mice (Fig. 2a-d, Supplementary Fig. 5a–f). These results indicate that activation of PreNMDARs in spinal nociceptor terminals depressed glutamate release under basal state via a presynaptic mechanism. Similar to PreNMDARs, negative regulation of spinal synaptic transmission has also been observed in presynaptic kainate receptors (KARs)[42].

We then asked whether and how PreNMDARs from spinal nociceptor terminals regulate synaptic transmission under pathological pain states. Chronic inflammatory pain was produced by unilateral injection of Complete Freund's Adjuvant (CFA) in the hindpaw. At 24 h post-CFA injection in NR1$^{fl/fl}$ mice, acute application of NMDA (50 μM, 30 s) produced a pronounced potentiation of C-eEPSCs instead of depression (Fig. 2e, f, 166.5 ± 23.1% of control). This potentiation was accompanied by a significant change in PPR, confirming the origin of presynaptic action (Supplementary Fig. 5g–i). Similarly, long-term exposure to NMDA (50 μM, 10 min) caused a long-term potentiation of C-eEPSCs associated with obvious changes in PPR (Fig. 2g, h, Supplementary Fig. 5j–l). Either transient or long-term potentiation of C-eEPSCs by NMDA was not observed in SNS-NR1$^{-/-}$ mice (Fig. 2e–h, Supplementary Fig. 5g–l). The above results led us to infer that PreNMDARs in spinal terminals of nociceptors regulate synaptic transmission in a nociceptive tone-dependent manner, i.e. depressing synaptic transmission under basal state but potentiating under pathological state. This inference was further supported by the conversion of synaptic transmission from depression to potentiation by PreNMDARs in the same spinal-PAG projection neurons. As shown in Fig. 2i–k, in slices from naive wild-type mice, bath-applied NMDA significantly inhibited C-eEPSCs in spinal-PAG neurons (Fig. 2i–k). Upon washout, conditioning LFS was delivered to the dorsal root to evoke LTP of C-eEPSCs which mimics the pathological state. Similar to reported above, C-eEPSCs were potentiated to 151.9 ± 23.5 % of control at 30 min post-conditioning LFS. Interestingly, a second application of NMDA at

this point in the same neuron was found to potentiate but not depress C-eEPSCs as compared to pre-drug level (Fig. 2i–k).

To further consolidate PreNMDARs-mediated state-dependent modulation of presynaptic transmission, we went on to see how blocking endogenous PreNMDARs affects presynaptic transmission in the basal and inflammatory state. Consistently, we observed that in NR1$^{fl/fl}$ mice, blockade of endogenous PreNMDARs with bath application of AP5 (50 μM) caused a robust facilitation of C-eEPSCs (210 ± 23.7% of control) in the basal state, while it led to a dramatic depression (54.6 ± 5.1% of control) in CFA-inflamed state (Fig. 3a–c). Both facilitation and depression of C-eEPSCs by AP5 in the basal and inflammatory state were accompanied by a clear change in PPR (Supplementary Fig. 6a, b, Fig. 3d, e), which indicates a presynaptic origin of AP5. In contrast, none of the facilitation or depression of C-eEPSCs by AP5 was observed in SNS-NR1$^{-/-}$ mice in either state (Fig. 3a–c). In parallel, no obvious changes in PPR of C-eEPSCs were seen after application of AP5 in SNS-NR1$^{-/-}$ mice in both states as well (Supplementary Fig. 6a, b, Fig. 3d, e). These results infer that endogenous PreNMDARs exerted a state-dependent modulation of presynaptic transmission via a presynaptic mechanism. This inference was further consolidated by miniature EPSCs (mEPSCs) recording in the presence of TTX (0.5 μM), gabazine (50 μM), strychnine (10 μM), as described routinely. As shown in Fig. 3f, in spinal-PAG projection neurons from NR1$^{fl/fl}$ mice, delivery of AP5 (50 μM) induced a drastic increase in mEPSCs frequency, but no alteration in mEPSCs amplitude in the basal state (Fig. 3f). Conversely, upon peripheral inflammation, AP5 application led to significant depression of mEPSCs frequency, but no change of mEPSCs amplitude (Fig. 3h). This state-dependent regulation of mEPSCs by AP5 was eliminated in SNS-NR1$^{-/-}$ mice (Fig. 3g, i). Postsynaptic NMDARs were blocked during the above recordings by the addition of MK 801 (1 mM) in the patch pipette. However, considering the presence of presynaptic Nav1.8 channels, TTX-resistant sodium channels in nociceptors, mEPSCs were further recorded and analyzed in the presence of A-803467 (0.5 μM), an Nav1.8 selective blocker besides TTX, gabazine and strychnine. Similar results were observed in this recording paradigm as shown in the above (Supplementary Fig. 7a–d). Taken together, both exogenous and endogenous evidence collectively strengthen the state-dependent modulation of presynaptic transmission by PreNMDARs in nociceptor terminals via a presynaptic mechanism involving transmitter release.

**Functional expression of PreNMDARs in spinal terminals of nociceptors.** Since transmitter release is known to be Ca$^{2+}$

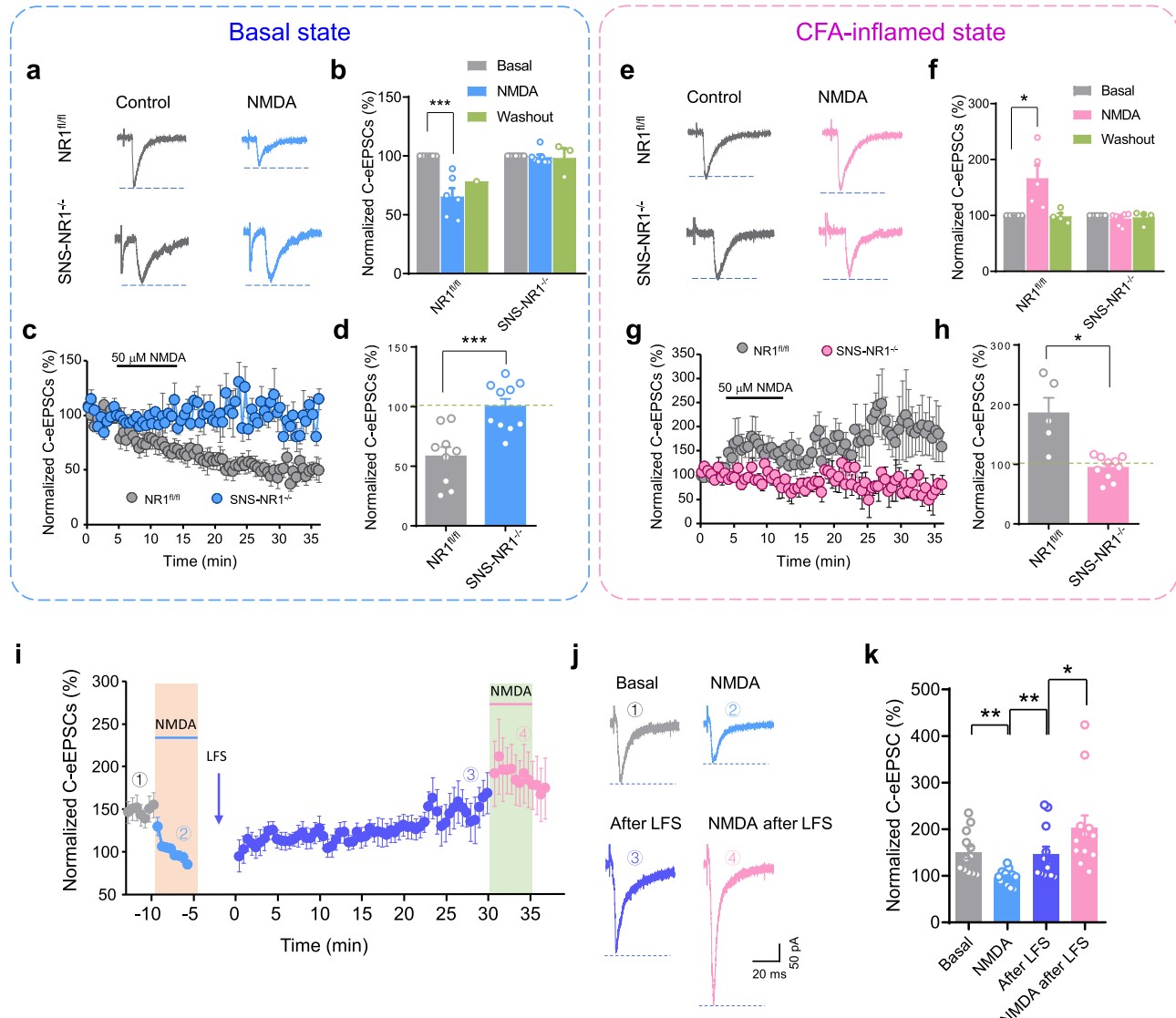

**Fig. 2 Exogenous NMDA-induced presynaptic depression of C-eEPSCs in the basal state, but presynaptic potentiation in the inflamed state. a, b** Typical examples (**a**) and quantitative summary (**b**) of synaptic depression of C-eEPSCs induced by acute bath application of NMDA (50 μM, 30 s) in NR1$^{fl/fl}$ mice, but not in SNS-NR1$^{-/-}$ mice in the basal state ($n$ = 6–7). ***$P$ < 0.001 by Kruskal–Wallis $H$ test. **c, d** Time course (**c**) and quantitative summary (**d**) showing pre-LTD of C-eEPSCs evoked by long-term exposure to NMDA (50 μM, 10 min) in NR1$^{fl/fl}$ mice, but not in SNS-NR1$^{-/-}$ mice in the basal state ($n$ = 9–10). ***$P$ < 0.001 by unpaired $t$ test. **e, f** Following CFA-evoked paw inflammation, acute bath application of NMDA led to presynaptic potentiation of C-eEPSCs in NR1$^{fl/fl}$ mice, but not in SNS-NR1$^{-/-}$ mice (representative examples are shown in e, quantitative summary shown in (**f**), $n$ = 5–7). *$P$ < 0.05 by Kruskal–Wallis $H$ test. **g, h** Typical traces of recordings (**g**) and quantitative summary (**h**) showing long-term exposure of NMDA (50 μM, 10 min) evoked pre-LTP of C-eEPSCs in CFA-inflamed NR1$^{fl/fl}$ mice, but not in inflamed SNS-NR1$^{-/-}$ mice ($n$ = 5–10). *$P$ < 0.05 by unpaired $t$ test. **i, j** Time course (**i**) and representative recording traces (**j**) as well as quantitative summary (**k**) showing the transition of NMDA-induced synaptic depression to potentiation following LFS-induced pre-LTP in the same spinal-PAG projection neuron ($n$ = 12). MK-801 was present in the patch pipette to block postsynaptic NMDARs in the above recordings. *$P$ < 0.05, **$P$ < 0.01 by Kruskal–Wallis $H$ test. Data are represented as mean ± S.E.M. See Supplemental Table 2 for detailed statistical information.

dependent, enhanced Ca$^{2+}$ influx via PreNMDARs may account for the requirement of functional PreNMDARs for spinal synaptic transmission and plasticity. However, the subcellular localization of functional NMDARs on DRG neurons has been a matter of debate for some time[43–45]. We examined this possibility by direct measurement of Ca$^{2+}$ changes in presynaptic terminals, using Ca$^{2+}$ sensor GCaMP6s[46] specifically expressed in presynaptic spinal nociceptor terminals. An AAV2/8 vector containing Cre-dependent GCaMP6s construct was injected into DRG of SNS-Cre and SNS-NR1$^{-/-}$ mice, and spinal slices of injected mice were examined 4–6 w after virus injection (Fig. 4a). Bath-applied NMDA (50 μM) induced a dramatic elevation of

GCaMP6s fluorescence in spinal nociceptor terminals from SNS-Cre mice that decayed to the baseline after washout, which was sensitive to pretreatment with NMDARs antagonist, AP5 (50 μM) (Fig. 4b, c, Supplementary Fig. 8a, b). In contrast, almost no fluorescence changes were found in SNS-NR1$^{-/-}$ mice (Fig. 4b–d). Furthermore, the magnitude of calcium gradients evoked by glutamate was reduced by nearly 42.8% in the presence of AP5 in SNS-Cre mice or 45.7% in SNS-NR1$^{-/-}$ mice (Fig. 4c, d, Supplementary Fig. 8c, d). In contrast, rapid Ca$^{2+}$ influx caused by KCl-induced depolarization or capsaicin-evoked influx of Ca$^{2+}$ via TRPV1 channels was comparable in presynaptic nociceptor terminals derived from SNS-Cre and SNS-NR1$^{-/-}$

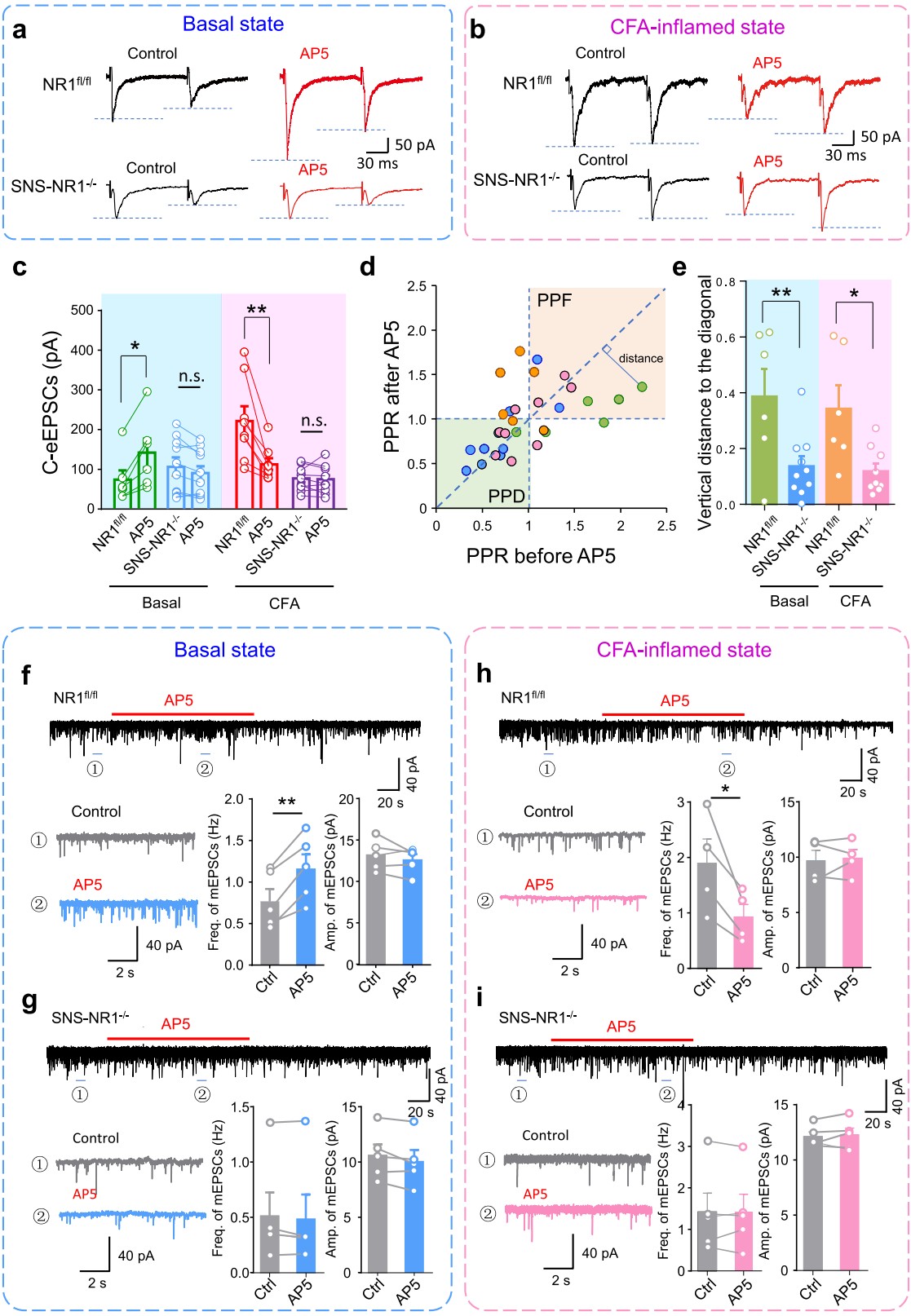

mice (Fig. 4c, d). These results indicate that NMDARs are functionally expressed in the presynaptic spinal terminals of nociceptors.

We further examined the possibility that spinal pre-LTP requires the enhancement of $Ca^{2+}$ caused by activation of functional PreNMDARs from spinal nociceptor terminals. Electrical stimulation of dorsal root with LFS pattern (2 Hz, 240

times, 3 mA) evoked a prolonged increase of $Ca^{2+}$ gradients in presynaptic terminals of nociceptors in SNS-Cre mice, peaking at 130 s upon the onset of LFS (Fig. 4e, f). Pretreatment with NMDARs antagonist, AP5 (50 μM) abolished LFS-induced long-lasting $Ca^{2+}$ accumulation, confirming the requirement of functional PreNMDARs for spinal pre-LTP induction (Fig. 4e±g). To further confirm the role of PreNMDARs in LFS-induced $Ca^{2+}$

**Fig. 3 Blockade of endogenous PreNMDARs with AP5 produced presynaptic potentiation in the basal state, but presynaptic depression in the CFA-inflamed state. a** Typical examples of synaptic potentiation of C-eEPSCs induced by bath application of AP5 (50 μM) in NR1$^{fl/fl}$ mice in the basal state, but not in SNS-NR1$^{-/-}$ mice ($n = 6$–10). **b** Upon paw inflammation with CFA injection, endogenous blockade of PreNMDARs with AP5 led to synaptic depression in NR1$^{fl/fl}$ mice, but not in SNS-NR1$^{-/-}$ mice ($n = 8$–9). **c** Showing quantitative summary for (**a**) and (**b**). *$P < 0.05$, **$P < 0.01$ by Kruskal–Wallis *H* test. **d** Paired-pulse ratio (PPR) prior to AP5 is plotted against PPR after AP5 in NR1$^{fl/fl}$ mice and SNS-NR1$^{-/-}$ mice in either basal state or CFA-inflamed state. **e** AP5 induced clear change of PPR in NR1$^{fl/fl}$ mice in both states, which is significantly different from SNS-NR1$^{-/-}$ mice ($n = 6$–10). *$P < 0.05$, **$P < 0.01$ by uncorrected Fisher's LSD one-way ANOVA. **f, g** Representative examples and quantitative summary showing bath application of AP5 (50 μM) caused a prominent increase in the frequency, but not amplitude of mEPSCs recorded in spinal neurons in NR1$^{fl/fl}$ mice in the basal state (**f**), whereas this did not come about in SNS-NR1$^{-/-}$ mice (**g**) ($n = 5$). **$P < 0.01$ by paired *t* test. **h, i** Following CFA-induced paw inflammation, AP5 application induced a marked reduction of mEPSCs frequency in NR1$^{fl/fl}$ mice (**h**), which was not observed in SNS-NR1$^{-/-}$ mice (**i**) ($n = 4$–5). *$P < 0.05$ by paired *t* test. Data are represented as mean ± S.E.M. See Supplemental Table 2 for detailed statistical information. Ctrl, control.

elevation, we analyzed with SNS-Cre and SNS-NR1$^{-/-}$ mice. In comparison with the marked Ca$^{2+}$ response induced by LFS in SNS-Cre mice, the same LFS failed to induce the sustained Ca$^{2+}$ elevation in SNS-NR1$^{-/-}$ mice (Fig. 4e–g). Thus, activation of PreNMDARs is indeed required for LFS-induced presynaptic Ca$^{2+}$ accumulation.

**Involvement of small-conductance Ca$^{2+}$-activated K$^+$ (SK) channels in PreNMDARs-mediated synaptic depression in the basal state.** In an effort to understand the molecular mechanisms underlying state-dependent modulation of spinal synaptic transmission by PreNMDARs, we then addressed potential substrates for NMDARs in nociceptors under different states. Activation of NMDARs and voltage-gated Ca$^{2+}$ channels leads to activation of small-conductance Ca$^{2+}$-activated K$^+$ (SK) channels in cortical neurons[47]. A crucial role of cross-talk between NMDARs and SK channels has been reported in the determination of cortical and DRG neuronal excitability[48,49]. However, whether SK channels are involved in the PreNMDARs-mediated presynaptic depression at spinal synapses has remained elusive. In naive wild-type mice, in the presence of apamin (100 nM), a selective SK channels blocker, bath-applied NMDA (50 μM) failed to depress C-eEPSCs, but tended to potentiate C-eEPSCs (Fig. 5a, b). Postsynaptic SK channels and NMDARs were blocked during the recording by the addition of BAPTA (10 mM) and MK 801 (1 mM) in the patch pipette (Fig. 5a), indicating a presynaptic origin of the above apamin action. This result suggests that activation of presynaptic SK channels is functionally linked to PreNMDARs-mediated synaptic depression at spinal synapses in the basal state. This inference was further supported by the observed reduction in SK current in DRGs as well as SK2 subunit expression in DRGs and dorsal roots from naive SNS-NR1$^{-/-}$ mice as compared to NR1$^{fl/fl}$ mice (Fig. 5c–f, Supplementary Fig. 9a, b). As described previously, SK current was recorded as apamin-sensitive outward AHP (afterhyperpolarization) currents[49–51] (Supplementary Fig. 9c, d). AHP currents were largely attenuated in small DRG neurons which are devoid of NR1 subunit of NMDARs as compared to control neurons (Fig. 5c, d). Previous studies have reported that SK channels (SK1, SK2, and SK3) are expressed in DRG neurons, especially SK2 subunit showing high colocalization with NR1 subunit of NMDARs[49]. Furthermore, our ultrastructural pre-embedding double immunostaining revealed that both NR1 subunit of NMDARs and SK2 channels are localized in CGRP-positive presynaptic terminals of nociceptors (Supplementary Fig. 9e, f). Western blot analysis with lysates of lumbar DRGs and dorsal roots from naive SNS-NR1$^{-/-}$ mice revealed a reduction of SK2 expression as compared to NR1$^{fl/fl}$ mice (Fig. 5e, f, Supplementary Fig. 9a, b, uncropped gels are not available for Fig. 5e, see "Methods" for details). Taken together, these results infer us that activation of presynaptic SK2 channels was involved in the spinal presynaptic depression by PreNMDARs in the basal state, which

might serve as a negative feedback and filter to counteract more pain signals flowing into the central nervous system.

**Secretion of BDNF via PreNMDARs-PKG-I signaling pathway is involved in the PreNMDARs-mediated presynaptic potentiation in inflammatory pain states.** Then how would PreNMDARs come to facilitate spinal synaptic transmission in the inflammatory pain states? We first examined the changes of NMDARs and SK channels activity in the inflammatory pain states. At 24 h following CFA inflammation, NMDARs NR1 subunit in L4/L5 DRGs as well as dorsal roots showed a dramatic upregulation of phosphorylation over the basal state in NR1$^{fl/fl}$ mice (Fig. 5g, h, Supplementary Fig. 9g, h, uncropped gels are not available for Fig. 5g, see "Methods" for details). In striking contrast, SK2 expression in DRGs and dorsal roots was significantly reduced over the basal state in NR1$^{fl/fl}$ mice, which did not occur in SNS-NR1$^{-/-}$ mice (Fig. 5g, h, Supplementary Fig. 9g, h). Further functional analysis revealed that SK currents in small DRG neurons were largely attenuated in CFA-inflamed NR1$^{fl/fl}$ mice (Fig. 5i, j). This resultant downregulation of SK channel activity may contribute to relieve synaptic transmission from depression to potentiation by PreNMDARs in the pathological states. Then what could be the possible cause for SK channel downregulation after inflammation?

Numerous studies have documented the key significance of spinal NMDA receptor-NO-cGMP pathway in the synaptic plasticity and chronic pain states[52,53]. cGMP-dependent protein kinase G I (PKG-I) has been reported to be highly expressed in nociceptive primary sensory neurons in the DRG[34,54]. Whether and how presynaptic NMDARs-cGMP-PKG-I pathway in spinal nociceptor terminals modulates spinal synaptic transmission is still unclear. To address this question, we first set up an assay system for testing the involvement of NMDARs substrates PKG-I in the DRGs as well as dorsal roots selectively upon persistent nociceptive stimulation in vivo. In the basal state, lysates of DRGs as well as dorsal roots from naive SNS-NR1$^{-/-}$ mice showed a comparable PKG-I expression as compared to NR1$^{fl/fl}$ mice (Fig. 5k, l, Supplementary Fig. 9i, j). In striking contrast, following CFA inflammation, DRGs as well as dorsal roots from NR1$^{fl/fl}$ mice showed a striking increase of PKG-I expression over the basal state (Fig. 5k, l, Supplementary Fig. 9i, j). This was markedly reduced in CFA-injected SNS-NR1$^{-/-}$ mice (Fig. 5l, Supplementary Fig. 9i, j). These results indicate that persistent activation of nociceptors leads to rapid activation of NMDARs-PKG-I signaling cascades in the DRG. We further examined the role of downstream PKG-I signaling pathway in the PreNMDARs-mediated spinal presynaptic modulation. In naive PKG-I$^{fl/fl}$ mice, bath application of NMDA (50 μM) reliably depressed C-eEPSCs similar to wild-type mice (Fig. 5m, n). Interestingly, when pretreating the slice with PKG-I activator, cGMP analog 8-pCPT-cGMP (100 μM) to mimic increased activity of PKG-I in the inflamed states, bath-applied NMDA became to potentiate but

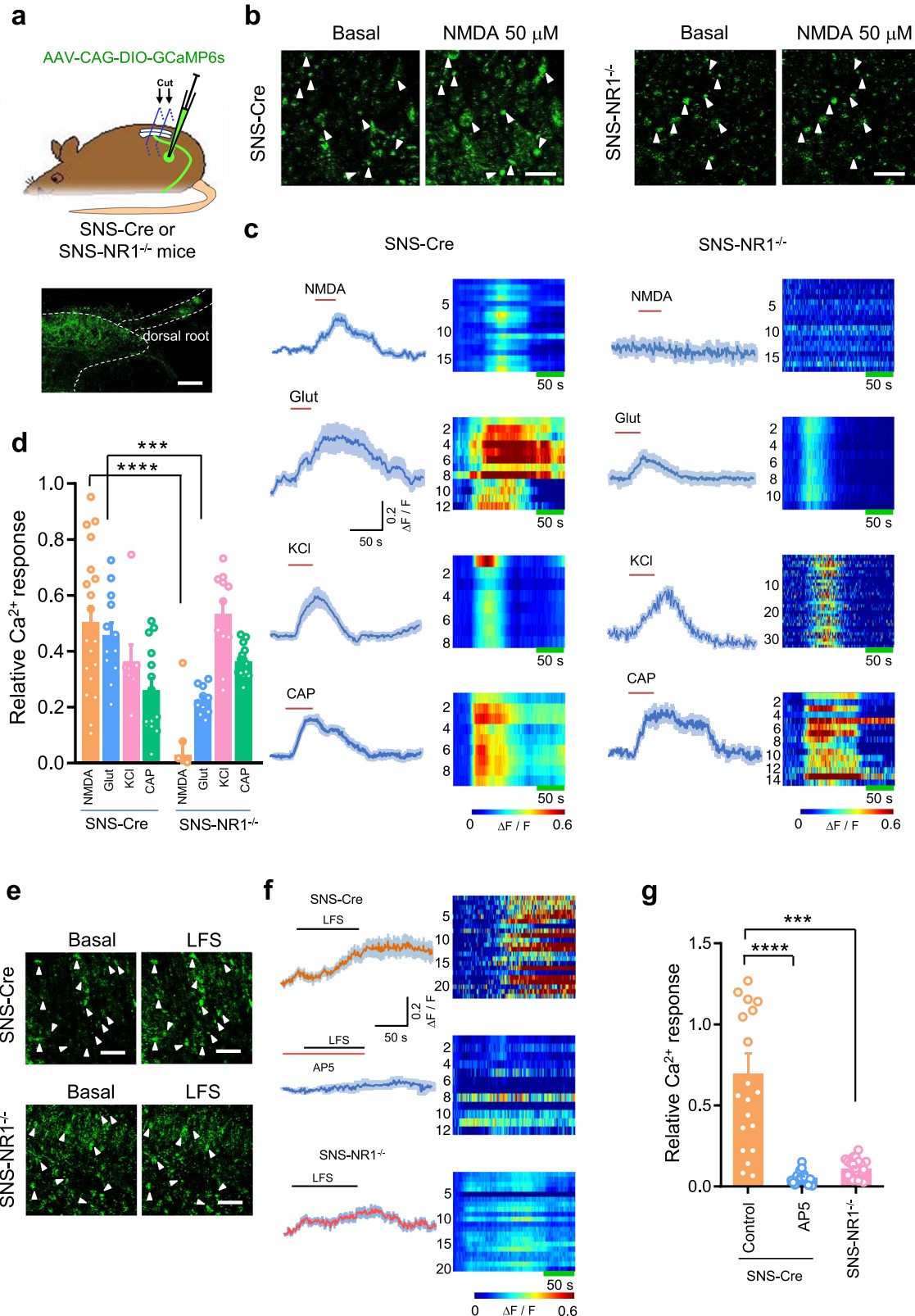

not depress C-eEPSCs (Fig. 5m, n). In striking contrast, specific deletion of PKG-I in nociceptors (SNS-PKG-I$^{-/-}$ mice) excluded NMDA-induced synaptic potentiation in the inflamed states (Fig. 5o, p). Thus, the NMDARs target, PKG-I, is functionally linked to presynaptic potentiation by NMDA in the pathological states. This is supported by our previous observation that presynaptic PKG-I in nociceptor terminals is a key determinant for spinal synaptic plasticity at synapses between nociceptors and spinal-PAG projection neurons[34]. Then what could be the downstream signaling targets of PKG-I to mediate PreNMDARs-dependent presynaptic potentiation?

Induction of synaptic potentiation at some glutamatergic synapses including spinal synapses depends on the action of BDNF[2,23,55,56]. However, whether BDNF is derived pre- or post-synaptically and

**Fig. 4 Super-resolution confocal calcium imaging showing functional expression of PreNMDARs in spinal terminals of nociceptors. a** Schematic demonstration of the experimental approach for specific expression of GCaMP6s in nociceptive DRG neurons and $Ca^{2+}$ imaging in the presynaptic spinal terminals of nociceptors. Below shown are typical examples of GCaMP6s expression in the presynaptic nociceptor terminals in the spinal dorsal horn. **b** Typical images of NMDA-induced changes in GCaMP6s fluorescence in presynaptic nociceptor terminals in spinal slices derived from SNS-Cre (left panels) and SNS-NR1$^{-/-}$ mice (right panels). **c, d** Typical examples of traces of calcium transients as shown in heat maps (**c**) and quantitative summary from 8 to 20 independent slice experiments of peak GCaMP6s signals (**d**) evoked by NMDA (50 μM), glutamate (Glut, 1 mM), capsaicin (CAP, 1 μM) and KCl (30 mM) in spinal slices from SNS-Cre and SNS-NR1$^{-/-}$ mice. Each row in y-axis represents a GCaMP6s-labeled presynaptic terminal puncta. A total of 8–20 puncta are illustrated for each agonist. ****$P < 0.0001$ by Kruskal–Wallis $H$ test. **e** Representative images of calcium transients in presynaptic nociceptor terminals evoked by delivering conditioning LFS to the dorsal root in SNS-Cre (upper panels) and SNS-NR1$^{-/-}$ mice (lower panels). **f, g** Typical traces and heat maps (**f**) as well as quantitative summary (**g**) of GCaMP6s-reflected calcium signals induced by conditioning LFS in SNS-Cre mice with the absence and presence of AP-5 (upper and middle panels), SNS-NR1$^{-/-}$ mice (lower panels) ($n = 17$–22). ***$P < 0.001$, ****$P < 0.0001$ by Brown–Forsythe ANOVA test. Data are represented as mean ± S.E.M. Scale bars represent 100 μm in (**a**), 20 μm in (**b**), 25 μm in (**e**). See Supplemental Table 2 for detailed statistical information. Glut, glutamate; CAP, capsaicin.

how PreNMDARs controls BDNF secretion at spinal synapses has remained elusive. In NR1$^{fl/fl}$ mice, persistent activation of nociceptor by hindpaw inflammation produced marked upregulation of BDNF expression in lumbar DRGs as well as dorsal roots over the basal state (Fig. 6a, b, Supplementary Fig. 10a, b, uncropped gels are not available for Fig. 6a, see "Methods" for details). This effect was markedly reduced in mice devoid of NR1 in nociceptors, indicating the dependence of BDNF expression upon activation of NMDARs in DRGs as well as dorsal roots (Fig. 6a, b, Supplementary Fig. 10a, b). Further analysis with SNS-PKG-I$^{-/-}$ mice revealed a crucial involvement of PKG-I in the NMDARs-dependent BDNF expression in DRGs as well as dorsal roots. CFA-injected PKG-I$^{fl/fl}$ mice demonstrated highly enhanced expression of BDNF in DRGs and dorsal roots over the basal state, which was found to be lacking in DRGs and dorsal roots obtained from inflamed SNS-PKG-I$^{-/-}$ mice (Fig. 6c, d, Supplementary Fig. 10c, d, uncropped gels are not available for Fig. 6c, see "Methods" for details). We then directly examined PreNMDARs-dependent BDNF secretion from spinal nociceptor terminals by expressing BDNF tagged with a pH-sensitive fluorescent protein (superecliptic pHluorin; BDNF-pH), which is known to undergo the same intracellular processing and exhibits similar biological activity as the native BDNF[57]. To restrict the expression of BDNF-pH in presynaptic nociceptor terminals, we injected an AAV2/8 vector containing double-floxed inversed BDNF-pH codon into the DRG of SNS-Cre mice (Fig. 6e). At 4 w after virus injection, we observed high-level BDNF-pH expression in spinal nociceptor terminals, with axonal BDNF-pH puncta largely localized to areas marked with the presynaptic marker synaptophysin and juxtaposed to the postsynaptic marker PSD-95 (Supplementary Fig. 10e–g). Using super-resolution confocal microscopy of dorsal root-attached spinal slices, activity-induced BDNF secretion was monitored by changes in the fluorescence intensity of BDNF-pH puncta. Upon LFS of dorsal root, most fluorescent puncta exhibited fusion with secretion, with an overall reduction of BDNF-pH fluorescence below the basal level in spinal slices derived from SNS-Cre mice (Fig. 6f–h). Such fluorescence reduction was abolished by AP-5 (50 μM) and diminished in SNS-NR1$^{-/-}$ mice (Fig. 6g, h). It also holds true for SNS-PKG-I$^{-/-}$ mice (Supplementary Fig. 10h–j). These results indicate that BDNF secretion from spinal nociceptor terminals was dependent on PreNMDARs and its downstream signaling target PKG-I.

To assess the role of BDNF on PreNMDARs-mediated synaptic potentiation in the pathological state, we depleted extracellular BDNF by adding soluble BDNF scavenger TrkB-IgG (2 μg/ml) prior to NMDA application in the spinal cord slice. As compared to vehicle, the addition of TrkB-IgG eliminated NMDA-potentiated C-eEPSCs in CFA-inflamed mice (Fig. 6i, j). The importance of axonal BDNF in PreNMDARs-mediated synaptic

potentiation in CFA-inflamed states was further examined by selectively eliminating BDNF expression with Cre-loxP deletion of Bdnf gene in spinal nociceptor terminals via injection of loxP-BDNF shRNA-expressing AAV2/8 into L3/L4 DRGs of SNS-Cre mice (Fig. 6k, Supplementary Fig. 11a, b)[58]. Immunofluorescence staining and western blot analysis confirmed a nociceptor-specific loss of BDNF by this virion strategy (Supplementary Fig. 11c, d). Bath application of NMDA failed to potentiate C-eEPSCs in CFA-inflamed mice with knockdown of BDNF at spinal nociceptor terminals (Fig. 6l, m). Furthermore, knockdown of BDNF in nociceptors prevented spinal pre-LTP induced by LFS of dorsal root (Fig. 6n–p). These results infer a crucial significance of BDNF secretion via NMDARs-PKG-I signaling pathway in PreNMDARs-mediated synaptic potentiation in the inflammatory state.

**Peripheral inflammation increases BDNF level in nociceptors to further suppress SK channel activity.** Finally, we asked whether BDNF secretion via PreNMDAR-PKG-I signaling pathway influences SK channel activity. To address this, we first tested the effect of BDNF on SK2 expression in the DRG. Incubation of lumbar DRGs from naïve mice with BDNF (100 ng/ml) for 1 h was found to greatly reduce SK2 expression as compared to vehicle treatment (Fig. 6q). Functional analysis with SK current recording showed that the same BDNF treatment produced a prominent inhibition of SK currents recorded from small DRG neurons (Fig. 6r, s). We can infer from the above that peripheral inflammation was able to increase BDNF production and secretion, which in turn inhibited SK channel activity, resulting in the conversion of PreNMDARs-mediated presynaptic depression to potentiation.

**Marked defects in the centrally mediated secondary mechanical hypersensitivity in nociceptor-specific NR1 deficient mice and BDNF knockdown mice.** We then went on to address whether the above synaptic changes bear significance to pain-related behavior in vivo and reveal a functional role of PreNMDARs and its downstream substrates in pain sensitization. Acute withdrawal responses to paw pressure (i.e. von Frey filament) and thermal stimuli (e.g. a radiant infrared heat ramp) were found to be comparable across SNS-NR1$^{-/-}$ mice and their littermate controls (Supplementary Fig. 12a, b). Acute nocifensive responses evoked by an intraplantar injection of capsaicin were not different between NR1$^{fl/fl}$ mice and SNS-NR1$^{-/-}$ mice (Supplementary Fig. 12c). This is in contrast to another study lacking NR1 in advillin-Cre mice (namely all DRG neurons)[49]. The discrepancy with advillin-NR1 knockout mice may be due to NR1 deletion in different populations of DRG neurons.

In the context of studying disease-related pain hypersensitivity, we focused on to explore whether PreNMDARs from nociceptor

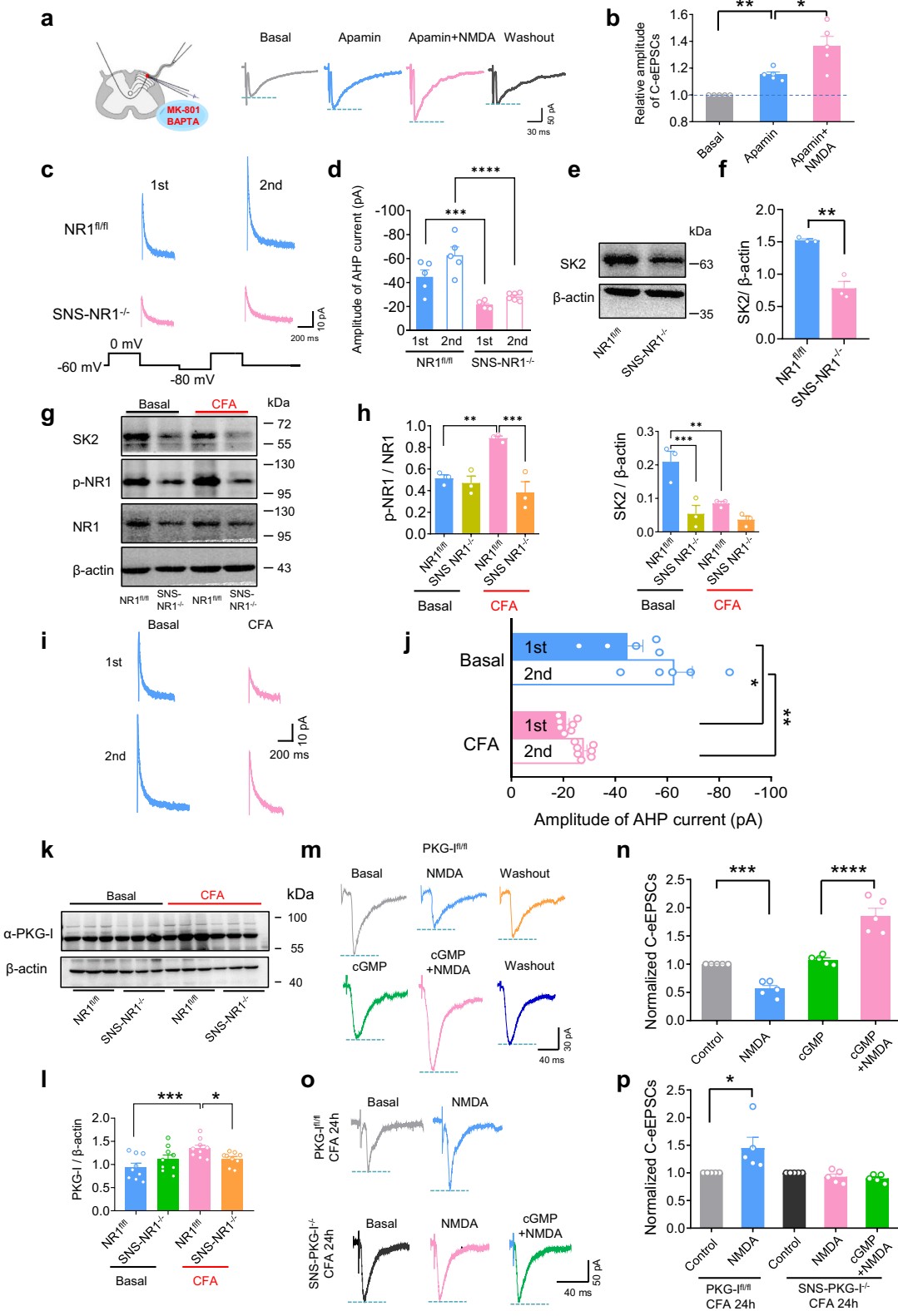

central terminals play a crucial role in the central mechanisms of secondary hypersensitivity associated with tissue injury. We utilized two models of aberrant pain which are particularly useful for studying central changes triggered initially by peripheral inputs but do not require ongoing nociceptor activity in the periphery for maintenance. Injection of capsaicin in the skin of lower thigh was able to produce a marked allodynia at the

hindpaw plantar surface (secondary hypersensitivity), which is out of the injured site (primary hyperalgesia)[34]. Upon capsaicin injection, NR1[fl/fl] mice demonstrated a dramatic secondary hypersensitivity, manifesting as obvious leftward and upward shift of stimulus-response curve over the basal curve (Fig. 7a) and prominent drop in mechanical response threshold to von Frey hairs (Fig. 7b). In contrast, this secondary hypersensitivity was

**Fig. 5 Molecular mechanisms underlying state-dependent modulation of spinal synaptic transmission by PreNMDARs in spinal terminals of nociceptors. a, b** Representative traces of recording (**a**) and quantitative summary (**b**) showing bath application of NMDA (50 μM) potentiated C-eEPSCs in the presence of apamin (100 nM), an SK channel antagonist ($n = 5$). *$P < 0.05$, **$P < 0.01$ by Brown–Forsythe ANOVA test. **c, d** AHP currents (typical traces in (**c**), amplitude of AHP currents in (**d**) were reduced in SNS-NR1$^{-/-}$ mice, as compared to NR1$^{fl/fl}$ mice ($n = 5$–6). ***$P < 0.001$, ****$P < 0.0001$ by uncorrected Fisher's LSD one-way ANOVA. **e, f** Expression of SK2 subunit were downregulated in nociceptor-specific NR1$^{-/-}$ mice as compared with NR1$^{fl/fl}$ mice ($n = 3$). **$P < 0.01$ by unpaired $t$ test. **g, h** A typical example (**g**) and quantitative summary (**h**) of levels of phosphorylated NR1 and SK2 in L4/L5 DRGs of SNS-NR1$^{-/-}$ mice and NR1$^{fl/fl}$ littermates in the naive state or following CFA injection in the hindpaws ($n = 3$). **$P < 0.01$, ***$P < 0.001$ by Uncorrected Fisher's LSD one-way ANOVA. **i, j** Typical recording traces (**i**) and magnitude quantification (**j**) of AHP currents in small nociceptive DRG neurons derived from naive or CFA-inflamed mice ($n = 5$–7). *$P < 0.05$, **$P < 0.01$ by Brown–Forsythe ANOVA test. **k, l** A typical example (**k**) and quantitative summary (**l**) of levels of PKG-I in DRGs of SNS-NR1$^{-/-}$ mice and NR1$^{fl/fl}$ littermates in the naive state or following CFA injection in the hindpaws ($n = 9$). *$P < 0.05$, ***$P < 0.01$ by Uncorrected Fisher's LSD one-way ANOVA test. **m, n** Following activation of PKG-I via bath application of cGMP analog, 8-pCPT-cGMP (100 μM), NMDA tended to potentiate the C-eEPSCs in naive mice ($n = 5$). Typical traces of recordings are shown in (**m**), and quantitative summary shown in (**n**). ***$P < 0.001$, ****$P < 0.0001$ by Brown–Forsythe ANOVA test. **o, p** Typical traces (**o**) and quantitative summary (**p**) showing NMDA-induced synaptic potentiation in CFA-inflamed state were abolished in SNS-PKG-I$^{-/-}$ mice ($n = 5$). *$P < 0.05$ by Kruskal–Wallis $H$ test. Data are represented as mean ± S.E.M. See Supplemental Table 2 for detailed statistical information.

much attenuated in SNS-NR1$^{-/-}$ mice (Fig. 7a, b). The stimulus-response curve was slightly deviated from the basal curve upon capsaicin challenge (Fig. 7a). Moreover, capsaicin-induced drop in mechanical response threshold was markedly reduced in SNS-NR1$^{-/-}$ mice as compared to NR1$^{fl/fl}$ mice (Fig. 7b). It has been shown that capsaicin-induced secondary mechanical hypersensitivity is independent of ongoing input from the periphery of nociceptors[34]. In further experiments, we addressed the contributions of PreNMDARs from central terminals. We intrathecally (i.t.) administered NMDARs antagonist, AP5 (1 mM, 10 μl) prior to capsaicin in NR1$^{fl/fl}$ mice and found that i.t. AP5 treatment significantly inhibited secondary hypersensitivity (Fig. 7c, d). To further delineate the origin of the central locus of NMDARs function, we undertook the same test in SNS-NR1$^{-/-}$ mice. Interestingly, AP-5 inhibited the mechanical hypersensitivity to a lesser degree in SNS-NR1$^{-/-}$ mice than NR1$^{fl/fl}$ mice (Fig. 7c, d), demonstrating thereby a crucial role of PreNMDARs.

Similarly, muscular injection of acidic saline in the flank muscle leads to secondary mechanical hypersensitivity in the ipsilateral and contralateral hindpaws, which lasts for weeks[59]. This secondary hypersensitivity is assumed to be independent of peripheral inputs, and is thus central in origin[59]. Following two intramuscular injections of acidic saline three days apart, NR1$^{fl/fl}$ mice demonstrated a pronounced leftward and upward shift in the stimulus-response curve to von Frey hairs applied to plantar paw surface, which is evident from 24 h through 3 w of testing period (Fig. 7e). Analysis of mechanical threshold to von Frey hairs revealed a marked drop at different time points after muscular injection (Fig. 7f). These changes come about in the paw ipsilateral to the injected muscle as well as in the contralateral paw. In striking contrast, secondary mechanical hypersensitivity observed in SNS-NR1$^{-/-}$ mice was almost abolished, manifesting as a little deviation in stimulus-response curve and eliminated drop in mechanical threshold (Fig. 7e, f). In conclusion, these analyses support an essential role for Pre-NMDARs in nociceptor terminals in central mechanisms of secondary mechanical hypersensitivity.

Furthermore, we went on to address whether PreNMDARs-dependent BDNF secretion from spinal nociceptor terminals was involved in secondary hypersensitivity. Analysis of mechanical threshold revealed that mice expressing AAV2/8-shRNA BDNF in nociceptors exhibited significantly reduced secondary mechanical hypersensitivity induced by lower thigh injection of capsaicin in comparison with that expressing AAV2/8-conRNA (Fig. 7g). The basal nociceptive pain was not different between two groups. This result suggests presynaptic BDNF released from nociceptors as a key determinant for secondary mechanical hypersensitivity.

**Reinstating NR1 expression in nociceptive DRG neurons of SNS-NR1$^{-/-}$ mice restores mechanical pain hypersensitivity.** Finally, we performed experiments to test whether NMDARs alone are responsible for the deficits in pain hypersensitivity observed in SNS-NR1$^{-/-}$ mice. We constructed Cre-dependent AAV2/8 expressing a Flag-tagged version of the murine NR1 cDNA (rAAV2/8-EF1α-DIO-NR1-3Flag, Supplementary Fig. 13a). Injection in unilateral L3/L4 DRGs in vivo led to a strong expression in nociceptive DRG neurons (Supplementary Fig. 13b). Western blot analysis revealed an overexpression of NR1 in injected DRGs upon injection into SNS-NR1$^{-/-}$ mice (Supplementary Fig. 13c). AAV2/8 virions expressing Flag alone (rAAV2/8-EF1α-DIO-3Flag) served as controls. SNS-Cre mice and SNS-NR1$^{-/-}$ mice expressing Flag-tagged NR1 or Flag alone showed normal basal sensitivity to graded von Frey stimuli (Fig. 8a, b). Upon lower thigh injection of capsaicin, SNS-Cre mice expressing Flag-tagged NR1 showed a slight increase in mechanical hypersensitivity than SNS-Cre mice expressing Flag alone (Fig. 8c). SNS-NR1$^{-/-}$ mice overexpressing Flag in DRG displayed reduced mechanical hypersensitivity as compared to SNS-Cre mice overexpressing Flag (Fig. 8d). More importantly, overexpressing Flag-tagged NR1 in SNS-NR1$^{-/-}$ mice significantly restored mechanical hypersensitivity to capsaicin injection (Fig. 8d). This indicates that PreNMDARs in spinal nociceptor terminals are both necessary and sufficient for inducing centrally maintained hypersensitivity upon peripheral inflammation.

We set out to test whether enhancing the functional activity of presynaptic central terminals of nociceptive DRG neurons in SNS-NR1$^{-/-}$ mice could rescue secondary pain hypersensitivity induced by peripheral inflammation. To enhance the activity of presynaptic central terminals of nociceptors, we used the stimulatory designer receptors exclusively activated by designer drugs (DREADD-hM3Dq (Gq, where hM3Dq indicates Gq-coupled human M3 muscarinic receptor), which couples through a Gq pathway to depolarize neuronal membrane potential upon the application of synthetic ligand clozapine-N-oxide (CNO)[60]. We injected Cre-dependent AAV2/8-CAG-DIO-hM3Dq-mCherry (hM3Dq-mCherry) into unilateral L3/L4 DRGs of SNS-NR1$^{-/-}$ mice to selectively express hM3Dq-mCherry in nociceptive DRG neurons (Fig. 8e). Immunofluorescence staining revealed that mCherry was specifically detected in small- to medium-diameter nociceptive DRG neurons (Supplementary Fig. 13d). Whole-cell recording from mCherry-labeled DRG neurons showed that bath application of CNO induced discharges of spikes, indicating expression of hM3Dq in nociceptive DRG neurons (Supplementary Fig. 13e). We next examined whether enhancing activity of presynaptic central terminals of nociceptive DRG neurons by DREADD-hM3Dq could rescue secondary pain

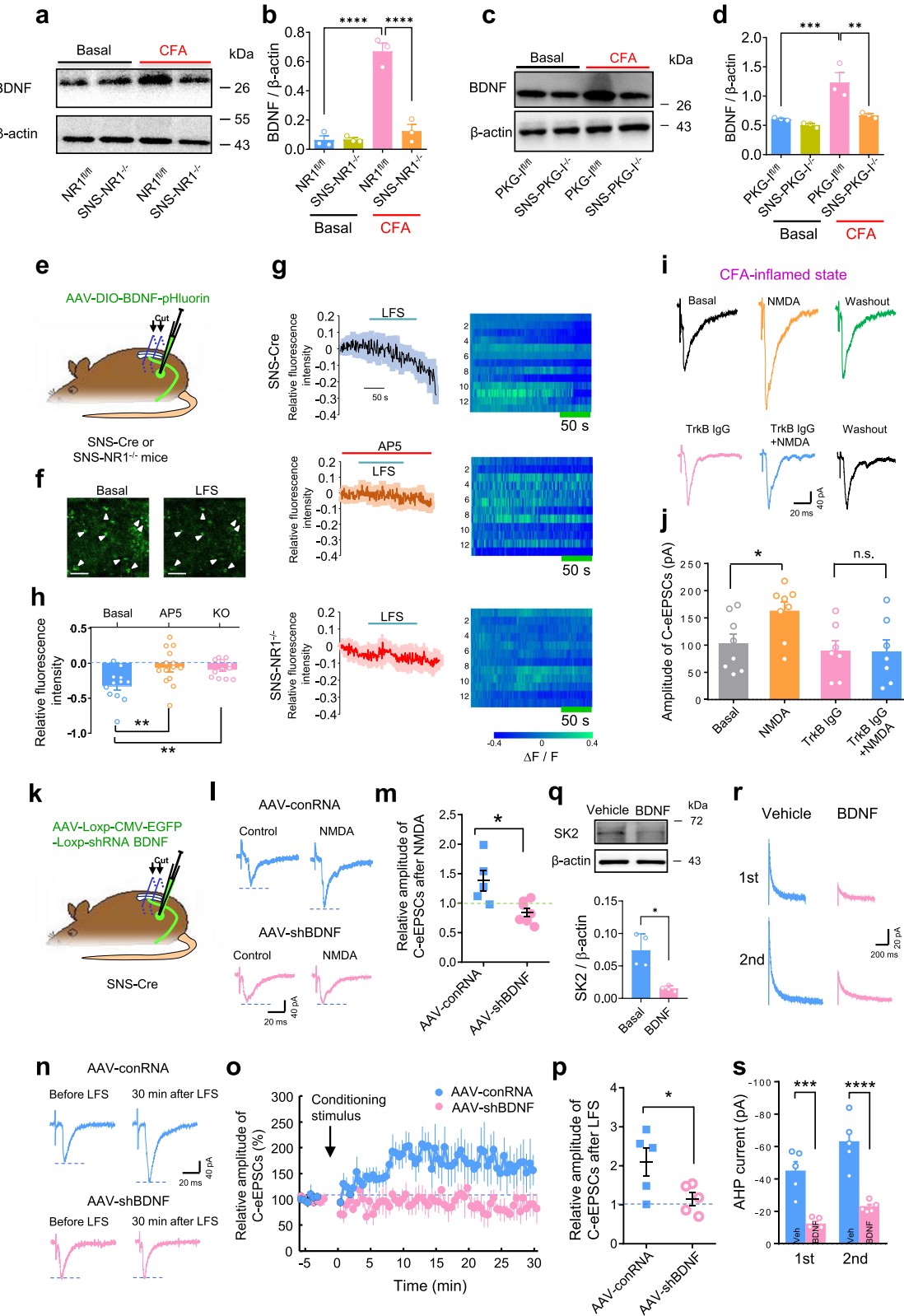

hypersensitivity in SNS-NR1$^{-/-}$ mice. To specifically observe the rescue effect in the pathological state and minimize the suffering of animals due to activation of nociceptors, we adopted intrathecal administration of CNO and lowered CNO at concentration of 10 μM, which induced little nociceptive behavior in the basal state. SNS-NR1$^{-/-}$ mice expressing AAV2/8-hM3Dq-mCherry showed identical normal sensitivity to von

Frey stimuli in response to intrathecal (i.t.) administration of CNO (10 μM) as compared to that expressing AAV2/8-mcherry control virions (Fig. 8f). Upon lower thigh injection of capsaicin, secondary mechanical sensitivity to graded von Frey stimuli was examined at 30, 45 min after i.t. CNO delivery. As compared to control virions, SNS-NR1$^{-/-}$ mice expressing hM3Dq-mCherry showed marked leftward and upward shift of stimulus-response

**Fig. 6 Presynaptic BDNF-pHluorin secretion depends on PreNMDARs localized in spinal terminals of nociceptors. a, b** A typical example (**a**) and quantitative summary (**b**) showing levels of BDNF in L4/L5 DRGs of SNS-NR1$^{-/-}$ mice and NR1$^{fl/fl}$ littermates in the basal or CFA-inflamed state ($n = 3$ mice in each lane). ****$P < 0.0001$ by Uncorrected Fisher's LSD one-way ANOVA. **c, d** Typical examples (**c**) and quantitative summary (**d**) showing levels of BDNF in DRGs of SNS-PKG-I$^{-/-}$ mice and PKG-I$^{fl/fl}$ littermates in the basal or CFA-inflamed state ($n = 3$ mice in each lane). **$P < 0.01$, ***$P < 0.001$ by Uncorrected Fisher's LSD one-way ANOVA. **e** Scheme illustrating the experimental approach for imaging activity-induced changes in BDNF-pHluorin fluorescence from presynaptic nociceptor terminals. **f** Example images showing BDNF-pHluorin expression in the presynaptic nociceptor terminals in the superficial spinal dorsal horn and changes in BDNF-pHluorin (arrowheads) fluorescence before (left: basal) and 150 s after onset of LFS (right: LFS) in SNS-Cre mice. **g, h** Sample traces (**g**) and quantitative summary (**h**) of BDNF-pHluorin fluorescence changes evoked by LFS in SNS-Cre mice in the absence (upper panels) and presence (middle panels) of AP5 as well as in SNS-NR1$^{-/-}$ mice (lower panels) ($n = 12$–14). Shown in right panels are changes of BDNF-pHluorin fluorescence evoked by LFS in heat maps. **$P < 0.01$ by Uncorrected Fisher's LSD one-way ANOVA. **i, j** Example traces of recording (**i**) and quantitative summary (**j**) showing application of BDNF scavenger, TrkB-IgG (2 μg/ml) blocked NMDA-induced synaptic potentiation ($n = 7$–8). *$P < 0.05$ by Uncorrected Fisher's LSD one-way ANOVA. **k** Schematic illustration for specific knockdown of BDNF in nociceptors via injection of Cre-dependent shRNA BDNF into lumbar DRGs of SNS-Cre mice. **l, m** Sample traces (**l**) and quantification analysis (**m**) showing NMDA-induced presynaptic potentiation in the CFA-inflamed state was abolished in spinal slices expressing AAV2/8-shRNA BDNF as compared to controls ($n = 5$–7). *$P < 0.05$ by unpaired $t$ test. **n–p** Example traces (**n**), time course (**o**) and quantitative summary (**p**) of conditioning LFS-induced pre-LTP in spinal-PAG projection neurons expressing AAV2/8-shRNA BDNF and AAV-shRNA control ($n = 5$). *$P < 0.05$ by unpaired $t$ test. **q** Typical examples and quantitative summary of western blot analysis of SK2 on dissected DRGs with vehicle and BDNF treatment (100 ng/ml, 1 h) ($n = 4$). *$P < 0.05$ by unpaired $t$ test with Welch's correction. **r, s** Bath application of BDNF greatly reduced AHP currents recorded in small DRG neurons ($n = 5$). ***$P < 0.001$, ****$P < 0.0001$ by uncorrected Fisher's LSD one-way ANOVA. Data are represented as mean ± S.E.M. See Supplemental Table 2 for detailed statistical information.

curve upon i.t. CNO administration (Fig. 8g–i), indicating restoration of mechanical pain hypersensitivity. Collectively, we thus infer from the above that PreNMDARs in central terminals of nociceptors are both necessary and sufficient for inducing centrally maintained hypersensitivity upon peripheral inflammation.

Last, we sought to address whether NR1 has any potential clinical implication as a key target of PreNMDARs in nociceptors for analgesia. To this end, we performed intraganglionic injection of CGP78608, an NR1-selective antagonist, at 20 min after lower thigh injection of capsaicin. As shown in Fig. 8i, intraganglionic delivery of CGP78608 (2, 10 μM) almost completely reversed the capsaicin-induced secondary mechanical hypersensitivity (Fig. 8i). This strongly suggests the potential clinical implication of NR1 as a key target of PreNMDARs in nociceptors in treating the injury-induced pain sensitization. Since the DRG lies outside of the blood-brain barrier[61], it can be inferred that targeting the NR1-containing PreNMDARs in nociceptors represents a promising strategy for fulfillment of optimal analgesic therapeutics with the least side effects.

## Discussion

The results of the present study lead us to propose a model represented schematically in Fig. 8j. In the basal state, Ca$^{2+}$ influx via PreNMDARs leads to activation of SK channels, which results in presynaptic depression of C-fiber synapses in the spinal cord. In striking contrast, following peripheral inflammation, upregulated PreNMDARs causes enhanced Ca$^{2+}$ influx and activates the downstream signaling cascade PKG-I. PreNMDARs-PKG-I brings about increased production and secretion of BDNF from spinal nociceptor terminals. BDNF secretion in turn depresses SK channels activity and thus results in the conversion of PreNMDARs-mediated synaptic depression to potentiation, which further leads to pain hypersensitivity caused by tissue inflammation. Thus, this study primarily clarifies the functional switch of PreNMDARs in spinal nociceptor terminals in different pain states and explores the mechanistic basis for this state-dependent switch. Moreover, we identify PreNMDARs in spinal nociceptor terminals as key determinants of activity-dependent pain sensitization.

Hebbian-type synaptic plasticity has been previously demonstrated at C-fiber synapses in the spinal dorsal horn by coapplication of presynaptic conditioning stimulation and postsynaptic depolarization[3,4,62]. Besides this, a form of non-Hebbian synaptic

plasticity was reported after a rise in Ca$^{2+}$ due to postsynaptic depolarization in the absence of any presynaptic conditioning stimulation[63]. Both of this Hebbian and non-Hebbian synaptic plasticity requires activation of postsynaptic NMDARs and was proposed to constitute a very fitting correlate of spinal amplification phenomena underlying pain hypersensitivity. However, here we demonstrate a form of non-Hebbian synaptic plasticity at C-fiber synapses with spinal lamina I projection neurons which is induced by only presynaptic conditioning stimulation without postsynaptic depolarization. In contrast to the involvement of postsynaptic NMDARs in the above Hebbian and postsynaptic non-Hebbian synaptic plasticity, this presynaptic plasticity at C-fiber synapses is not dependent on postsynaptic NMDARs, but requires PreNMDARs in spinal terminals of nociceptors. Minimal stimulation and PPR analysis suggest that presynaptic plasticity at C-fiber synapses involves an increase of transmitter release via a presynaptic mechanism. This is consistent with previous studies showing that peripheral tissue inflammation greatly enhances excitatory synaptic transmission by increasing presynaptic transmitter release[12]. Similar presynaptic plasticity has been reported in several brain regions, i.e. hippocampus, amygdala and ACC[8–10,64]. Unlike the requirement of PreNMDARs in spinal presynaptic plasticity, the trigger for cortical presynaptic plasticity involves presynaptic kainite receptors and downstream adenylyl cyclase-PKA signaling cascades[8,64].

The physiological significance of PreNMDARs has been addressed in neurotransmission and plasticity[15]. PreNMDARs are reported to regulate presynaptic neurotransmitter release[19,65]. Evoked and spontaneous release are modulated by PreNMDARs via an independent and non-overlapping mechanism[14]. Besides the influence of PreNMDARs on transmitter release, there is increasing evidence for the involvement of PreNMDARs in activity-dependent synaptic plasticity. For example, PreNMDARs are required for the induction of spike timing-dependent LTD and a form of presynaptic spike pattern-dependent LTD in sensory cortices[25,66]. PreNMDARs are also required for hetero-synaptic and homosynaptic LTP at cortical projections to the amygdala and striatum[22,23]. In contrast to the increasingly studied PreNMDARs in cortical plasticity, the involvement of Pre-NMDARs in presynaptic plasticity at C-fiber synapses in the spinal cord remains to be fully explored. Here we generated nociceptor-specific NR1 knockout mice and demonstrated that LFS-induced pre-LTP at C-fiber synapses requires activation of PreNMDARs via a presynaptic mechanism. The requirement of

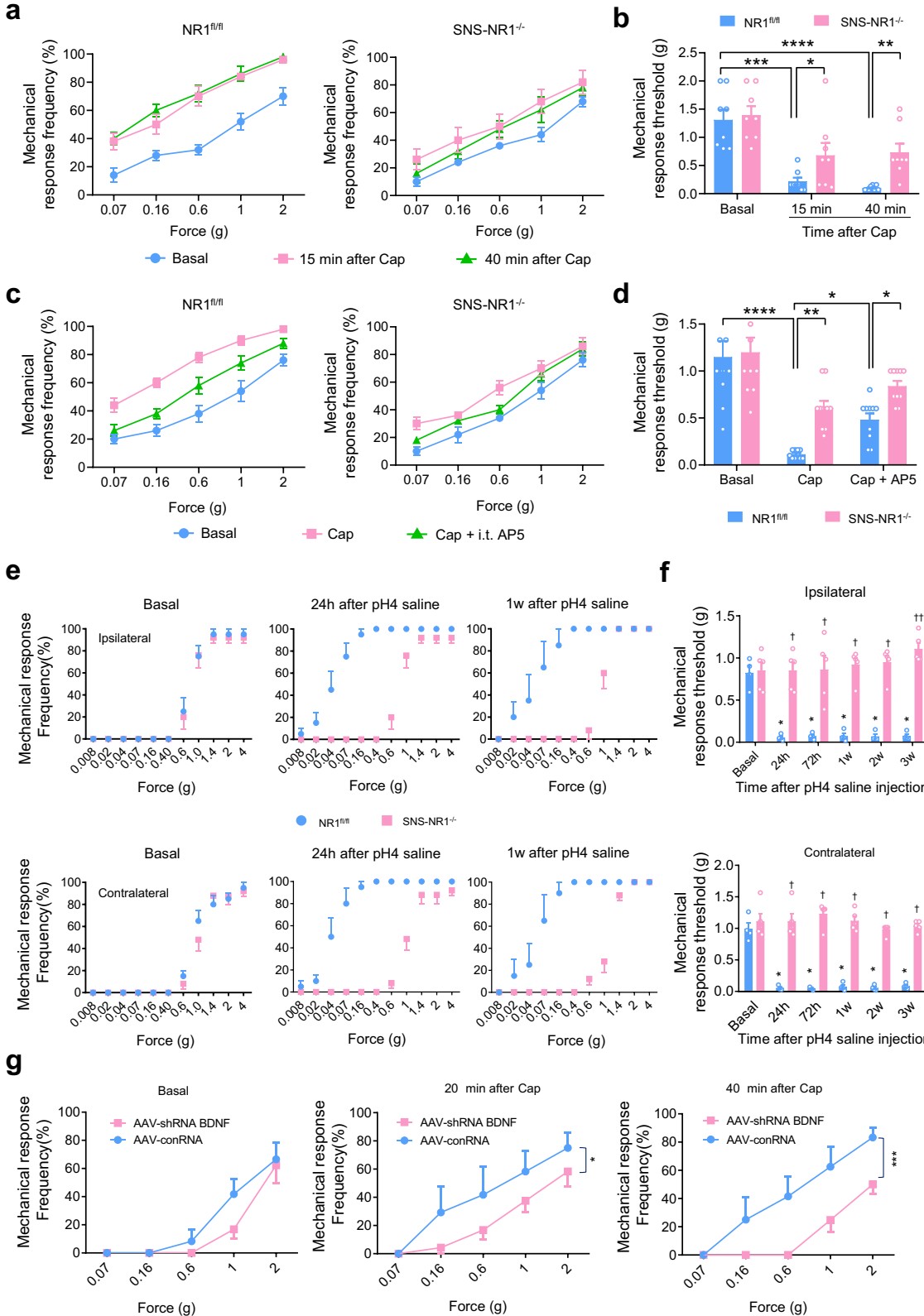

PreNMDARs in spinal pre-LTP is in contrast with cingulate pre-LTP which does not require PreNMDARs but involves GluK1-containing kainite receptors and downstream adenylyl cyclase-PKA signaling cascades[8].

To further determine whether PreNMDARs are sufficient for the induction of presynaptic plasticity at C-fiber synapses, exogenously applied NMDA or endogenously blocked PreNMDARs

by AP5 was conducted to see whether it alone could induce presynaptic plasticity. Interestingly, we found that PreNMDARs in spinal nociceptor terminals modulate presynaptic transmission at C-fiber synapses in a nociceptive tone-dependent manner, i.e. depressing presynaptic transmission under basal state but potentiating it under inflammatory state. Although PreNMDARs in primary afferents have been reported on modulation of

**Fig. 7 PreNMDARs and its downstream target BDNF in nociceptors are required for the behavioral manifestation of nociceptive hypersensitivity. a, b**
Secondary hyperalgesia and allodynia to von Frey hairs applied to the plantar surface of the paw, which lies outside of the flare evoked by capsaicin injection to the lower thigh, developed to a much lower degree in SNS-NR1$^{-/-}$ mice than NR1$^{fl/fl}$ mice ($n = 8$). Shown are stimulus-response curves (**a**) and the summary of response thresholds to von Frey hair to the hindpaw prior to and at 15, 40 min following capsaicin injection (**b**). *$P < 0.05$, **$P < 0.01$, ***$P < 0.001$, ****$P < 0.0001$ by Kruskal–Wallis $H$ test. **c, d** Intrathecal injection of NMDARs antagonist, AP-5 (1 mM) inhibited capsaicin-induced secondary mechanical hypersensitivity in NR1$^{fl/fl}$ mice (left panel), but not in SNS-NR1$^{-/-}$ mice (right panel). Quantitative summary is shown in (**d**) ($n = 10$). *$P < 0.05$, **$P < 0.01$, ****$P < 0.0001$ by Kruskal–Wallis $H$ test. **e** Following injection of acidic saline in the flank muscle, increase of response frequency to von Frey hairs is much greater in NR1$^{fl/fl}$ mice than SNS-NR1$^{-/-}$ mice in both ipsilateral (upper panels) and contralateral (lower panels) hindpaws ($n = 4$ for NR1$^{fl/fl}$, $n = 5$ for SNS-NR$^{-/-}$). **f** Summary of mechanical response thresholds to von Frey hairs application to the ipsilateral (upper panel) and contralateral (lower panel) paws, prior to and at different time points following muscular injection in NR1$^{fl/fl}$ and SNS-NR1$^{-/-}$ mice. *$P < 0.05$ at all time points for NR1$^{fl/fl}$ *vs* basal, †$P < 0.05$ at all time points for NR1$^{fl/fl}$ *vs* SNS-NR1$^{-/-}$ by Friedman $M$ test. **g** Specific knockdown of BDNF in nociceptors reduced capsaicin-induced secondary mechanical hypersensitivity. Shown are stimulus-response curves to von Frey hairs applied to hindpaws following capsaicin injection into the flank in SNS-Cre mice expressing shRNA BDNF and shRNA control in the L3/L4 DRGs ($n = 4$). *$P < 0.05$, ***$P < 0.001$ by Friedman $M$ test. Data are represented as mean ± S.E.M. See Supplemental Table 2 for detailed statistical information. Cap, capsaicin.

transmitter release in previous studies, these reports are contradictory and conflicting, i.e. facilitating[29–32] or depressing[28] glutamate release or even no effect[29,67]. None of studies linked these two disparate findings and clarified that state-dependent characteristics of PreNMDARs mediate both mutually opposite effects. Thus, to the best of our knowledge, our current study demonstrates a critical finding which links mutually contradictory findings from previous studies and enables a mechanistic understanding of a fundamentally important switch in NMDARs function at presynaptic nociceptor terminals. This idea is reinforced by another report observing functional switch of PreNMDARs in the visual cortical synapses during development due to NR3A to NR2B subunit composition switch[68,69]. The present results inferred us that PreNMDARs in spinal nociceptor terminals might serve as a key regulator for pain signals inflow in different conditions. For example, in physiological state, activation of PreNMDARs depresses primary afferent synaptic transmission and thus prevents the transmission of excessive pain signals from the periphery to CNS, forming a negative feedback for peripheral pain signal transmission. However, in states of tonic pain resulting from peripheral inflammation, upregulation of PreNMDARs enhances Ca$^{2+}$ influx and activates downstream signaling cascades, which paradoxically leads to facilitation of presynaptic transmission and exaggerates pain signals inflow into CNS and thus resulting in central sensitization and pain hypersensitivity. Then how does PreNMDARs in spinal nociceptor terminals achieve state-dependent modulation of synaptic plasticity?

Another most important and intriguing finding of the present study is that we revealed a mechanism underlying this state-dependent transition. A close interaction between NMDARs and SK channels in the postsynaptic neuron are demonstrated in determining intrinsic excitability and NMDARs-dependent synaptic plasticity in several brain regions, i.e. hippocampus and lateral amygdala[47,48,70,71]. However, whether presynaptic SK channels interacts with PreNMDARs and modulates presynaptic plasticity remains elusive. SK channels, especially SK2 subunits are found in nociceptive primary sensory neurons and acts with Ca$^{2+}$ influx via NMDARs to regulate DRG excitability[49,72]. This indicates a crucial cross-talk of NMDARs with SK2 channels in nociceptors. However, whether SK2 channels are expressed in presynaptic nociceptor terminals and contribute to PreNMDARs-mediated synaptic modulation at C-fiber synapses is not fully understood. Our ultrastructural data showed an expression of SK2 channels and NR1 subunit of NMDARs in the presynaptic terminals of CGRP-positive nociceptive DRG neurons. Blockade of SK channels with apamin abolished NMDA-induced synaptic depression in the basal state, and even convert depression to potentiation via a presynaptic action. These results inferred that

presynaptic SK channels in spinal nociceptor terminals might act as a key regulator in determining the direction of PreNMDARs-mediated synaptic plasticity. In the basal state, activation of SK channels was involved in the spinal synaptic depression by PreNMDARs, which could serve as a negative feedback and filter to counteract more pain signals flowing into CNS. However, the observed dramatic downregulation of SK channels in the pathological state causes facilitation of PreNMDARs-mediated synaptic transmission, leading to exaggerated pain signals inflow, which in turn results in pain hypersensitivity. Similar to our observation in presynaptic SK channels, blockade of dendritic SK channels is reported to convert NMDAR-dependent LTP to strong LTD in electrosensory lateral line lobe pyramidal cells[73]. Then what could cause the downregulation of SK channels activity in pathological states?

Previous studies showed that BDNF play a crucial role in the induction of spinal synaptic potentiation[56,74,75]. However, BDNF can be derived from pre- and post-synaptically as well as glial cells in the pathological states[76–78], the precise identity of critical signaling molecules, their mechanisms of action and their locus in the spinal circuitry have remained elusive. Although much evidence has accumulated that BDNF contributes to spinal synaptic transmission and pain hypersensitivity via potentiation of downstream postsynaptic NMDARs in different pathological states[76–81], much less is known whether and how PreNMDARs act as an upstream to shape presynaptic plasticity at C-fiber synapses by triggering BDNF cascade. Previous studies described an essential role of PreNMDARs-dependent secretion of BDNF in corticostriatal LTP[23]. Here, we demonstrated a PreNMDARs-dependent BDNF secretion by imaging of spinal nociceptor terminals expressing BDNF-pH in spinal slices, which further leads to SK channels downregulation and induces spinal presynaptic potentiation. This thereby answered a long-standing question regarding the source of BDNF in spinal plasticity. Mechanistically, PreNMDARs-dependent BDNF secretion may come about via the involvement of PKG-I in nociceptors terminals. Numerous studies have documented the key significance of the postsynaptic NMDA receptor-NOS-cGMP-PKG-I pathway in the synaptic plasticity and chronic pain states[52,53]. However, whether the presynaptic NMDARs-NOS-cGMP-PKG-I pathway in spinal nociceptor terminals modulates spinal synaptic potentiation is still unclear. Our observations that repetitive activation of nociceptors in vivo leads to PKG-I activation in NMDARs dependent manner in the DRG and dorsal roots and activation of PKG-I induces NMDA-induced synaptic potentiation to implicate PKG-I as a key downstream signaling target of NMDARs in the DRG and dorsal roots in the pathological state. Furthermore, we provide direct evidence that axonal PreNMDARs in spinal nociceptor terminals are responsible for triggering presynaptic

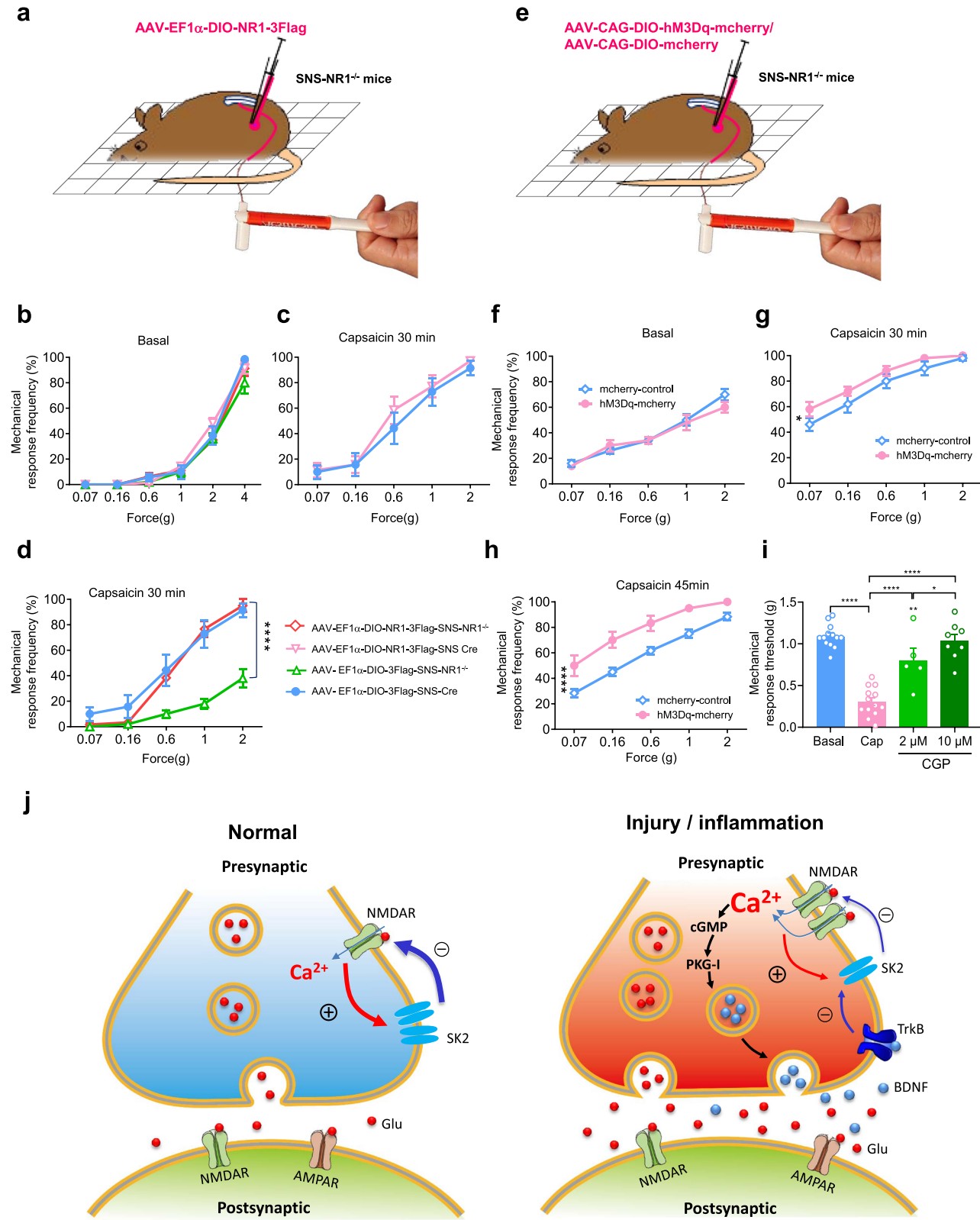

BDNF secretion by mediating prolonged Ca²⁺ elevation and PKG-I activation.

Then what is the function of PreNMDARs-PKG-I-dependent BDNF vesicular release specifically from spinal nociceptor terminals? Although converging evidence has described a clear action of BDNF in spinal synaptic potentiation[56,74,75], pharmacological approaches with BDNF scavenger TrkB-IgG in these studies

make it hard to dissect the contribution of BDNF secretion from spinal nociceptor terminals. Our observations on nociceptor-specific BDNF knockdown mice revealed that presynaptic BDNF release in spinal nociceptor terminals facilitates PreNMDARs-mediated synaptic potentiation in the inflamed states, which mimics the effect of blockade of SK2 channels on PreNMDARs-mediated synaptic potentiation. Tissue inflammation-induced

**Fig. 8 Rescue of defects in pain hypersensitivity in SNS-NR1$^{-/-}$ mice by viral expression of DIO-Flag-tagged NR1 or by chemogenetically activation of nociceptive DRG neurons via viral injection of DIO-hM3Dq specifically in DRGs. a** Schematic illustration of expression of DIO-Flag-tagged NR1 in the L3/L4 DRGs upon delivery via adeno-associated virions serotype2/8 (AAV2/8). **b** Basal mechanical sensitivity to von Frey hairs application was unaltered upon overexpression of DIO-Flag-tagged NR1 or DIO-Flag alone in DRG neurons of SNS-Cre mice and SNS-NR1$^{-/-}$ mice. **c** SNS-Cre mice expressing DIO-Flag-tagged NR1 show a slight increase in magnitude of capsaicin-induced mechanical hypersensitivity than SNS-Cre mice expressing DIO-Flag alone. **d** SNS-NR1$^{-/-}$ mice overexpressing DIO-Flag in DRGs showed markedly reduced mechanical hypersensitivity with capsaicin than SNS-Cre mice overexpressing DIO-Flag. However, overexpression of DIO-Flag-tagged NR1 fully restored mechanical hypersensitivity in SNS-NR1$^{-/-}$ mice. $n = 5$–7, Friedman $M$ test, ****$P < 0.0001$ indicates significant differences in the AAV2/8-DIO-Flag-SNS-NR1$^{-/-}$ mice as compared to the AAV2/8-DIO-Flag-NR1-SNS-NR1$^{-/-}$ mice. **e** Schematic demonstration of expression of AAV2/8-CAG-DIO-hM3Dq-mcherry and AAV2/8-mcherry control in the L3/L4 DRGs of SNS-NR1$^{-/-}$ mice. **f** Intrathecal injection of CNO did not change the basal mechanical sensitivity to von Frey hair stimuli in mice subjected to injection of AAV2/8 expressing DIO-hM3Dq-mcherry as compared to control AAV. **g, h** Increase in response frequency to von Frey hairs at 30, 45 min following intrathecal injection of CNO is significantly greater in SNS-NR1$^{-/-}$ mice expressing DIO-hM3Dq-mcherry than DIO-mcherry alone. $n = 5$-6, Friedman $M$ test, *$P < 0.05$, ****$P < 0.0001$ indicates significant differences between two groups at 30, 45 min, respectively. **i** Intraganglionic administration of CGP78608 (2 and 10 μM) almost completely reversed the capsaicin-induced secondary hyperalgesia ($n = 5$–13). *$P < 0.05$, ****$P < 0.0001$ by Uncorrected Fisher's LSD one-way ANOVA. **j** A schematic model proposing a nociceptive tone-dependent role of PreNMDARs expressed in spinal terminals of nociceptors in spinal presynaptic plasticity in different conditions (see text for details). Data are represented as mean ± S.E.M. See Supplemental Table 2 for detailed statistical information.

profound upregulation of BDNF but downregulation of SK2 channels led us to ask whether depression of SK2 channels by BDNF serve to mediate the conversion of PreNMDARs-mediated synaptic depression to potentiation in the inflamed states. In support of our hypothesis, immunoblot analysis and patch-clamp recordings collectively showed that exposure of DRGs to BDNF greatly reduces the expression and function of SK2 channels. Similar to our observation, suppression of BK channels (big-conductance Ca$^{2+}$-activated K$^{+}$ channels) by BDNF was reported in the DRG after nerve injury[82]. BDNF-facilitated hippocampal LTP was reported to be mediated by suppression of SK2 channels[83]. These findings point to the conclusion that increase of PreNMDARs-PKG-I-dependent BDNF secretion from spinal nociceptor terminals induced by tissue inflammation triggers downregulation of presynaptic SK2 channels, which promotes conversion of PreNMDARs-mediated synaptic depression to potentiation. Taken together with the fact that BDNF potentiates NMDARs in peripheral sensory neurons[84], our results uncover a feed-forward regulatory network driven by PreNMDARs in spinal nociceptor terminals that may facilitate synaptic transmission and hence promote the transition from acute pain to chronic pain. Apart from this, other mechanisms relating PreNMDARs with calcium channel α2δ-1 subunits in mediating synaptic transmission and development of pain hypersensitivity can not be excluded[85].

The possibility of functionally linking synaptic changes described here to changes in nociceptive behavior simultaneously represents a good opportunity and a major challenge. To test this relationship, we adopted pain models in which pain hypersensitivity is triggered by peripheral nociceptors, but maintained via central mechanisms that are independent upon peripheral inputs, i.e. chronic muscle pain model[59] and lower thigh capsaicin model[34]. We observed dramatic defects in central hypersensitivity in SNS-NR1$^{-/-}$ mice, as compared with NR1$^{fl/fl}$ mice in both models. Furthermore, intrathecal delivery of AP5 greatly attenuated secondary mechanical hypersensitivity in NR1$^{fl/fl}$ mice, but to a much lesser degree in SNS-NR1$^{-/-}$ mice. Thus, our current study reveals an essential role for PreNMDARs in spinal nociceptor terminals in central mechanisms of secondary hyperalgesia and allodynia. We further determined whether PreNMDARs promotes pain sensitization via presynaptic BDNF secretion in spinal nociceptor terminals. Specific loss of BDNF in nociceptors was found to cause a prominent reduction in capsaicin-induced secondary hypersensitivity, indicating a crucial role of presynaptic BDNF in secondary pain hypersensitivity. Although multiple studies described the pro-nociceptive role of

BDNF in pain processing[78,86–88], the exact sources of BDNF responsible for nociceptive hypersensitivity have not been clarified in those studies, except for some studies reporting the origin of microglia for BDNF in mediating neuropathic pain[77,89]. Recent evidence on microglia RNAseq in the spinal dorsal horn, however, suggest a lack of expression of the *bdnf* gene in microglia, while primary afferents do express BDNF[90]. Finally, either reinstating NR1 expression in the nociceptive DRG neurons or chemogenetically activating nociceptive DRG neurons in adult SNS-NR1$^{-/-}$ mice fully restored capsaicin-evoked mechanical hypersensitivity. Furthermore, intraganglionic injection of NR1-selective antagonist relieved capsaicin-induced secondary pain hypersensitivity. Taken together, these data indicate that PreNMDARs and subsequent BDNF secretion from presynaptic nociceptor terminals are functional determinants as well as necessary and sufficient for presynaptic plasticity and centrally mediated hypersensitivity upon peripheral inflammation.

In summary, this study sheds light on the characteristics of nociceptive tone-dependent-modulation of spinal presynaptic plasticity by PreNMDARs in spinal nociceptor terminals and revealed a mechanism underlying this state-dependent transition. Furthermore, it suggests that PreNMDARs-mediated presynaptic potentiation is functionally involved in activity-dependent centrally mediated nociceptive hypersensitivity caused by peripheral inflammation and PreNMDARs in spinal nociceptor terminals might be a promising target in treating chronic pain without incurring side effects given by actions on the brain.

## Methods

**Genetically modified mice.** Homozygous mice carrying the flox allele of the mouse *nr1* gene, which encodes NR1 subunit of NMDARs (NR1$^{fl/fl}$), have been described previously in detail. NR1$^{fl/fl}$ mice were crossed with SNS-Cre mice[33] to obtain NR1$^{fl/fl}$; SNS-Cre$^{+}$ mice (referred to as SNS-NR1$^{-/-}$ mice in this manuscript) and NR1$^{fl/fl}$ mice (control littermates). Mice were crossed into the C57BL6 background for more than 8 generations. SNS-PKG-I$^{-/-}$ mice were generated in a similar way to SNS-NR1$^{-/-}$ mice. Only littermates were used in all experiments to control for background effects. All experimental protocols were approved by Institutional Animal Care and Use Committee of the Fourth Military Medical University (FMMU). Mice were housed up to 5 per cage and maintained on a 12 h light/dark cycle with ad libitum access to food and water under the ambient temperature at 22–26 °C and humidity at 40-70%. All testing was done in a double-blinded manner. The experimenter was blind to the treatment and/or genotype of mice throughout. See the details for mouse lines in Supplementary Table 1.

**DiI labeling of spinal projection neurons in vivo and spinal slice patch-clamp recordings.** Mice were placed in a stereotaxic apparatus and received a single injection of 100 nl of 2.5% DiI into the right ventrolateral PAG according to coordinates derived from the atlas of Paxinos and Watson (AP: 0.1 mm; depth: 5.1 mm; DL: 0.4 mm measured from the lambda). After a 2- to 3-day survival

period, transverse 350–450 μm thick spinal cord slices with dorsal roots attached were obtained. The slices were stored in an incubation solution at room temperature (in mM: NaCl, 95; KCl, 1.8; KH$_2$PO$_4$, 1.2; CaCl$_2$, 0.5; MgSO$_4$, 7; NaHCO$_3$, 26; glucose, 15; sucrose, 50; oxygenated with 95% O$_2$, 5% CO$_2$; pH 7.4). A slice was then transferred into a recording chamber and superfused with oxygenated recording solution at 3 ml min$^{-1}$ at room temperature. The recording solution was identical to the incubation solution except for (in mM): NaCl 127, CaCl$_2$ 2.4, MgSO$_4$ 1.3 and sucrose 0. All injection sites were confirmed histologically. To detect lamina I projection neurons which were labeled by DiI from the injection sites of PAG (as described above), slices were illuminated with a monochromator, and visualized with an upright fluorescence Olympus BX51WI microscope (Olympus, Japan), equipped with Dodt-infrared optics using a 40X, 0.80 NA water-immersion objective and a cooled CCD camera (TILL Photonics, Gräfelfing, Germany).

Standard whole-cell patch-clamp recordings were performed with glass pipettes having a resistance of 4-6 MΩ in lamina I of spinal dorsal horn. The pipette solution consisted of (in mM): K-gluconate, 135; KCl, 5; CaCl$_2$, 0.5; MgCl$_2$, 2; EGTA, 5; HEPES, 5 and Mg-ATP, 5, pH 7.4 with KOH, measured osmolarity 300 mOsm. QX-314 (5 mM) was added to the pipette solution to prevent discharge of action potentials (Aps). The electrophysiological properties of the recorded neurons were acquired with an Axon700B amplifier (Molecualr Devices Corporation) and Clampex 9.2 software. Signals were low-pass filtered at 5 kHz, sampled at 10 kHz and analyzed offline with Clampfit 10.6 software. The membrane potential was held at -70 mV. Evoked excitatory postsynaptic currents (eEPSCs) from labeled neurons in lamina I was recorded by stimulating dorsal root with a suction electrode in the presence of inhibitory synaptic transmission antagonists, gabazine (10 μM) and strychnine (1 μM). Test pulses of 0.1 ms with intensity of 3 mA were given at 30 s intervals. Aδ-fiber or C-fiber evoked EPSCs (eEPSCs) were distinguished on the basis of the conduction velocity (CV) of afferent fibers (Aδ: 2–13 m/s; C: < 0.8 m/s; calculated from the latency of EPSC from a stimulus artifact and the length of dorsal root), as described previously[34,91]. Aδ-fiber or C-fiber responses, respectively, were considered as monosynaptic in origin when the latency remained constant and there was no failure during stimulation at 20 Hz for 1 s, or when failures did not occur during repetitive stimulation at 2 Hz for 10 s[34,91]. Spinal pre-LTP was induced by low-frequency stimulation (LFS, 2 Hz for 2 min) delivering to dorsal root at a holding potential of −70 mV with same intensity as test stimulation. Synaptic strength was quantified by the peak amplitudes of EPSCs. The mean amplitude of 10 EPSCs evoked by test stimuli prior to conditioning stimulation served as a control. Significant changes from control were assessed by measuring the peak amplitudes of five consecutive EPSCs every 5 min after conditioning stimulation. In a subset of experiments, MK-801 (1 mM) or BAPTA (20 mM) was added in the patch pipette internal solution to block postsynaptic NMDARs or Ca$^{2+}$-activated small-conductance K$^+$ channels[4,29]. In addition, in a subset of experiments, miniature EPSCs (mEPSCs) were recorded in the presence of TTX (0.5 μM), gabazine (50 μM) and strychnine (10 μM) at a holding potential of -70 mV, as described routinely[28–32]. However, considering the presence of presynaptic Nav1.8 channels, TTX-resistant sodium channels in nociceptors, mEPSCs were further recorded and analyzed in the presence of TTX (0.5 μM), gabazine (50 μM), strychnine (10 μM) and A-803467 (0.5 μM), an Nav1.8 selective blocker. See the list of reagents used in Supplementary Table 1.

**Intact whole-mount DRG preparations and whole-cell patch-clamp recording.** As described previously[11], L4/L5 DRGs were carefully removed and placed into artificial cerebrospinal fluid (ACSF). After removing the connective tissue, the ganglia were digested with a mixture of 0.4 mg/ml trypsin and 1.0 mg/ml type-A collagenase (Sigma) for 40 min at 37 °C, in the presence of an NMDAR antagonist, ketamine (200 μM) to reduce the desensitization of NMDARs, as described previously[44]. The intact ganglia were then incubated in ACSF oxygenated with 95% O$_2$ and 5% CO$_2$ at 28 °C for at least 1 h before transferring them to the recording chamber. The ACSF contained (in mM): 125 NaCl, 2.5 KCl, 1.2 NaH$_2$PO$_4$, 1.0 MgCl$_2$, 2.0 CaCl$_2$, 26 NaHCO$_3$ and 10 glucose. DRG neurons were visualized with a 40X water-immersion objective using a microscope (BX51WI; Olympus, Tokyo, Japan) equipped with infrared differential interference contrast (IR-DIC). Whole-cell current and voltage recordings were acquired using an Axopatch 700B amplifier and data analyzed using pCLAMP10.0 software. NMDA-induced currents were recorded at a holding potential of −40 mV under voltage clamp. Apamin-sensitve outward tail curents (afterhyperpolarization (AHP) currents) were recorded as described previously[49]. Briefly, SK currents were recorded under voltage clamp with a 400 ms depolarizing pulse (from -60 to 0 mV, holding at −60 mV) followed by a second 400 ms depolarizing pulse (from −80 to 0 mV, holding at -60 mV). The interval between two pulses was 900 ms. The pipette solution for DRG recording contained the following (in mM): K-gluconate, 126; NaCl, 10; MgCl$_2$, 1; EGTA, 10; NaATP, 2 and MgGTP, 0.1, adjusted to pH 7.4 with KOH and osmolarity 295–300 mOsm.

For live identification of IB4-labeled non-peptidergic nociceptive neurons, DRGs were incubated with a vital marker, Alexa 488-conjugated IB4 (10 μg/ml; Molecular probes) for 10 min as described by Han et al.[92] and washed in ACSF solution before whole-cell patch-clamp recording. For confirmation of functional NMDARs deletion in peptidergic nociceptive DRG neurons, IB4-negative small

DRG neurons were patched in the presence of neurobiotin (1%, Vectorlabs) in the intrapipette solution. After recording, DRG containing neurobiotin-filled cells was fixed with 4% paraformaldehyde for 2 h. Cryosections of DRG (16 μm) was immunostained with Alexa Fluor® 488-conjugated streptavidin (Invitrogen; dilution 1:1000) and double-stained with anti-CGRP antibody (Abcam, 1:500) using standard protocol as described below. See the list of reagents used in Supplementary Table 1.

**Immunofluorescence labeling.** Mice were anesthetized with pentobarbital sodium and transcardially perfused with saline followed by 4% paraformaldehyde. Vibratome sections (30 μm) of the spinal cord and brain or Cyrostat-sections (16 μm) of the L4/L5 DRG were immunostained using standard protocol with goat anti-NR1 antibody (Santa Cruz, 1:100), biotinylated isolectin B4 antibody (vector laboratories, 1:200), rabbit anti-CGRP antibody (Calbiochem, 1:500), goat anti-CGRP antibody (Abcam, 1:500), mouse anti-NF200 antibody (Sigma-Aldrich, 1:200), rabbit anti-PSD-95 antibody (Abcam, 1:500), rabbit anti-synaptophysin antibody (Abcam, 1:500), goat anti-GFP antibody (Rockland, 1:500), rabbit anti-RFP antibody (Abcam, 1:500), mouse anti-Flag (Abbkine, 1:800). The number of immunoreactive neurons per DRG section was counted and numbers were averaged over 10 sections per mouse and 4 mice per treatment group. All images were captured with Olympus Fluoview version 3.1 software in an Olympus confocal microscope. See also the list of antibodies used in Supplementary Table 1.

**Biochemical analysis of DRG and dorsal root lysates.** DRGs and dorsal roots were collected and homogenized, respectively in ice-cold lysis buffer containing (in mM): 50 Tris-HCl, pH 7.4, 150 NaCl, 5 EDTA, 1% Triton X-100, 0.5% Sodium deoxycholate, 0.1% SDS and standard protease inhibitors. Insoluble material was removed by centrifugation (16,060 $g$ × 10 min) and the supernatant was collected. Protein concentration for each sample was determined by the bicinchoninic acid (BCA) method using the MICRO BCA protein assay kit (Pierce). The membrane blots were blocked with 10% non-fat dry milk for 12 h and incubated with primary antibodies: rabbit anti-NR1 (Abcam, 1:1000), rabbit anti-phosphoNR1 (Santa, 1:500), rabbit anti-SK2 (Alomone, 1:200), rabbit anti-PKG-I (a kind gift from Robert Feil, Universität Tübingen, Tübingen, Germany, 1:4000), rabbit anti-BDNF (Novus, 1:500) and rabbit anti-β actin (Sigma-Aldrich, 1:5000) overnight at 4 °C. The membranes were then incubated with secondary HRP labeled anti-rabbit (Sigma Aldrich) antibodies for 2 h. Blots were then developed using chemiluminescent HRP substrate (Millipore). Full gels/blots were run in all the experiments with dorsal roots and some DRGs blotting (Fig. 5e, g; Fig. 6a, c, q; Supplementary Fig. 1e–g; Supplementary Fig. 13c), uncropped blots are not available for these experiments because of the following reasons. These parts of experiments were done in earlier years and uncropped blots were not required at that time. Thus, for the sake of saving the antibodies, cropped blots were run for these experiments. The scanned images were quantified using ImageJ software. See also the list of antibodies used in Supplementary Table 1.

**Injection of AAV virions in DRG in vivo.** Virus injection in DRG was performed as described previously[34,93,94]. Briefly, 4–6-week old mice were anesthetized with isoflurane and two lumbar DRGs exposed by removal of the lateral processes of the vertebrae. The epineurium over the DRG was opened, and the glass pipette with fine tip was inserted into the ganglion, to a depth of 100 μm from the surface of the exposed ganglion. After waiting 2 min to allow sealing of the tissue around the pipette tip, 1.0 μl of AAV2/8 virions expressing DIO-GCaMP6s, DIO-BDNF-pHluorin, DIO-hM3Dq, DIO-NR1, BDNF-shRNA-loxp was injected into DRGs of SNS-Cre and SNS-NR1$^{-/-}$ mice at a rate of 0.1 μl/min using microprocessor-controlled minipump (RWD). The pipette was removed after a further delay of 5 min. The muscles overlying the spinal cord were carefully sutured and mice allowed to recover at 37 °C warming blanket. Mice were allowed to recover for 4 weeks before commencing behavioral and calcium imaging as well as patch-clamp analysis. At the end of the experiment, mice were perfused as described above and expression of virus was confirmed via fluorescence analysis or combined with western blotting analysis for some viruses. See the list of viruses used in Supplementary Table 1.

**Calcium imaging with GCaMP6s in presynaptic terminals of nociceptors.** AAV2/8-CAG-DIO-GCaMP6s was injected in L4/L5 DRGs of SNS-Cre and SNS-NR1$^{-/-}$ mice as described above. Four weeks after virus expression, transverse 350-450 μm thick spinal cord slices with dorsal roots attached were obtained. Spinal cord slice was perfused with oxygenated (95% O$_2$, 5% CO$_2$) recording solution as described above in whole-cell patch-clamp recording in spinal cord slice. GCaMP6s signal in the presynaptic terminal of nociceptors in the superficial spinal dorsal horn was visualized using an upright super-resolution Olympus FV1200 confocal microscope (Olympus, Japan). Images were acquired at 1 Hz with Olympus Fluoview version 3.1 software. Fluorescence intensity of each puncta was measured. Bath application of NMDA (50 μM), glutamate (1 mM), high KCl (30 mM), capsaicin (1 μM) was used to activate the Ca$^{2+}$ signal in the presynaptic terminals of nociceptors. In parallel, conditioning low-frequency stimulation (LFS) (2 Hz, 2 min, 3 mA) was delivered to the dorsal root to investigate Ca$^{2+}$ signals

during LFS-induced spinal LTP. See the list of reagents and viruses used in Supplementary Table 1.

### Analysis of BDNF secretion in presynaptic terminals of nociceptors with BDNF-pHluorin

AAV2/8-DIO-BDNF-pHluorin was constructed by inserting BDNF-pHluorin fragment into AAV2/8-DIO vector, as shown in detail in Supplementary Fig. 10e. Briefly, for making a custom AAV2/8-DIO vector, double floxed (floxed with two loxPs and lox2272) gene insertion cassette was inserted into AAV2/8-EF1a vector through AscI and NheI restriction enzyme sites. Then, PCR-amplified BDNF-pHluorin from puc-BDNF-pHluorin was inserted into AAV2/8-DIO vector through NheI and AscI restriction enzyme site to construct AAV2/8-DIO-BDNF pHluroin. Sequencing and restriction enzyme reactions were performed to verify the plasmid. Packaging (serotype 8) and purification of AAV2/8-DIO-BDNFpHluorin were carried out by BrainVTA as described in key resources table. The titre for each virion is above $2.00E + 12$ vg/mL and with high quality. AAV2/8-DIO-BDNF-pHluorin was injected into the DRG of SNS-Cre mice and SNS-NR1$^{-/-}$ mice or SNS-PKG-I$^{-/-}$ mice (1 μl per DRG) as described above. Mice were tested in various tests four weeks after injection. At the end of the experiment, mice were perfused as described above and expression of virus was confirmed via fluorescence analysis.

### Nociceptor-specific knockdown of BDNF

Nociceptor-specific knockdown of BDNF was achieved with the methods described previously[58]. Briefly, according to the mouse cDNA sequence of BDNF, we generated oligonucleotides corresponding to BDNF-specific shRNA (sense strand: GCATGAAGGCGGCCGCCGGAG; antisense strand: CTCCGGCGGCCGCCTTCATGC), annealed them and cloned them into an AAV2/8 plasmid that designed with a floxed enhanced green fluorescent protein (EGFP)-tagged stop sequence (rAAV2/8-u6-Loxp-CMV-EGFP-Loxp-shRNA BDNF) (Supplementary Fig. 11a). Similarly, a control shRNA (rAAV2/8-u6-Loxp-CMV-EGFP-Loxp) was cloned and packaged (AAV2/8-con-RNA) to serve as a negative control for potential unspecific effects associated with delivery of shRNA. To accomplish nociceptor-specific knockdown of BDNF, loxp-BDNF shRNA-expressing AAV2/8 (AAV-shRNA), and AAV2/8-conRNA (1.0 μl) was injected into L3/L4 DRGs of SNS-Cre mice as described above. AAV2/8-conRNA was injected in a similar way as controls. See the list of reagents and viruses used in Supplementary Table 1.

### Overexpression of NR1 in nociceptive DRG neurons in vivo

The open reading frame of mouse NR1 fused C-terminally with FLAG or FLAG alone was cloned in an AAV expression construct, and chimeric AAV2/8 virions were generated using standard protocols, as shown in detail in Supplementary Fig. 13a. Virions were injected unilaterally into L3/L4 DRGs (1 μl per DRG) in deeply anesthetized mice as described in detail previously[34,93,94]. Mice were tested in behavioral tests 4 weeks after injection. At the end of the experiment, mice were perfused as described above and expression of FLAG was confirmed via immunofluorescence analysis. See the list of viruses used in Supplementary Table 1.

### Expression of AAV2/8-DIO-hM3Dq in DRG in vivo

AAV2/8-CAG-DIO-hM3Dq was constructed and injected into L3/L4 DRGs of SNS-NR1$^{-/-}$ mice as described above. Four weeks after virus expression, vehicle or CNO (10 μM) was intrathecally administered to activate presynaptic terminals of nociceptors. Mechanical hypersensitivity was tested with Von Frey hairs in capsaicin-induced secondary hypersensitivity model in SNS-NR1$^{-/-}$ mice with vehicle or CNO treatment. See the list of reagents and viruses used in Supplementary Table 1.

### Pain models and behavioral testing

Unilateral injection of complete Freund's adjuvant (20 μl) (CFA) was performed into the intraplantar surface of mouse hindpaw. Various tests including western blotting, patch-clamp recordings were conducted at 24 h post CFA injection. The intraplantar capsaicin test was performed by injection of capsaicin (0.06%) into the intraplantar surface of mouse hindpaw, as described previously[34]. Muscle pain was evoked by injecting 20 μl of acidic saline, pH 4 in the gastrocnemius muscle two times at an interval of 3 days and behavioral analysis was started at 24 h until 3 weeks after the second injection[34,59]. Capsaicin-induced secondary mechanical hypersensitivity was induced by injection of capsaicin (0.06%) into the lower thigh of one leg and Von Frey test was performed in the plantar surface of hindpaw where the flare did not reach.

Analysis of latency of paw withdrawal in response to noxious heat was done as described previously in detail (Ugo Basile Inc.)[34]. Mechanical sensitivity was via manual application of von Frey hairs to the plantar surface. Response frequency was calculated as the mean number of withdrawals out of 10 applications of the respective filament at 10 s intervals.

All behavioral measurements were done in awake, unrestrained, age-matched mice of both sexes which were more than 3 months old by individuals who were blinded to the genotype of the mice being analyzed. To minimize the possible influence of gender difference, the number of female or male mice were strictly matched in different groups, i.e. NR1$^{fl/fl}$ group and SNS-NR1$^{-/-}$ group. See the list of reagents used in Supplementary Table 1.

### Intrathecal delivery of drugs in vivo

To enable intrathecal delivery at the level of lumbar spinal segments in mice, a polytetrafluoroethylene catheter was stereotactically inserted after hemilaminectomy at S1-S2 under anesthesia by 1% pentobarbital sodium, as described previously[34]. After a flush with 10 μl of 0.9% saline, the exterior end of catheter was sealed by heat. Penicillin antibiotics were used to prevent infection at the end of intrathecal catheterization. The mice were allowed to recover for 3 days after surgery and only mice showing a complete lack of motor abnormalities were used for further experiments. 5 μl of drugs was applied followed by flushing of the catheter with 10 μl of saline.

### Intraganglionic injection of drugs in vivo

Intraganglionic delivery of CGP78608 (2, 10 μM) was performed via intervertebral foramen route procedures as described previously[95]. Briefly, the bilateral iliac spines were used to locate the L3 and L4 vertebrae of mice. The 26-gauge needle mated to a Hamilton syringe (Hamilton, Reno, NV) was inserted at a 45° angle at the intersection of the lower edge of the ipsilateral L3 and L4 vertebrae and the paravertebral line, infiltrating the intervertebral foramen with 50 μl CGP78608 (2, 10 μM). There was a sense of restriction when the needle entered the transverse foramen, and the paw retraction reaction of mice was the sign of the needle entering the transverse foramen. See the list of reagents used in Supplementary Table 1.

### Pre-embedding immunogold-silver cytochemistry

As previously described[96], animals were anesthetized with pentobarbital sodium and transcardially perfused with 4% paraformaldehyde and 0.05% glutaraldehyde. Vibrotome sections (50 μm) of spinal cord were cut. NR1 or SK2 was detected by the immunogold-silver staining method and CGRP by the immunoperoxidase method. Briefly, sections were incubated in the primary antibodies of goat anti-NR1 (Santa cruz, 1:50) and rabbit anti-CGRP (Abcam, 1:500) or rabbit anti-SK2, (Alomone, 1:1000) and goat anti-CGRP (Abcam, 1:500). The secondary antibodies were a mixture of anti-goat or anti-rabbit IgG conjugated to 1.4 nm gold particles (Nanoprobes, Stony Brook, NY, 1:100) and horseradish peroxidase-conjugated anti-rabbit or anti-goat IgG (1:200 dilution). Silver enhancement was performed with HQ Silver Kit (Nanoprobes) for visualization of NR1 or SK2 immunoreactivity. They were then visualized by the glucose oxidase-3, 3′-diaminobenzidine method for CGRP immunoreactivity.

Immuno-labeled sections were fixed with 0.5% osmium tetroxide, dehydrated in graded ethanol series, then in propylene oxide, and finally flat-embedded in Epon 812 between sheets of plastic. Three to four sections containing both NR1 (or SK2) and CGRP immunoreactivity in the superficial spinal dorsa horn were selected, trimmed and glued onto blank resin stubs. Serial ultrathin sections were cut with an Ultramicrotome (Leica EM UC6, Wetzlar, Germany) using a diamond knife (Diatome, Hatfield, PA) and mounted on formvar-coated mesh grids (6–8 sections/grid). They were then counterstained with uranyl acetate and lead citrate, and observed under a JEM-1230 electron microscope (JEOL LTD, Tokyo, Japan) equipped with CCD camera and its application software (832 SC1000, Gatan, Warrendale, PA). Electron micrographs were arranged and contrast-enhanced by the computer. See the list of reagents and antibodies used in Supplementary Table 1.

### Data analysis & statistics

All statistical analyses were performed in Prism v.8.0 (GraphPad Software, Inc.) and SPSS v.21.0. Data were graphed and analyzed using Graphpad software Prism v.8.0, Microsoft Excel 2013. All data are presented as mean ± standard error of the mean (S.E.M.). The normality test was performed by the Shapiro–Wilk test. The homogeneity of variance test was performed by Levene's test. Data that met these two conditions were analyzed using a two-tailed unpaired or paired $t$ test, one-way analysis of variance (ANOVA) or two-way ANOVA. Data sets that were not normally distributed were analyzed with a nonparametric test, data sets that were normally distributed but not conform to homogeneity of variance test were analyzed with a corrected two-tailed unpaired or paired $t$ test, one-way or two-way ANOVA (Supplementary Table 2). $P < 0.05$ was considered significant.

### Reproducibility

Experiments were repeated independently with similar results at least three times. Micrographic images presented in figures are representative ones from experiments repeated independently: Fig. 4a, b, e (four times), Supplementary Fig. 2d (three times), Supplementary Fig. 9e, f (three times), Supplementary Fig. 10f, g (four times), Supplementary Fig. 11c (three times), Supplementary Fig. 13b–d (three times).

### Reporting summary

Further information on research design is available in the Nature Research Reporting Summary linked to this article.

## Data availability

Source data are provided with this paper. All data are contained in the main text and the supplementary materials, and are available from the corresponding author upon reasonable request. Source data are provided with this paper.

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

## Acknowledgements

The authors are grateful to Robert Feil (Universität Tübingen, Tübingen, Germany) for the kind gift of antibodies. We thank Dunja Baumgartl-Ahlert, Bei Miao, Shengnan Li and Xiaohua Hao for their kind technical help. This work was supported by Natural Science Foundation of China (NSFC) grants (No. 32071002, 31671088, 31471059) to C.L.; grants from Natural Science Foundation of Shaanxi province (No. 2017ZDJC-01) to C.L.; grants from State Key Laboratory of Neuroscience, Shanghai Institutes for Biological Sciences, Chinese Academy of Sciences (No. SKLN-2015B02) to C.L., NSFCs grants to R.G.X. (No. 81870867, 82171212, 31400949) and to H.Z. (No. 81502102); NSFC grants (No. 81730035) and grants from Innovation Teams in Priority Areas Accredited by the Ministry of Science and Technology (No. 2014RA4029) to S.X.W.; grants from Sanming Project of Medicine in Shenzhen (No. SZSM201911011) to H.R.T and S.X.W.; grants from the Ministry of Science and Technology of China to C. L. (No. 2021ZD0203104) and to R.G.X. (No. 2021ZD0203205).

## Author contributions

R.G.X performed spinal cord slice electrophysiology, R.G.X. and W.G.C. conducted virus injection, calcium imaging and behavioral surveys, X.W., S.B.M., and Y.Y.L. performed western blot analysis, immunofluorescence staining, and ultrastructural staining experiments, D.L.L. did spinal cord slice recording, S.B.M., F.W., Z.L. performed pain behavioral and pharmacological tests, F.D.W. and W.B.W conducted virus injection, N.L. and W.J.H. provided help in immunofluorescence and behavioral experiments, H.Z. provided help in data analysis, Z.T.B., S.J.H., H.R.T., T.K., X.Z., R.K. provided critical input on study design and interpretation, S.X.W., C.L. designed studies, supervised the experiments and wrote the manuscript.

## Competing interests

The authors declare no competing interests.
