## [Peer Review File · Nature Communications]

Presynaptic NMDARs on spinal nociceptor terminals
state-dependently modulate synaptic transmission and painReviewers' comments:

Reviewer #1 (Remarks to the Author):

In this study, the authors investigated the role of presynaptic NMDARs (PreNMDARs) in modulating spinal presynaptic plasticity and pain hypersensitivity. By generating nociceptor-specific NR1 knock-out mice and performing spinal slice patch recording in spinal lamina I neurons projecting to the periaqueductal grey (PAG), the authors show that activation of PreNMDARs state-dependently modulate spinal synaptic plasticity. In basal states, activation of PreNMDARs leads to synaptic depression by activating SK channels. Upon injury, tissue inflammation induces PreNMDARs-PKG-I-dependent BDNF secretion from nociceptor terminals, downregulating SK channels, leading to synaptic potentiation.

This is a very comprehensive study on the role of NMDARs in regulating the release probability of spinal nociceptor terminals. The study is a nice extension of the author's previous work in 2012, which showed the involvement of presynaptic mechanisms in spinal LTP and the key role of presynaptic PKG-I signaling in spinal plasticity (Luo et al., Plos Biology 2012). The current study further reveals the involvement of presynaptic BDNF release in downregulating SK channels after NMDAR and PKG-I activation during inflammation and provides an overall picture on the state dependent-modulation of presynaptic plasticity by PreNMDARs in spinal nociceptor terminals. However, I do have some concerns about this work.

Major points:

1. A major concern is related to the use of term pre-LTP. For example, In Fig. 1b, the authors recorded C-fiber-evoked EPSC on spino-PAG neurons under a conditioning LFS and claimed it as pre-LTP. In my opinion, what is showed here is the expression of synaptic LTP between C-fibers and spinal projection neurons, which has been shown to involve both postsynaptic mechanisms (Ikeda et al., Science 2006) and presynaptic mechanisms (Luo et al., Plos Biology 2012). Is it appropriate to claim it solely as pre-LTP?
2. Moreover, PAG-projecting spinal lamina I neurons represent direct information input from periphery. Since C-fiber afferents have both direct (monosynaptic) and indirect (polysynaptic, through local interneurons) to these projection neurons, to what extent does the current monosynaptic recording represent the final C fiber stimulation-induced output to PAG?
3. In both Fig. 1k and Fig. 3e, the authors quantified the distance to diagonal line in PPR. It seems problematic to use positive values for data points both above and below the diagonal line as they represent different biological meaning. For example, in Fig. 1k, the data points above the line indicate the increase of PPR after LFS, while data points below the line represent the decrease of PPR after LFS. Is it appropriate to average the absolute values of the vertical distance between the points and the diagonal line, and then make the conclusion that the release probability is increased?
4. Fig. 3, please clarify how AP5 was applied. It is unclear whether the effect of postsynaptic NMDARs was excluded. In the recording of mini EPSC, since the presynaptic Nav1.8 channels are TTX-resistant, how was action potential blocked?
5. Functionally, peripheral inflammation induces changes in primary afferent neurons, which can affect the same molecule differently if located at different compartments, i.e. the peripheral terminal, soma or central roots. In the SK channel and BDNF exploration, protein levels were checked using whole DRG lysate, were dorsal roots included/checked separately? As the cre-flox /shBDNF knocks out the PKGI, NR1 and BDNF from the neurons, the abolished CFA-induced protein changes could also be attributed to somatic NMDAR changes. I'm concerned about figure 8J summary as well as the article title to conclude the effect of preNMDAR as solely in the terminal.
6. Is BDNF only expressed in peptidergic (CGRP) neurons? What about SK channels? Since the NMDAR, BDNF and SK channels are not nociceptor-specific, even though all the data was examined in Nav1.8 neurons, how specific is this preNMDAR switch mechanism to

pain/inflammation?

Minor points:

1. Fig. 6q-s, it unclear whether BDNF treatment was performed in CFA-inflamed mice in these experiments.

2. The authors confirmed the deletion of functional NMDAR by performing recording in IB4 labelled non-peptidergic nociceptors in DRG (Fig. S2). Since NR1 is deleted in both IB4 and CGRP cells of SNS-NR1^{-/-} mice, it would be necessary to confirm the depletion of functional NMDAR in peptidergic neurons as well.

3. The authors reported that SNS-NR1^{-/-} mice have normal mechanical and thermal threshold (Fig. S10). Using a similar strategy to conditionally knockout NR1 in primary sensory neurons (Advilin-Cre; NR1^{fl/fl}), it has been previously reported that NR1-cKO mice have increased thermal and mechanical pain sensitivity compared with wild-type littermates (Pagadala et al., The Journal of Neuroscience, 2013). Nav1.8-Cre and Advilin-Cre largely target the same group of primary sensory neurons. The authors need to discuss this discrepancy.

4. Page 6, 1st paragraph, the last sentence, "Supplementary Fig. 1de-g" should be "Supplementary Fig. 1e-g".

Reviewer #2 (Remarks to the Author):

In this report, authors investigated possible roles of pre-synaptic PreNMDARs at DRG-spinal dorsal horn sensory transmission. There are several findings related to this work, although mostly are linked by pharmacological and genetic deletion approaches. These findings seems to be specific for pain synapses at the spinal cord. For example, this form of pre-LTP is NMDA receptor dependent, in contrast to previous reports of other central synapses. Furthermore, some of experimental approaches are problematic, for example postsynaptic injection of MK801 cannot block postsynaptic NMDA receptors in neighboring cells. It is difficult to exclude the possible contribution of other postsynaptic NMDA receptors. I would suggest that authors can focus on this NMDA receptor dependent pre-LTP.

1. In the PPR analysis of pre-LTP in the spinal-PAG neurons, NR1^{fl/fl} mice showed much stronger changes of PPR after LFS as compared to SNS-NR1^{-/-} mice. In the pre-LTP condition, the PPR value is supposed to significantly decrease instead of changing strongly after the LFS. Please explain further about the meaning of high variability of the PPR value.

2. Bath application of NMDA receptors may affect NMDA receptors located at other postsynaptic neurons as well as possible astrocyte cells. Inhibition postsynaptic NMDA receptors by injection of MK801 cannot exclude the contribution of other postsynaptic neurons. For example, postsynaptic NMDA receptors located at other cells may be activated, and affect presynaptic functions through releasing diffusible messengers. Additional experiments are needed to check this possibility.

3. Pre-LTP in central synapses are known to be sensitive to kainate receptors. Authors need to perform additional experiments to test if kainate receptor may contribute to this form of pre-LTP at the spinal cord. Also, it will be useful for authors to confirm that this pre-LTP can be completely blocked by AP5 as well as its sensitivity to NR2A vs NR2B receptor inhibition.

4. Can authors record post-LTP in this synapse? It will be great to see the selective blockade from the same recorded neurons of SNS-NR1^{-/-} mice. Furthermore, some examples to show normal NMDA currents from these cells will be helpful (in addition to dorsal horn neurons).

5. In case of injury model (such as CFA), I would expect some synapses were not affected. It will be great if authors can record from neurons that are not affected by CFA injection. Furthermore, some of large DRG neurons may contribute to allodynia. Authors should consider to check these cells as well.

6. According to this study, NR1 may play essential role in PreNMDARs. Is it possible that any conclusion above can be repeated by using selective antagonist of NR1? Does the NR1 have any clinical implication as a key target of PreNMDAR?

7. Please include the number of animals in the figure legends for each averaged data (e.g. n=xx slices /xx mice) if it is possible.

8. Since results can be quite different in different areas, the effect of the AP-5 on the frequency of mEPSCs in visual cortex and in spinal-PAG projecting neurons for instance, it would be much clearer to summarize results on the DRG and spinal-PAG neurons individually at somewhere in the paper.

Reviewer #3 (Remarks to the Author):

The report by Xie et al described a novel and highly interesting role of presynaptic NMDA receptors in regulating nociceptive transmission of information entering the CNS. The paper shows how these pre-synaptic receptors are able to regulate primary afferent excitability and modulate dorsal horn neuronal responses. They show an interaction with SK channels in primary afferents and that this interaction state-dependently regulates excitability in basal and pain states. The studies are appropriate and test of overlying hypothesis. I do have some concerns especially with regard to conditions for patch clamp electrophysiology detailed below. A minor point is that whilst the vast majority of the paper is well written so are some problems with pluralisation throughout which need to be addressed.

Abstract

Line 1 "is" should be "are"

Line 3 "nociceptors" should be "nociceptor"

I think the number of unattributed abbreviations are a problem, for example PKG-I should be explained as should BDNF.

Introduction

Line 6- "amounting" should be "mounting"

The concept of Pre-LTP is interesting but I feel may be a bit of a misnomer. LTP refers to enhanced synaptic transmission, and to my mind involves both the pre and post-synapse. From what I understand the authors are referring to enhanced presynaptic neurotransmitter release from nociceptors as Pre-LTP. This may be long-term and yes, involves the presynaptic terminal but it does not necessarily mean that synaptic transmission MUST be enhanced. I feel that the intro would benefit from a better explanation of what Pre-LTP is- mechanistically.

Results

The authors cannot say that the expression of NR1 in the Dorsal horn (DH) decreased in the mutant mice. There are no data presented to indicate this. Supp Fig 1d does not show this and the western in Supp 1f doesn't support this statement either.

According to the methods DRG patching was performed in the presence of magnesium. As NMDA receptors are blocked by these ions at resting membrane potential how did the authors measure changes in response to NMDA? Were neurons held at a membrane potential which removed the Mg²⁺ block? If so what is this? It seems that most patching was performed at -70mV at which nearly 100% of NMDA receptors would be blocked by Mg²⁺. This seems to be missing from the methods. If neither this or use of Mg²⁺ free aCSF then I cannot understand how recordings were made. Clarification is required as is the use of an appropriate control.

Why were the NR1^{fl/fl} group so variable in panel Supp2h? How was n calculated for these comparisons per animal/per slice or per FOV?

Was any quantification of termination pattern of the primary afferent fibres in the two groups performed? We have a nice image of DH expression of CGRP and IB4 however the density of staining/immunofluorescence the extent of it has not been performed. I think it is reasonable to expect a more thorough analysis to be performed.

I expected Fig 1 a to be a picture of a labelled DH-PAG projection neuron, I think this needs to be included. How can we be certain that the DiI injections in the PAG were accurate? We need to see injection sites in PAG as well. Which part of the PAG (other than right hand side) was injected? What were the coordinates? What was the rationale for this area?

Why was the LTP stimulation protocol chosen? How does this compare to other stimulation protocols for LTP used in vivo and also for other forms of short term plasticity like spinal wind-up?

How much/what percentage of CFA was used in studies?

In Fig 2f are we expecting to see a 165% increase in C-ePSCs? We see a 1.65% increase. Is the y-axis scale incorrect?

While I agree with the general statements about the impact of NMDA and LFS on C-ePSCs in Fig 2i the percentages quoted in the text seem to not match up. For example the authors state a 151.9% increase 30min post-LFS, to me this appears to be nearer 175%?

On this last point whilst the increases are reported whether they are statistically significant is not indicated either in the text or on the figures.

The heat maps in Panel 4c. I cannot find information about what the y-axis shows. X-axis is time, colour in intensity but what is y-axis?

Fig d-Could the changes in calcium signalling to NMDA and Glut be blocked with MK-801 or AP5 in control mice? This seems a logical study to have made especially following similar ones with the electrophysiological recordings. Either MK801 or AP5 could have been applied along with the agonists.

Page 15 line 3 "become" should be "come"

My interpretation of the blot 5k is that the PKG-I expression in both groups increases following CFA and that the differences between the two groups of mice are not reflected in the blot at all. How large were groups in Fig 5L for each column? The authors state group sizes of 3-6 in the legend why the variability in groups? This needs to be addressed.

In the description of intrathecal delivery of drugs more detail is required about the insertion of the catheter. Was this made at the level of the spinal cord drugs were administered to or was the catheter fed from thoracic. Spinal segments down the spinal canal?

The decrease in threshold following capsaicin seems to be present in the SNS-NR1 -/- group compared to basal (Fig 7d), certainly this decrease is comparable to other changes in this study with similar variability in each group. The lack of detailed reporting of statistics in the paper is an issue and should be addressed. The reader would benefit from knowing effect size, details of stats, degrees of freedom etc as is not routinely reported in papers. This may be in the text of as supplementary information/ table.

Responses to reviewers by Xie et al.

The authors would like to thank the reviewers and editors for their scholarly feedback and comments, which are very helpful in improving this study. We have now addressed the reviewers' concerns in details and revised the manuscript accordingly. Please find below detailed point-by-point responses, changes in the main text of the revised manuscript are marked in red color.

Reviewers' Comments:

Reviewer #1:

- 1. In this study, the authors investigated the role of presynaptic NMDARs (PreNMDARs) in modulating spinal presynaptic plasticity and pain hypersensitivity. By generating nociceptor-specific NR1 knock-out mice and performing spinal slice patch recording in spinal lamina I neurons projecting to the periaqueductal grey (PAG), the authors show that activation of PreNMDARs state-dependently modulate spinal synaptic plasticity. In basal states, activation of PreNMDARs leads to synaptic depression by activating SK channels. Upon injury, tissue inflammation induces PreNMDARs-PKG-I-dependent BDNF secretion from nociceptor terminals, downregulating SK channels, leading to synaptic potentiation.*

This is a very comprehensive study on the role of NMDARs in regulating the release probability of spinal nociceptor terminals. The study is a nice extension of the author's previous work in 2012, which showed the involvement of presynaptic mechanisms in spinal LTP and the key role of presynaptic PKG-I signaling in spinal plasticity (Luo et al., Plos Biology 2012). The current study further reveals the involvement of presynaptic BDNF release in downregulating SK channels after NMDAR and PKG-I activation during inflammation and provides an overall picture on the state dependent-modulation of presynaptic plasticity by PreNMDARs in spinal nociceptor terminals. However, I do have some concerns about this work.

Response: We greatly appreciate the reviewer for his (her) positive appraisal and constructive comments. We are pleased that the reviewer finds the study interesting, very comprehensive and the efforts laudable. We have addressed all the concerns raised by the reviewer point by point. Please find the detailed

response as below.

Major points:

2. *A major concern is related to the use of term pre-LTP. For example, In Fig. 1b, the authors recorded C-fiber-evoked EPSC on spino-PAG neurons under a conditioning LFS and claimed it as pre-LTP. In my opinion, what is showed here is the expression of synaptic LTP between C-fibers and spinal projection neurons, which has been shown to involve both postsynaptic mechanisms (Ikeda et al., Science 2006) and presynaptic mechanisms (Luo et al., Plos Biology 2012). Is it appropriate to claim it solely as pre-LTP?*

Response: We understand the reviewer for this concern and apologize for not explaining this clearly. It is important to note that distinct forms of LTP can be induced at synapses between nociceptive C-fibers and spinal-PAG projection neurons with an identical presynaptic stimulation protocol (2 Hz, 2 min, 3 mA), depending on the level of postsynaptic membrane polarization. One form of Hebbian-type LTP, resulting from pairing of postsynaptic depolarization (+30 mV) and low-frequency presynaptic stimulation (2 Hz, 2 min), was induced postsynaptically, namely post-LTP (Ikeda et al., 2006). Besides this, another form of non-Hebbian LTP was reported after a rise in Ca^{2+} due to postsynaptic depolarization in the absence of any presynaptic conditioning stimulation (Naka et al., 2013). Both of this Hebbian and non-Hebbian LTP involves postsynaptic mechanisms, requiring activation of postsynaptic NMDARs. In contrast, our present study demonstrates another form of non-Hebbian LTP at C-fiber synapses with spinal-PAG projection neurons, which is induced by only presynaptic conditioning stimulation without postsynaptic depolarization, namely pre-LTP. In contrast to the involvement of postsynaptic NMDARs in the induction of the above Hebbian and postsynaptic non-Hebbian LTP, this pre-LTP at C-fiber synapses is not dependent on postsynaptic NMDARs, but requires PreNMDARs in spinal terminals of nociceptors. Minimal stimulation and PPR analysis suggest that this pre-LTP at C-fiber synapses involves an increase of transmitter release via a presynaptic mechanism. This is consistent with previous studies showing that peripheral tissue inflammation greatly

enhances excitatory synaptic transmission by increasing presynaptic transmitter release (Xiao et al., 2016).

The concept of pre-LTP has been widely accepted and assumed as a fundamental basis for different brain functions (Castillo, 2012; Monday et al., 2018; Monday and Castillo, 2017; Yang and Calakos, 2013). Arguably, the best characterized form of pre-LTP can be found at the mossy fiber synapse between dentate gyrus granule cells and hippocampal CA3 pyramidal neurons (Nicoll and Schmitz, 2005). In recent years, the list of brain regions found to express presynaptic LTP has markedly grown, which includes but not limited to ACC (Koga et al., 2015), amygdala (Shin et al., 2010; Lopez de Armentia and Sah, 2007), cerebellum (Salin et al., 1996), thalamus (Castro-Alamancos and Calcagnotto, 1999), subiculum (Behr et al., 2009), and neocortex (Chen et al., 2009). Unlike the requirement of PreNMDARs in spinal pre-LTP, the trigger for cortical pre-LTP in some brain regions involves presynaptic kainite receptors and downstream adenylyl cyclase-PKA signalling cascades (Koga et al., 2015; Contractor et al., 2011).

3. *Moreover, PAG-projecting spinal lamina I neurons represent direct information input from periphery. Since C-fiber afferents have both direct (monosynaptic) and indirect (polysynaptic, through local interneurons) to these projection neurons, to what extent does the current monosynaptic recording represent the final C fiber stimulation-induced output to PAG?*

Response: We thank the reviewer for his (her) constructive comments. PAG-projecting spinal lamina I neurons have been reported to be mainly nociceptive-specific and to receive nociceptive primary afferent input from C-fibers and A δ -fibers (Hylden et al., 1986). Further studies revealed that input from C-fibers was found in 94% of the spinal-PAG projection neurons receiving input from the segmental dorsal root (Ruscheweyh et al., 2004). Most prominent was the high incidence of monosynaptic input from primary afferent C-fibres (about two-thirds) to spinal-PAG projection neurons (Ruscheweyh et al., 2004). Detailed information is shown in the table below.

Table: Primary afferent input to spinal lamina I neurons projecting to PAG (cited from Ruscheweyh et al., 2004)

Primary afferent input	Spinal-PAG projection neurons recorded (n = 47)
A δ -fiber monosynaptic	1 (2%)
A δ -fiber polysynaptic	11 (23%)
C-fiber monosynaptic	27 (58%)
C-fiber polysynaptic	17 (36%)
Convergent A- and C-fiber input	9 (19%)

4. *In both Fig. 1k and Fig. 3e, the authors quantified the distance to diagonal line in PPR. It seems problematic to use positive values for data points both above and below the diagonal line as they represent different biological meaning. For example, in Fig. 1k, the data points above the line indicate the increase of PPR after LFS, while data points below the line represent the decrease of PPR after LFS. Is it appropriate to average the absolute values of the vertical distance between the points and the diagonal line, and then make the conclusion that the release probability is increased?*

Response: We thank the review for his (her) constructive comments and apologize for not interpreting this data clearly. PPR analysis is well accepted as an indication of presynaptic mechanisms of LTP, e.g. in the hippocampus (Schulz et al., 1995). In hippocampal neurons, PPR can decrease as well as increase in conjunction with LTP in a manner inversely proportional to the PPR prior to the conditioning stimulus (Schulz et al., 1995). Indeed, we obtained similar results in recordings at spinal synapses between C-fibers and spinal-PAG projection neurons. In spinal slices derived from mice of both genotypes, we found evidence for PPF as well as PPD prior to LTP-inducing conditioning stimulus (LFS) (Fig. 1i). Following conditioning stimulus, a majority of neurons derived from NR1^{fl/fl} mice demonstrated a clear change in PPF or PPD, while neurons derived from SNS-NR1^{-/-} mice did not (see typical examples in Fig. 1i). We plotted the PPR of the entire cohort of recorded neurons at 30 min after LFS as a function of the basal PPR recorded prior to LFS (Fig. 1j). As the

reviewer suggested, we reanalyzed the absolute value of change of PPF and PPD following conditioning LFS, respectively. We found that most of neurons (92% of PPF-expressing neurons and 39% of PPD-expressing neurons) in NR1^{fl/fl} mice consistently showed a decrease in PPR after LFS (blue spots below the diagonal line in Fig. 1j), which is indicative of an increase in the probability of presynaptic neurotransmitter release. This drop in PPR following LFS did not come about or was reduced in SNS-NR1^{-/-} mice. In addition, a cohort of neurons with initial PPD (61% of PPD-expressing neurons) prior to LFS in NR1^{fl/fl} mice showed an increase of PPD after conditioning stimulation. This increase of PPD was also reported in opioid withdrawal-induced spinal LTP (Zhou et al., 2010) as well as in other systems (Maru et al., 1989; Balassa et al., 2013). This increase in PPD after LFS suggest that LFS-induced increase in readily releasable pool of synaptic vesicles may result in more synaptic vesicles which can be released from nerve terminals in response to the second stimulus. Furthermore, we plotted the magnitude of LTP to the change of PPR and observed that in NR1^{fl/fl} mice, higher magnitudes of LTP were consistently linked with a big change in PPR (decrease / increase), which is indicative of presynaptic mechanisms (revised Supplementary Fig. 4). In striking contrast, these changes in PPR were not observed or greatly reduced in SNS-NR1^{-/-} mice, which corresponds to the impaired LTP in SNS-NR1^{-/-} mice. This detailed analysis has been illustrated in the revised Supplementary Fig. 4 and described in lines 19-26, page 9 and lines 1-25, page 10 of the revised manuscript.

Exactly because the above results showed that almost all of PPF and partial PPD display decrease and partial PPD increase in conjunction with spinal LTP, both of which reflect the presynaptic mechanisms, we had originally analyzed the relative change of PPR (decrease/increase) in total to reflect the presynaptic mechanisms involved in spinal pre-LTP, as described previously (Ma et al., 2020). We established a new parameter which can reflect the magnitude of PPR change following LFS, that is the vertical distance to the

diagonal line (shown in Fig. 1j). The higher the magnitude of vertical distance to the diagonal line is, the more the presynaptic mechanisms are reflected. As we mentioned above **in the revised Supplementary Fig. 4**, the higher PPR change is consistently linked with the higher magnitude of LTP. Taking this data together with the above new analysis of absolute change of PPR value as well as failure rate analysis, we can conclude that pre-LTP in spinal-PAG projection neurons comes about via presynaptic mechanisms largely involving an increase in release probability via PreNMDAR.

5. *Fig. 3, please clarify how AP5 was applied. It is unclear whether the effect of postsynaptic NMDARs was excluded. In the recording of mini EPSC, since the presynaptic Nav1.8 channels are TTX-resistant, how was action potential blocked?*

Response: We apologize for not describing this information in details. In Fig. 3, we delivered AP5 by bath application in the presence of MK801 (1 mM) in the intrapipette solution to block the postsynaptic NMDARs. We have described this information in details in **lines 24-25, page 12 and lines 17-18, page 13** of the revised manuscript.

Considering the reviewer's concern about the presence of TTX-resistant Nav1.8 channels in the presynaptic terminals, we have performed a new set of experiments to record mEPSCs in the presence of TTX (0.5 μ M), gabazine (50 μ M), strychnine (10 μ M) as well as A-803467 (0.5 μ M), an Nav1.8 selective blocker, at a holding potential of -70 mV. We found similar results as previously shown. In spinal-PAG projection neurons from NR1^{fl/fl} mice, bath application of AP5 (50 μ M) induced a drastic increase in mEPSCs frequency, but no alteration in mEPSCs amplitude in the basal state. Conversely, upon peripheral inflammation, AP5 application led to a significant depression of mEPSCs frequency, but no change of mEPSCs amplitude. This state-dependent regulation of mEPSCs by AP5 was eliminated in SNS-NR1^{-/-} mice. The above results have been illustrated in **the revised Supplementary Fig. 7a-d** and

described in lines 10-23, page 13 and lines 2-8, page 37 of the revised manuscript.

6. *Functionally, peripheral inflammation induces changes in primary afferent neurons, which can affect the same molecule differently if located at different compartments, i.e. the peripheral terminal, soma or central roots. In the SK channel and BDNF exploration, protein levels were checked using whole DRG lysate, were dorsal roots included/checked separately? As the cre-flox /shBDNF knocks out the PKGI, NR1 and BDNF from the neurons, the abolished CFA-induced protein changes could also be attributed to somatic NMDAR changes. I'm concerned about figure 8J summary as well as the article title to conclude the effect of preNMDAR as solely in the terminal.*

Response: We fully understand the reviewer's concern. To address this concern, we have performed a new set of experiments to observe the protein changes, i.e. PKG-I, NR1, BDNF, SK2 using lysates from dorsal roots of lumbar enlargement under basal and CFA-inflamed state. In general, we observed similar results in dorsal roots lysates as the whole DRG lysates. Detailed results are described below.

First, lysates from L3/L4 dorsal roots in naïve SNS-NR1^{-/-} mice displayed a reduced SK2 expression as compared to NR1^{fl/fl} mice. This strongly supports the assumption that PreNMDARs mediates synaptic depression via interaction with presynaptic SK channels in the basal state. This data has been illustrated in the revised Supplementary Fig. 9a, b and described in lines 4-5, page 16 of the revised manuscript.

Second, in CFA-inflamed state, lysates from L3/L4 dorsal roots showed a dramatic upregulation of phosphorylation of NR1 over basal state in NR1^{fl/fl} mice. In striking contrast, SK2 expression in dorsal roots was significantly reduced over basal state in NR1^{fl/fl} mice, which did not occur in SNS-NR1^{-/-} mice. This resultant downregulation of SK channels in dorsal roots may contribute to relieve synaptic transmission from depression to potentiation by PreNMDARs in the pathological states. This data has been illustrated in the revised

Supplementary Fig. 9g, h and described in lines 1 and 3-5, page 17 of the revised manuscript.

Third, following CFA inflammation, dorsal roots from NR1^{fl/fl} mice also showed a striking increase of PKG-I expression over the basal state. This was markedly reduced in CFA-inflamed SNS-NR1^{-/-} mice. This data has been illustrated in the revised Supplementary Fig. 9i, j and described in lines 17-24, page 17 of the revised manuscript.

Fourth, similar to the DRGs lysates, BDNF derived from dorsal roots lysates also showed dramatic upregulation upon CFA inflammation in NR1^{fl/fl} mice. This upregulation in NR1^{fl/fl} mice was not observed in SNS-NR1^{-/-} mice. It also holds true in PKG-I^{fl/fl} and SNS-PKG-I^{-/-} mice. These data have been illustrated in the revised Supplementary Fig. 10a-d and described in lines 19-26, page 18 and line 1, page 19 of the revised manuscript.

Based on the above new results, we think that the figure 8j summary as well as the article title should be appropriate.

7. *Is BDNF only expressed in peptidergic (CGRP) neurons? What about SK channels? Since the NMDAR, BDNF and SK channels are not nociceptor-specific, even though all the data was examined in Nav1.8 neurons, how specific is this preNMDAR switch mechanism to pain/inflammation?*

Response: We thank the reviewer for his (her) constructive comments. Although NMDARs, BDNF and SK channels are not expressed exclusively in nociceptors, we in the present study utilized the SNS-Cre mice (where Cre recombinase is expressed under Nav1.8 promotor) to produce specific deletion of NMDARs, BDNF etc in nociceptors and unraveled the role and underlying mechanisms of PreNMDARs localized in nociceptors in the spinal synaptic plasticity and pain sensitization. In addition, since large DRG neurons are also well documented to contribute to allodynia under pathological states (Devor, 2009; Song et al., 2012; Xiao and Bennett, 2007), these proteins expressed in large primary sensory neurons may also contribute to pain sensitization via a

similar or alternative mechanism, which remains to be investigated. More importantly, a new set of experiments in our revised manuscript demonstrated that intraganglionic injection of NR1 selective antagonist produced a marked relief of injury-induced pain hypersensitivity. It can be inferred that the mechanisms revealed on PreNMDARs can be switched to pain/inflammation.

Minor points:

8. *Fig. 6q-s, it unclear whether BDNF treatment was performed in CFA-inflamed mice in these experiments.*

Response: We're sorry for not describing this information clearly. In Fig. 6q-s, BDNF treatment was performed in the naïve mice, not CFA-inflamed mice. This is because artificial BDNF incubation of DRGs was to mimic the increased BDNF production condition following CFA inflammation. So BDNF treatment was not performed in CFA-inflamed mice. To make this point clearly, we have described this information **in lines 16-18, page 20** of the revised manuscript.

9. *The authors confirmed the deletion of functional NMDAR by performing recording in IB4 labelled non-peptidergic nociceptors in DRG (Fig. S2). Since NR1 is deleted in both IB4 and CGRP cells of SNS-NR1-/- mice, it would be necessary to confirm the depletion of functional NMDAR in peptidergic neurons as well.*

Response: To confirm the deletion of functional NMDAR in nociceptive DRG neurons, we used Fluor488-conjugated IB4 for live identification of non-peptidergic nociceptors and then performed patch clamp recording in these identified nociceptors. Since no approaches can be used for live identification of peptidergic nociceptors, we had thought to take the deletion confirmation in IB4-labelled non-peptidergic neurons as an example for nociceptive DRG neurons. As the reviewer assumes it necessary to confirm the deletion of functional NMDAR in peptidergic neurons as well, we have conducted a new set of experiments to record IB4-negative small DRG neurons with the addition

of neurobiotin in the intrapipette solution, and then immunostain with CGRP after recording. As shown in the **revised Supplementary Fig. 2d**, NMDAR was confirmed to be functionally deleted in peptidergic nociceptive neurons in SNS-NR1^{-/-} mice. All these results have been illustrated **in the revised Supplementary Fig. 2d** and described in **lines 21-23, page 6 and lines 10-17, page 38** of the revised manuscript.

10. The authors reported that SNS-NR1^{-/-} mice have normal mechanical and thermal threshold (Fig. S10). Using a similar strategy to conditionally knockout NR1 in primary sensory neurons (Advillin-Cre; NR1^{fl/fl}), it has been previously reported that NR1-cKO mice have increased thermal and mechanical pain sensitivity compared with wild-type littermates (Pagadala et al., The Journal of Neuroscience, 2013). Nav1.8-Cre and Advillin-Cre largely target the same group of primary sensory neurons. The authors need to discuss this discrepancy.

Response: We thank the reviewer for this constructive suggestion. In a previous study by Pagadala et al, NR1 was deleted in Advillin-expressing DRG neurons (Pagadala et al., 2013), which resulted in the absence of NR1 in almost all types of DRG neurons. In these advillin-NR1^{-/-} mice, hypersensitivity to mechanical von Frey and thermal stimuli was observed in the basal state, but spontaneous nociceptive response was normal in the formalin test. Unlike advillin-NR1^{-/-} mice, NR1 deletion in Nav1.8-expressing nociceptive DRG neurons (SNS-NR1^{-/-} mice in the present study) showed normal mechanical and thermal sensitivity in the basal state and capsaicin-induced acute spontaneous nociception. This discrepancy may result from complete deletion of NR1 in all DRG neurons using Advillin-Cre versus partial deletion of NR1 in a subset of nociceptive DRG neurons using Nav1.8-Cre driver. As the reviewer suggested, we have discussed this point **in lines 8-10, page 21** of the revised manuscript.

11. Page 6, 1st paragraph, the last sentence, “Supplementary Fig. 1de-g” should be “Supplementary Fig. 1e-g”.

Response: We're very sorry for this typo. We have made corrections in **lines 10-11, page 6** of the revised manuscript.

Reviewer #2:

1. *In this report, authors investigated possible roles of pre-synaptic PreNMDARs at DRG-spinal dorsal horn sensory transmission. There are several findings related to this work, although mostly are linked by pharmacological and genetic deletion approaches. These findings seem to be specific for pain synapses at the spinal cord. For example, this form of pre-LTP is NMDA receptor dependent, in contrast to previous reports of other central synapses. Furthermore, some of experimental approaches are problematic, for example postsynaptic injection of MK801 cannot block postsynaptic NMDA receptors in neighboring cells. It is difficult to exclude the possible contribution of other postsynaptic NMDA receptors. I would suggest that authors can focus on this NMDA receptor dependent pre-LTP.*

Response: We thank the reviewer for his (her) constructive comments. We have addressed all the concerns raised by the reviewer point by point. Please find the detailed response as below.

2. *In the PPR analysis of pre-LTP in the spinal-PAG neurons, NR1fl/fl mice showed much stronger changes of PPR after LFS as compared to SNS-NR1-/- mice. In the pre-LTP condition, the PPR value is supposed to significantly decrease instead of changing strongly after the LFS. Please explain further about the meaning of high variability of the PPR value.*

Response: We thank the review for constructive comments and apologize for not interpreting this data clearly. As we addressed in the above Point #4 for reviewer #1, PPR analysis is well accepted as an indication of presynaptic mechanisms of LTP, e.g. in the hippocampus (Schulz et al., 1995). In hippocampal neurons, PPR can decrease as well as increase in conjunction with LTP in a manner inversely proportional to the PPR prior to the conditioning stimulus (Schulz et al., 1995). Indeed, we obtained similar results in recordings

at spinal synapses between C-fibers and spinal-PAG projection neurons. In spinal slices derived from mice of both genotypes, we found evidence for PPF as well as PPD prior to LTP-inducing conditioning stimulus (LFS) (Fig. 1i). Following conditioning stimulus, a majority of neurons derived from NR1^{fl/fl} mice demonstrated a clear change in PPF or PPD, while neurons derived from SNS-NR1^{-/-} mice did not (see typical examples in Fig. 1i). We plotted the PPR of the entire cohort of recorded neurons at 30 min after LFS as a function of the basal PPR recorded prior to LFS (Fig. 1j). As the reviewer suggested, we analyzed the absolute value of change of PPF and PPD following conditioning LFS, respectively. We found that most of neurons (92% of PPF-expressing neurons and 39% of PPD-expressing neurons) in NR1^{fl/fl} mice consistently showed a decrease in PPR after LFS (blue spots below the diagonal line in Fig. 1j), which is indicative of an increase in the probability of presynaptic neurotransmitter release. This drop in PPR following LFS did not come about or was reduced in SNS-NR1^{-/-} mice. In addition, partial neurons with initial PPD (61% of PPD-expressing neurons) prior to LFS in NR1^{fl/fl} mice showed an increase of PPD after conditioning stimulation. This increase of PPD was also reported in opioid withdrawal-induced spinal LTP (Zhou et al., 2010) as well as in other systems (Maru et al., 1989; Balassa et al., 2013). This increase in PPD after LFS suggest that LFS-induced increase in readily releasable pool of synaptic vesicles may result in more synaptic vesicles which can be released from nerve terminals in response to the second stimulus. Furthermore, we plotted the magnitude of LTP to the change of PPR and observed that in NR1^{fl/fl} mice, higher magnitudes of LTP were consistently linked with a big change in PPR (decrease/increase), which is indicative of presynaptic mechanisms (revised Supplementary Fig. 4). In striking contrast, these changes in PPR were not observed or greatly reduced in SNS-NR1^{-/-} mice, which corresponds to the impaired LTP in SNS-NR1^{-/-} mice. This detailed analysis has been illustrated in the revised Supplementary Fig. 4 and described in lines 19-26, page 9 and lines 1-25, page 10 of the revised

manuscript.

Exactly because the above results showed that almost all of PPF and partial PPD display decrease and partial PPD increase in conjunction with spinal LTP, both of which reflect the presynaptic mechanisms, we had originally analyzed the relative change of PPR (decrease/increase) in total to reflect the presynaptic mechanisms involved in spinal pre-LTP, as described previously (Ma et al., 2020). We established a new parameter which can reflect the magnitude of PPR change following LFS, that is the vertical distance to the diagonal line (shown in Fig. 1j). The higher the magnitude of vertical distance to the diagonal line is, the more the presynaptic mechanisms are reflected. As we mentioned above **in the revised Supplementary Fig. 4b**, the higher PPR change is consistently linked with the higher magnitude of LTP. It is thus reasonable that NR^{fl/fl} mice show high variability in vertical distance to the diagonal line because there is variability in LTP magnitude in individual neurons. Taken together, we have reanalyzed the absolute change value of PPF and PPD and illustrated these data **in the revised Supplementary Fig. 4** and described **in lines 19-26, page 9 and lines 1-26, page 10** of the revised manuscript. In conclusion, PPR analysis with failure rate analysis strongly support the inference that pre-LTP in spinal-PAG projection neurons comes about via presynaptic mechanisms largely involving an increase in release probability via PreNMDARs.

- 3. Bath application of NMDA receptors may affect NMDA receptors located at other postsynaptic neurons as well as possible astrocyte cells. Inhibition postsynaptic NMDA receptors by injection of MK801 cannot exclude the contribution of other postsynaptic neurons. For example, postsynaptic NMDA receptors located at other cells may be activated, and affect presynaptic functions through releasing diffusible messengers. Additional experiments are needed to check this possibility.*

Response: We thank the reviewer for his (her) constructive comments. We fully agree with the reviewer that in addition to affect PreNMDARs in spinal

nociceptor terminals, bath application of NMDA may also affect the postsynaptic NMDA receptors located at other cells, e.g. other neurons or glial cells, and then affect presynaptic functions even in the blockade of postsynaptic NMDA receptors in the recorded neuron with intrapipette injection of MK801. However, it is important to note that the primary aim of this experiment is to address whether PreNMDARs located in spinal nociceptor terminals play a crucial role on exogenous NMDA-induced modulation of synaptic transmission. Since the above situation that the reviewer mentioned exists equivalently in both NR1^{fl/fl} and SNS-NR1^{-/-} mice, it does not influence the conclusion that PreNMDARs in spinal nociceptor terminals play a critical role in exogenous NMDA-induced modulation of synaptic transmission.

4. *Pre-LTP in central synapses are known to be sensitive to kainate receptors. Authors need to perform additional experiments to test if kainate receptor may contribute to this form of pre-LTP at the spinal cord. Also, it will be useful for authors to confirm that this pre-LTP can be completely blocked by AP5 as well as its sensitivity to NR2A vs NR2B receptor inhibition.*

Response: We fully agree with the reviewer that pre-LTP in central synapses has been shown to be sensitive to kainate receptors. For example, GluK1-containing kainite receptors are required for pre-LTP in the anterior cingulate cortex (ACC), amygdala and hippocampus (Koga et al., 2015; Shin et al., 2010, 2013; Bortolotto et al., 1999). To address whether kainate receptors contribute to spinal pre-LTP, we have conducted new experiments by observing the change of spinal pre-LTP in the presence of kainite receptor antagonist, UBP310 (10 μ M). As shown in the **revised Supplementary Fig. 3g, h**, spinal pre-LTP remained intact in the presence of UBP310 (10 μ M), indicating no involvement of kainite receptors in spinal pre-LTP. In contrast, spinal pre-LTP was completely blocked by bath application of AP5 (50 μ M), an NMDARs antagonist or CGP78608 (1 μ M), an NR1-selective antagonist (shown in **the revised Supplementary Fig. 3c-f**). However, this pre-LTP was not sensitive to

NR2A and NR2B receptor inhibition, as characterized by the unaltered pre-LTP in the presence of PEAQX (0.4 μ M), an NR2A antagonist or Ro25-6981, an NR2B antagonist (0.5 μ M) (as shown below). These results collectively indicate that NR1-containing PreNMDARs are functionally linked to pre-LTP at synapses between nociceptors and spinal-PAG projection neurons. The insensitivity of spinal pre-LTP to NR2A and NR2B inhibition suggests the possible involvement of other NMDARs composition, i.e. NR1/NR2D (or NR2C), NR1/NR3 in nociceptors. This is consistent with the previous observation that DRG neurons, especially C-type nociceptive neurons were found to contain diheteromeric NMDARs containing NR1/NR2D or NR1/NR2C subunits with a lower sensitivity to extracellular magnesium (Marvizon et al., 2002). In addition, our preliminary data has shown an involvement of NR3 subunit localized in nociceptors in the spinal pre-LTP. Since this has been out of the scope of the present study and will be investigated in the future study, we just showed the data for kainite receptors antagonist and AP5 in the revised Supplementary Fig. 3 and described in lines 20-24, page 8 of the revised manuscript. The data for NR2A and NR2B are shown below.

Figure legend: The effect of NR2A and NR2B subunits on the spinal pre-LTP induced by low frequency conditioning stimulus. (a, b) Time course (a) and quantitative summary (b) showing that conditioning LFS-induced spinal pre-LTP in spinal-PAG projection neurons remained intact in the presence of NR2A antagonist, PEAQX (0.4 μ M). (c, d) Time course (c) and quantitative summary (d) of bath application of NR2B antagonist, Ro25-6981 (0.5 μ M) on the spinal pre-LTP. Data are presented as mean \pm S.E.M. Paired t test. * $P < 0.05$, $n = 3-4$.

5. *Can authors record post-LTP in this synapse? It will be great to see the selective blockade from the same recorded neurons of SNS-NR1^{-/-} mice. Furthermore, some examples to show normal NMDA currents from these cells will be helpful (in addition to dorsal horn neurons).*

Response: We thank the reviewer for his (her) constructive comments. A non-Hebbian post-LTP has been reported at C-fiber synapses in spinal lamina I neurons by postsynaptic depolarization conditioning stimulation in the absence of presynaptic conditioning stimulation (Naka et al., 2013). As the reviewer suggested, we have now recorded this post-LTP and compared their differences between NR1^{fl/fl} and SNS-NR1^{-/-} mice. In NR1^{fl/fl} mice, postsynaptic conditioning stimulation (pulsed and sustained depolarization protocol: 6 depolarizing pulses for 1s combined with 22s steady depolarization with 13s interstimulus interval in between) induced LTP of C-eEPSCs to $155.6 \pm 9.72\%$ of control after 30 min ($n = 7, P = 0.009$). This post-LTP remained intact in SNS-NR1^{-/-} mice ($157.4 \pm 18.83\%$ of control, $n = 5, P = 0.0243$). This result suggests that PreNMDARs are not involved in the post-LTP induced by postsynaptic depolarization. This inference is consistent with the notion that post-LTP induced by this paradigm did not require the involvement of presynaptic mechanism (Naka et al., 2013).

In the recorded postsynaptic neurons, normal NMDA currents were observed in SNS-NR1^{-/-} mice as compared to NR1^{fl/fl} mice. All the above results are shown below.

Figure legend: Post-LTP induced by postsynaptic depolarization in spinal-PAG

projection neurons is comparable in NR1^{fl/fl} and SNS-NR1^{-/-} mice. Time course (a), representative traces (b) as well as quantitative summary (c) of LTP induction after pulsed and sustained postsynaptic depolarization in spinal-PAG projection neurons derived from NR1^{fl/fl} and SNS-NR1^{-/-} mice. (d) NMDA-induced inward current in the postsynaptic patched neurons remained intact in SNS-NR1^{-/-} mice in comparison with NR1^{fl/fl} mice. Data are presented as mean \pm S.E.M. Kruskal-Wallis *H* test, **P* < 0.05, ****P* < 0.001, n = 5-7.

6. *In case of injury model (such as CFA), I would expect some synapses were not affected. It will be great if authors can record from neurons that are not affected by CFA injection. Furthermore, some of large DRG neurons may contribute to allodynia. Authors should consider to check these cells as well.*

Response: We thank the reviewer for his (her) constructive comments. In the present study, our primary aim is to address the role of presynaptic NMDARs (PreNMDARs) localized on spinal nociceptor terminals on spinal synaptic transmission and its action on pain. Especially, we mainly focus on addressing how and by which mechanisms PreNMDARs in spinal nociceptor terminals contribute to presynaptic LTP and hence mediate pain sensitization upon inflammation/injury. Since LTP can only be evoked at synapses between C-nociceptors and spinal lamina I projection neurons, but not at synapses between A-fibers (large DRG neurons) and spinal lamina I projection neurons (Ikeda et al., 2003, 2006; Sandkühler, 2009, 2015), the present study is not targeting on large DRG neurons. We fully agree with the reviewer that large DRG neurons may contribute to allodynia, but other mechanisms may underlie this action which is out of the scope of our present study. This is what we're going to elucidate in the future.

7. *According to this study, NR1 may play essential role in PreNMDARs. Is it possible that any conclusion above can be repeated by using selective antagonist of NR1? Does the NR1 have any clinical implication as a key target of PreNMDAR?*

Response: We appreciate the reviewer for the constructive comments. It is well known that NR1 is the essential subunit of NMDARs, deletion of NR1 subunit

would render the absence of functional NMDARs. As the reviewer suggested, we have tested the effect of NR1-selective antagonist, CGP78608 (1 μ M) on spinal pre-LTP. Bath application of CGP78608 (1 μ M) was found to largely eliminate spinal pre-LTP ($84 \pm 5.98\%$ of control, $n = 5$). This data has now been illustrated in **the revised Supplementary Fig. 3e, f** and described in **lines 21-23, page 8** of the revised manuscript.

To address the potential clinical implication of NR1 as a key target of PreNMDARs, we have conducted a new set of experiments to assess whether intraganglionic injection of CGP78608 exerts any effect on capsaicin-induced secondary hyperalgesia. This intraganglionic delivery of CGP78608 can affect the DRG and its attached dorsal roots. Lower thigh injection of capsaicin induced a dramatic secondary hyperalgesia in the ipsilateral hindpaw. Intraganglionic administration of CGP78608 was applied at 20 min after capsaicin injection when secondary hyperalgesia was fully established. As shown in the revised Fig. 8i, intraganglionic administration of CGP78608 (2 and 10 μ M) almost completely reversed the capsaicin-induced secondary hyperalgesia. This strongly suggests the potential clinical implication of NR1 as a key target of PreNMDARs in nociceptors in treating the injury-induced pain sensitization. Since the DRG lies outside of the blood-brain barrier (Devor, 1999), it can be inferred that targeting the NR1-containing PreNMDARs in nociceptors represents a promising strategy for fulfillment of optimal analgesic therapeutics with least side effects. The above data has now been illustrated in **the revised Fig. 8i** and described in **lines 10-20, page 25** of the revised manuscript.

8. *Please include the number of animals in the figure legends for each averaged data (e.g. $n=xx$ slices / xx mice) if it is possible.*

Response: We apologize for missing this information. We have now added the detailed information of the number of slices/animals used in each experiment in the new **Supplementary Table 2**, including detailed information about

statistical analysis. Please see in the Supplementary Table 2 of the revised manuscript.

9. *Since results can be quite different in different areas, the effect of the AP-5 on the frequency of mEPSCs in visual cortex and in spinal-PAG projecting neurons for instance, it would be much clearer to summarize results on the DRG and spinal-PAG neurons individually at somewhere in the paper.*

Response: We thank the reviewer for his (her) constructive comments. In the present study, we mainly focus on addressing the role of presynaptic NMDARs (PreNMDARs) located on spinal nociceptor terminals (central terminals of C-type DRG neurons) in the synaptic transmission at synapses between C-nociceptors and spinal-PAG projection neurons as well as pain sensitization. Therefore, DRG and spinal-PAG neurons are not like visual cortex and spinal-PAG projecting neurons, they are connected with the first synapse in pain pathway. Therefore, it is hard to isolate the DRG and spinal-PAG neurons as two different regions individually in the present study. What we demonstrated in the present study is that PreNMDARs localized in spinal nociceptor terminals (central terminals of C-type DRG neurons) modulate synaptic transmission at synapses between C nociceptors and spinal-PAG projection neurons in a nociceptive tone-dependent manner. PPR and mEPSCs analyses revealed that this is mediated via modulation of presynaptic transmitter release probability by PreNMDARs in spinal nociceptor terminals. All of these has been described in the manuscript. Of course, it remains to be further elucidated how PreNMDARs modulates the neuronal excitability of the cell bodies of DRG neurons in the future study.

Reviewer #3:

1. *The report by Xie et al described a novel and highly interesting role of presynaptic NMDA receptors in regulating nociceptive transmission of information entering the CNS. The paper shows how these pre-synaptic receptors are able to regulate primary afferent excitability and modulate*

dorsal horn neuronal responses. They show an interaction with SK channels in primary afferents and that this interaction state-dependently regulates excitability in basal and pain states. The studies are appropriate and test of overlying hypothesis. I do have some concerns especially with regard to conditions for patch clamp electrophysiology detailed below. A minor point is that whilst the vast majority of the paper is well written so are some problems with pluralisation throughout which need to be addressed.

Response: We greatly appreciate the reviewer for his (her) positive appraisal and constructive comments. We are pleased that the reviewer finds the study novel, highly interestingly and well-written. We have addressed all the concerns raised by the reviewer point by point. Please find the detailed response as below.

Abstract

2. *Line 1 “is” should be “are”*

Response: We're very sorry for this grammar error. We have now corrected “is” into “are” in **line 2, page 2** of the revised manuscript.

3. *Line 3 “nociceptors” should be “nociceptor”*

Response: We thank the reviewer for his (her) careful corrections. We have now corrected “nociceptors” into “nociceptor” in **line 4, page 2** of the revised manuscript.

4. *I think the number of unattributed abbreviations are a problem, for example PKG-I should be explained as should BDNF.*

Response: We apologize for these unattributed abbreviations and appreciate the reviewer for kind reminder very much. We have thoroughly checked the whole manuscript and provided the full name for all the abbreviations when they appear at first time. Please see changes in **lines 4-5 and 16, page 5 and lines 2-3, page 8** of the revised manuscript.

Introduction

5. Line 6- “amounting” should be “mounting”

Response: We thank the reviewer for very careful corrections. We have now corrected “amounting” into “mounting” in **line 7, page 3** of the revised manuscript.

6. *The concept of Pre-LTP is interesting but I feel may be a bit of a misnomer. LTP refers to enhanced synaptic transmission, and to my mind involves both the pre and post-synapse. From what I understand the authors are referring to enhanced presynaptic neurotransmitter release from nociceptors as Pre-LTP. This may be long-term and yes, involves the presynaptic terminal but it does not necessarily mean that synaptic transmission MUST be enhanced. I feel that the intro would benefit from a better explanation of what Pre-LTP is- mechanistically.*

Response: Long-term synaptic plasticity is critically for experience-dependent adjustments of many brain functions. While most research has focused on the mechanisms that underlie postsynaptic forms of plasticity, comparatively little is known about how neurotransmitter release is altered in a long-term manner. Recent studies demonstrate that this form of presynaptic plasticity is ubiquitously expressed in a variety of brain regions, and accumulating evidence indicates that it may underlie behavioral adaptations occurring *in vivo* (Yang and Calakos, 2013; Monday and Castillo, 2017; Monday et al., 2018; Castillo, 2012). Presynaptic plasticity can express as presynaptic LTP (pre-LTP) or LTD (pre-LTD), depending on the increase or decrease of neurotransmitter release. Interestingly, presynaptic forms of LTP/LTD can coexist with classical forms of postsynaptic plasticity and can occur at both excitatory and inhibitory synapses.

Arguably, the best characterized form of pre-LTP can be found at the mossy fiber synapse between dentate gyrus granule cells and hippocampal CA3 pyramidal neurons (Nicoll and Schmitz, 2005). In recent years, the list of brain regions found to express presynaptic LTP has markedly grown, which includes

but not limited to ACC (Koga et al., 2015), amygdala (Shin et al., 2010; Lopez de Armentia and Sah, 2007), cerebellum (Salin et al., 1996), thalamus (Castro-Alamancos and Calcagnotto, 1999), subiculum (Behr et al., 2009), and neocortex (Chen et al., 2009). In our case, we reported a form of pre-LTP at primary afferent synapses between C-nociceptors and spinal-PAG projection neurons and unraveled its underlying mechanisms. Low-frequency conditioning stimulus applied to the dorsal root at C-fiber intensity without postsynaptic depolarization produced a LTP of synaptic transmission between C-nociceptors and spinal-PAG projection neurons. This was accompanied by an increase in neurotransmitter release in a long-term manner. The reviewer mentions that enhanced presynaptic neurotransmitter release does not necessarily mean that synaptic transmission must be enhanced. Yes, it depends on which neurotransmitter is released, e.g. excitatory neurotransmitter such as glutamate or inhibitory neurotransmitter such as GABA. In our case at primary afferent synapses, the primary afferents use glutamate as main neurotransmitter. Thus long-term enhanced neurotransmitter release from spinal nociceptor terminals can lead to presynaptic LTP of synaptic transmission. The concept of pre-LTP has been widely accepted and assumed as a fundamental basis for different brain function (Castillo, 2021; Monday et al., 2018; Monday and Castillo, 2017; Yang and Calakos, 2013).

Results

7. *The authors cannot say that the expression of NR1 in the Dorsal horn (DH) decreased in the mutant mice. There are no data presented to indicate this. Supp Fig 1d does not show this and the western in Supp 1f doesn't support this statement either.*

Response: We're very sorry for not explaining this data very clearly. We meant that in the spinal cord of SNS-NR1^{-/-} mice, the intensity of anti-NR1 immunoreactivity was largely decreased in the superficial dorsal horn laminae where nociceptive primary afferents mainly terminate. We have quantitatively

analysed the intensity of NR1 immunoreactivity in the superficial dorsal horn between two genotypes and found that anti-NR1 immunoreactivity was greatly reduced in SNS-NR1^{-/-} mice as compared to NR1^{fl/fl} mice (revised Supplementary Fig. 1d). This result would be expected from SNS-Cre-mediated gene deletion in nociceptors. The quantitative summary has been illustrated in the revised Supplementary Fig. 1d.

In contrast, intrinsic neurons in the spinal cord entirely maintained NR1 immunoreactivity (indicated by arrow in the revised Supplementary Fig. 1d). Furthermore, NR1 expression was entirely unaltered in the brains of SNS-NR1^{-/-} mice (Supplementary Fig. 1d, right panel). These results further confirmed a DRG-specific loss of NR1 in SNS-NR1^{-/-} mice.

Lysates for western blot analysis in the spinal dorsal horn consist of several sources, e.g. intrinsic spinal neurons, primary afferent terminals (from both nociceptors and non-nociceptors), descending terminals, glial cells, it is thus reasonable that no alteration of NR1 expression was seen using western blotting in the spinal dorsal horn in SNS-NR1^{-/-} mice. We have described the above detailed results in lines 3-11, page 6 of the revised manuscript.

8. *According to the methods DRG patching was performed in the presence of magnesium. As NMDA receptors are blocked by these ions at resting membrane potential how did the authors measure changes in response to NMDA? Were neurons held at a membrane potential which removed the Mg2+ block? If so what is this? It seems that most patching was performed at -70mV at which nearly 100% of NMDA receptors would be blocked by Mg2+. This seems to be missing from the methods. If neither this or use of Mg2+ free aCSF then I cannot understand how recordings were made. Clarification is required as is the use of an appropriate control.*

Response: We apologize for not explaining the methods very clearly. NMDA-induced current in the DRG in the present study was performed at a holding potential of -40 mV in the presence of magnesium, at which magnesium blocking effect on the NMDA receptors was removed (as shown below). Another

reason that we did DRG patching in the physiological concentration of magnesium is because DRG neurons, especially C-type nociceptive neurons were found to contain diheteromeric NMDARs containing NR1/NR2D or NR1/NR2C subunits with a lower sensitivity to extracellular magnesium (Marvizon et al., 2002). A previous study by Marvizon *et al* reported that DRG neurons have two different NMDA receptors, one containing the NR1, NR2D, and possibly the NR2C subunits, found only in C-type nociceptive neurons, and the diheteromer NR1/NR2B, present in both A and C-type DRG neurons. Diheteromeric NMDARs containing NR2A or NR2B subunits generate 'high-conductance' channel openings with a high sensitivity to block by magnesium, whereas NR2C- or NR2D-containing receptors give rise to 'low-conductance' openings with a lower sensitivity to extracellular magnesium (Monyer et al., 1994; Cull-Candy et al., 2001). The above detailed information for current recording has now been described in lines 23-24, page 37 of the revised manuscript.

```
Experiment type: other.
File GUID:      {4986991D-F3E3-46BF-A478-64D2319E0DBA}.

Input signal: Vol_Clamp1.    MultiClamp 700 telegraph parameters:
  Additional gain: xl.00.
  Lowpass filter frequency: 10000.00 Hz.
  Membrane capacitance: 0.00 pF.
Triggering not supported

Sampling rate: 10.00 kHz (interval 100.00 衞).

Input signal:
Vol_Clamp1.

Signal Cmd 0 holding level: 0.00 mV.
Signal VolClamp1 holding level: -40.00 mV.
Signal AO #2 holding level: 0.00 mV.
Signal AO #3 holding level: 0.00 mV.
Signal AO #4 holding level: 0.00 mV.
```

Figure legend: Raw data showing that NMDA-induced current in the DRG in the present study was performed at a holding potential of -40 mV in the presence of magnesium, at which magnesium blocking effect on the NMDA receptors was removed.

Regarding the other patching in the spinal-PAG projection neurons for spinal pre-LTP recording, we patched the recorded spinal neurons at a holding

potential of -70 mV. This is the typical stimulating protocol for inducing pre-LTP, including brain regions such as amygdala, ACC etc (Koga et al., 2015; Shin et al., 2010). Under this condition, only presynaptic conditioning stimulus application to the primary afferents without depolarization of postsynaptic neurons can be achieved. This is exactly what our primary aim is.

9. *Why were the NR1fl/fl group so variable in panel Supp2h? How was n calculated for these comparisons per animal/per slice or per FOV?*

Response: We're very sorry for the big variation in panel Supp2h. We have now added more new cases. The new results have now been illustrated **in the revised Supplementary Fig. 2i** (8-10 sections from 3 animals per group).

10. *Was any quantification of termination pattern of the primary afferent fibres in the two groups performed? We have a nice image of DH expression of CGRP and IB4 however the density of staining/immunofluorescence the extent of it has not been performed. I think it is reasonable to expect a more thorough analysis to be performed.*

Response: We apologize for not showing quantification of CGRP and IB4 staining in the spinal dorsal horn. As the reviewer suggested, we have now shown the detailed analysis of the density of CGRP and IB4 staining in the spinal dorsal horn from two genotypes **in the revised Supplementary Fig. 2j**.

11. *I expected Fig 1 a to be a picture of a labelled DH-PAG projection neuron, I think this needs to be included. How can we be certain that the Dil injections in the PAG were accurate? We need to see injection sites in PAG as well. Which part of the PAG (other than right hand side) was injected? What were the coordinates? What was the rationale for this area?*

Response: We thank the reviewer for his (her) constructive comments and apologize for not showing this information. Regarding Dil injection, mice received a single injection of 100 nl of 2.5% Dil into the ventrolateral part of right PAG according to coordinates (AP: 0.1 mm; DV: 5.1 mm; ML: 0.4 mm measured from the lambda) derived from the atlas of Paxinos and Watson.

Ventrolateral part of PAG was chosen for injection since it has been shown to be the predominant area for termination of spinal projection neurons (Spike et al., 2003; Ikeda et al., 2006). We have provided detailed Dil injection site and Dil-labelled neurons in spinal lamina I **in the revised Fig. 1a**. All of the above detailed information about injection coordinates has been supplemented in **lines 11-14, page 35** of the revised manuscript.

12. Why was the LTP stimulation protocol chosen? How does this compare to other stimulation protocols for LTP used in vivo and also for other forms of short term plasticity like spinal wind-up?

Response: Based on previous studies (Ikeda et al., 2006; Luo et al., 2012, 2014; Sandkühler, 2015), we chose this low frequency conditioning stimulation protocol for inducing long-term potentiation (LTP) in spinal-PAG projection neurons in the present study. It has been shown that activation of nociceptive nerve afferents at high frequency (100 Hz) and low frequency (2 Hz) together with postsynaptic depolarization can trigger LTP at spinal synapses between nociceptor terminals and different population of spinal neurons which project to different region of brain (Ikeda, et al., 2003, 2006; Sandkühler, 2015; Luo et al., 2014). In brief, spinal-PAG projection neurons undergo LTP in response to conditioning low frequency stimulation (LFS, 2 Hz) of nociceptive primary afferents, whereas spinal-PB (parabrachial nucleus)-projection neurons produce LTP in response to high frequency stimulation (HFS, 100 Hz) of nociceptive primary afferents (Ikeda, et al., 2003, 2006; Sandkühler, 2015). However, for most of the C-fiber afferents, it is not typical to discharge at rates as high as 100 Hz. Upon inflammation or injury, C-fibers usually discharge at considerably lower rates: for example, 1–10 Hz (Sun et al., 2012). Therefore, conditioning low frequency stimulation (e.g., 2 Hz), but not high frequency bursts, more accurately resembles the low-frequency afferent barrage that occurs under pathological states. Evidence from our and others' previous studies has accumulated that synaptic LTP evoked by natural, asynchronous

low-rate discharges in C-nociceptors on spinal-PAG neurons constitute a very fitting correlate of spinal amplification phenomena underlying pain and hyperalgesia (Ikeda et al., 2006; Luo et al., 2012, 2014; Sandkühler, 2015). Therefore, we took low frequency conditioning stimulation as LTP-inducing protocol in lamina I spinal-PAG projection neurons in the present study. This is further supported by *in vivo* study where conditioning low frequency stimulation of sciatic nerve at C-fiber intensity (2 Hz, 2 min, C-fiber strength) has been found to induce LTP of C-fiber evoked field potentials in the intact animals (Drdla and Sandkuhler, 2008; Ikeda et al., 2006).

Spinal wind-up is a form of activity-dependent plasticity characterized by a progressive increase in action potential output from dorsal horn neurons during a train of repeated low-frequency C-fiber stimuli (< 5 Hz). It is mostly observed in the wide-dynamic range neurons in the deep dorsal horn (mostly lamina V) with a stimulation frequency between 0.5 and 3 Hz at C-fiber intensity (Schouenborg and Sjölund, 1983; Gozariu et al., 1997; Herrero et al, 2000). Below the critical frequency of 0.2-0.3 Hz, wind-up is not observed, and above frequencies of 20 Hz the usual observation is a habituation of the response or wind-down (Schouenborg, 1984). In contrast, wide-dynamic range neurons in the superficial dorsal horn showed a lesser degree of wind-up, and nociceptive-specific neurons (mostly located in superficial lamina) did not show marked wind-up (Schouenborg and Sjölund, 1983; Herrero et al, 2000). Spinal lamina I neurons with a projection to PAG have been reported to be mainly nociceptive-specific and to receive nociceptive primary afferent input (Hylden et al., 1986).

13. How much/what percentage of CFA was used in studies?

Response: We used 20 µl of original concentration of CFA (Sigma-Aldrich, F5881) and described this information in “Pain models and behavioral testing” section in the materials and methods part of our original manuscript. This volume is widely used in the literatures (Luo et al., 2012; Simonetti et al., 2013,

2015; Han et al., 2016). Please see this detailed information in **lines 6-7, page 43** of the revised manuscript.

14. *In Fig 2f are we expecting to see a 165% increase in C-eEPSCs? We see a 1.65% increase. Is the y-axis scale incorrect?*

Response: We thank the reviewer for his (her) careful and kind reminder. We're very sorry for this error. The y-axis should be multiplied by 100. We have corrected this in the revised manuscript. Please see changes in the **revised Fig. 2f**.

15. *While I agree with the general statements about the impact of NMDA and LFS on C-ePSCs in Fig 2i the percentages quoted in the text seem to not match up. For example the authors state a 151.9% increase 30min post-LFS, to me this appears to be nearer 175%? On this last point whilst the increases are reported whether they are statistically significant is not indicated either in the text or on the figures.*

Response: We thank the reviewer for his (her) careful reading. The magnitude of LTP at 30 min post-LFS was calculated as the average of 4 eEPSCs in last two minutes (29-30 min post-LFS), which we think more reasonable. That is why the magnitude of LTP reads the increase of 151.9% but not nearer 175%. For the effect of 2nd NMDA on eEPSCs after LFS, we compared the amplitude of eEPSCs at time points indicated by ④ in Fig. 2i to that prior to NMDA indicated by ③, which is significantly different and shown in Fig. 2k (please see last two columns). The last point the reviewer mentioned is already after washout of NMDA, we should not compare with this time point. This misunderstanding may be caused by our marked light blue column background, which is out of the NMDA perfusion time. We're very sorry for this and have now drawn this marked background back to exactly after NMDA application.

16. *The heat maps in Panel 4c. I cannot find information about what the y-axis shows. X-axis is time, colour in intensity but what is y-axis?*

Response: We apologize for not explaining this clearly in the figure legend. Each row in y-axis represents a GCaMP6s-labelled presynaptic terminal puncta. We have now described this detailed explanation in **the revised figure legend for the revised Fig. 4c in lines 9-11, page 58** of the revised manuscript.

17. *Fig d-Could the changes in calcium signalling to NMDA and Glut be blocked with MK-801 or AP5 in control mice? This seems a logical study to have made especially following similar ones with the electrophysiological recordings. Either MK801 or AP5 could have been applied along with the agonists.*

Response: We agree with the reviewer. As the reviewer suggested, we have applied AP5 along with NMDA or Glut in this calcium imaging experiment. Pretreatment with NMDARs antagonist, AP5 (50 μ M) abolished NMDA-induced Ca^{2+} accumulation. Similarly, AP5 greatly inhibited the magnitude of Glut-induced Ca^{2+} response as well. These results have now been illustrated in the **revised Supplementary Fig. 8 and described in lines 16-20, page 14** of the revised manuscript.

18. *Page 15 line 3 “become” should be “come”.*

Response: We thank the reviewer for careful correction. We have now corrected “become” into “come” **in line 24, page 16 of the revised manuscript.**

19. *My interpretation of the blot 5k is that the PKG-I expression in both groups increases following CFA and that the differences between the two groups of mice are not reflected in the blot at all. How large were groups in Fig 5L for each column? The authors state group sizes of 3-6 in the legend why the variability in groups? This needs to be addressed.*

Response: We are very sorry for this inappropriate typical example. To further confirm this data and reduce variability in groups, we have performed a new set of experiments and provided new typical example. Please see changes **in the revised Fig. 5k and l.**

By the way, as the **reviewer #2** suggested, we have added a new set of experiments to examine the changes of the above proteins in dorsal root lysates. As shown **in the revised Supplementary Fig. 9i, j**, similar results were obtained from dorsal root lysates as the DRG lysates. In the basal state, lysates of dorsal roots from naive SNS-NR1^{-/-} mice showed a comparable PKG-I expression as compared to NR1^{fl/fl} mice. In striking contrast, following CFA inflammation, dorsal roots from NR1^{fl/fl} mice showed a striking increase of PKG-I expression over the basal state. This was markedly reduced in CFA-injected SNS-NR1^{-/-} mice. These results indicate that persistent activation of nociceptors leads to rapid activation of NMDARs-PKG-I signalling cascades in the DRG and dorsal roots.

20. In the description of intrathecal delivery of drugs more detail is required about the insertion of the catheter. Was this made at the level of the spinal cord drugs were administered to or was the catheter fed from thoracic. Spinal segments down the spinal canal?

Response: We apologize for not describing the procedure for intrathecal delivery in more details. To enable intrathecal delivery at the level of lumbar spinal segments in mice, a polytetrafluoroethylene catheter was stereotactically inserted till the lumbar enlargement after hemilaminectomy at S1-S2 under anaesthesia by 1% pentobarbital sodium, as described previously (Luo et al., 2012). We have added this detailed information into the methods for intrathecal delivery of drugs in vivo **in lines 3-5, page 44** of the revised manuscript.

21. The decrease in threshold following capsaicin seems to be present in the SNS-NR1 -/- group compared to basal (Fig 7d), certainly this decrease is comparable to other changes in this study with similar variability in each group. The lack of detailed reporting of statistics in the paper is an issue and should be addressed. The reader would benefit from knowing effect size, details of stats, degrees of freedom etc as is not routinely reported in papers. This may be in the text of as supplementary information/ table.

Response: We greatly appreciate the reviewer for his (her) constructive

comments. To reduce the variability in each group, we have added more mice in each group in Fig. 7d and b. The new data has been illustrated in the revised Fig. 7d and b. See also the detailed information in the Supplementary Table 2. Regarding the statistical analysis, we fully agree with the reviewer that the reader would benefit a lot from knowing the detailed statistical analysis including effect size, degree of freedom etc. As the reviewer suggested, we have provided new Supplementary Table 2 to show all the statistic details for each data in each figure. We have now thoroughly checked and performed normality test by the Shapiro–Wilk test and homogeneity of variance test by Levene’s test for every single data and chose the corresponding statistical analyses for each data. Data that met these two conditions were analyzed using a two-tailed unpaired or paired t-test, one-way analysis of variance (ANOVA) or two-way ANOVA. Data sets that were not normally distributed were analyzed with a nonparametric test, data sets that were normally distributed but not conform to homogeneity of variance test were analyzed with a corrected two-tailed unpaired or paired t-test, one-way or two-way ANOVA. Additionally, detailed statistical analyses for each panel have now been described in the figure legends.

References

- Balassa T, Varró P, Elek S, Drozdovszky O, Szemerszky R, Világi I, Bárdos G. Changes in synaptic efficacy in rat brain slices following extremely low-frequency magnetic field exposure at embryonic and early postnatal age. *Int J Dev Neurosci.* **31**, 724-730 (2013).
- Behr J, Wozny C, Fidzinski P, Schmitz D. Synaptic plasticity in the subiculum. *Prog Neurobiol.* **89**, 334-342 (2009).
- Bortolotto ZA, Clarke VR, Delany CM, Parry MC, Smolders I, Vignes M, Ho KH, Miu P, Brinton BT, Fantaske R, Ogden A, Gates M, Ornstein PL, Lodge D, Bleakman D, Collingridge GL. Kainate receptors are involved in synaptic plasticity. *Nature* **402(6759)**, 297-301 (1999).
- Castillo PE. Presynaptic LTP and LTD of excitatory and inhibitory synapses. *Cold Spring Harb Perspect Biol.* **4(2)**, a005728 (2012).
- Castro-Alamancos MA, Calcagnotto ME. Presynaptic long-term potentiation in corticothalamic synapses. *J Neurosci* **19**, 9090-9097 (1999).
- Chen HX, Jiang M, Akakin D, Roper SN. Long-term potentiation of excitatory synapses on neocortical somatostatin-expressing interneurons. *J Neurophysiol.* **102**, 3251-3259 (2009).
- Contractor A, Mulle C, Swanson GT. Kainate receptors coming of age: milestones of two decades of research. *Trends Neurosci.* **34**, 154-163 (2011).
- Cull-Candy S, Brickley S, Farrant M. NMDA receptor subunits: diversity, development and disease. *Curr Opin Neurobiol.* **11(3)**, 327-335 (2001).
- Devor M. Unexplained peculiarities of the dorsal root ganglion. *Pain Suppl* **6**, S27-35 (1999).
- Devor M. Ectopic discharge in Abeta afferents as a source of neuropathic pain. *Exp. Brain Res.* **196**, 115e128 (2009).
- Drdla R, Sandkühler J. Long-term potentiation at C-fibre synapses by low-level presynaptic activity in vivo. *Mol Pain* **4**, 18 (2008).
- Gozariu M, Bragard D, Willer JC, Bars DL. Temporal summation of C-fiber

- afferent inputs: competition between facilitatory and inhibitory effects on C-fiber reflex in the rat. *J Neurophysiol.* **78(6)**, 3165-3179 (1997).
- Han Q, Kim YH, Wang X, Liu D, Zhang ZJ, Bey AL, Lay M, Chang W, Berta T, Zhang Y, Jiang YH, Ji RR. SHANK3 deficiency impairs heat hyperalgesia and TRPV1 signaling in primary sensory neurons. *Neuron* **92(6)**, 1279-1293 (2016).
- Herrero JF, Laird JM, López-García JA. Wind-up of spinal cord neurones and pain sensation: much ado about something? *Prog Neurobiol.* **61(2)**, 169-203 (2000).
- Hylden JL, Hayashi H, Dubner R, Bennett GJ. Physiology and morphology of the lamina I spinomesencephalic projection. *J Comp Neurol.* **247**, 505–515 (1986).
- Ikeda H, Heinke B, Ruscheweyh R, Sandkühler J. Synaptic plasticity in spinal lamina I projection neurons that mediate hyperalgesia. *Science* **299(5610)**, 1237-1240 (2003).
- Ikeda H, Stark J, Fischer H, Wagner M, Drdla R, Jäger T, Sandkühler J. Synaptic amplifier of inflammatory pain in the spinal dorsal horn. *Science* **312**, 1659-1662 (2006).
- Koga K, Descalzi G, Chen T, Ko HG, Lu J, Li S, Son J, Kim T, Kwak C, Huganir RL, Zhao MG, Kaang BK, Collingridge GL, Zhuo M. Coexistence of two forms of LTP in ACC provides a synaptic mechanism for the interactions between anxiety and chronic pain. *Neuron* **85**, 377-389 (2015).
- Lopez de Armentia M, Sah P. Bidirectional synaptic plasticity at nociceptive afferents in the rat central amygdala. *J Physiol.* **581**, 961–970 (2007).
- Luo C, Gangadharan V, Bali KK, Xie RG, Agarwal N, Kurejova M, Tappe-Theodor A, Tegeder I, Feil S, Lewin G, Polgar E, Todd AJ, Schlossmann J, Hofmann F, Liu DL, Hu SJ, Feil R, Kuner T, Kuner R. Presynaptically localized cyclic GMP-dependent protein kinase 1 is a key determinant of spinal synaptic potentiation and pain hypersensitivity. *PLoS Biol.* **10**, 363-394

- (2012).
- Luo C, Kuner T, Kuner R. Synaptic plasticity in pathological pain. *Trends Neurosci.* **37**, 343-355 (2014).
- Ma SB, Xian H, Wu WB, Ma SY, Liu YK, Liang YT, Guo H, Kang JJ, Liu YY, Zhang H, Wu SX, Luo C, Xie RG. CCL2 facilitates spinal synaptic transmission and pain via interaction with presynaptic CCR2 in spinal nociceptor terminals. *Mol Brain* **13**(1), 161 (2020).
- Maru E, Ashida H, Tatsuno J. Long-lasting reduction of dentate paired-pulse depression following LTP-inducing tetanic stimulations of perforant path. *Brain Res.* **478**, 112-120 (1989).
- Marvizón JC, McRoberts JA, Ennes HS, Song B, Wang X, Jinton L, Corneliussen B, Mayer EA. Two N-methyl-D-aspartate receptors in rat dorsal root ganglia with different subunit composition and localization. *J Comp Neurol.* **446**(4), 325-341 (2002).
- Monday HR, Castillo PE. Closing the gap: long-term presynaptic plasticity in brain function and disease. *Curr Opin Neurobiol.* **45**, 106-112 (2017).
- Monday HR, Younts TJ, Castillo PE. Long-term plasticity of neurotransmitter release: emerging mechanisms and contributions to brain function and disease. *Annu Rev Neurosci.* **41**, 299-322 (2018).
- Monyer H, Burnashev N, Laurie DJ, Sakmann B, Seeburg PH. Developmental and regional expression in the rat brain and functional properties of four NMDA receptors. *Neuron* **12**(3), 529-40 (1994).
- Naka A, Gruber-Schoffnegger D, Sandkuhler J. Non-Hebbian plasticity at C-fiber synapses in rat spinal cord lamina I neurons. *Pain* **154**, 1333-1342 (2013).
- Nicoll RA, Schmitz D. Synaptic plasticity at hippocampal mossy fibre synapses. *Nat Rev Neurosci.* **6**, 863-876 (2005).
- Pagadala P, Park CK, Bang S, Xu ZZ, Xie RG, Liu T, Han BX, Tracey WD Jr, Wang F, Ji RR. Loss of NR1 subunit of NMDARs in primary sensory neurons

- leads to hyperexcitability and pain hypersensitivity: involvement of Ca²⁺-activated small conductance potassium channels. *J Neurosci.* **33**, 13425-13430 (2013).
- Ruscheweyh R, Ikeda H, Heinke B, Sandkühler J. Distinctive membrane and discharge properties of rat spinal lamina I projection neurones in vitro. *J Physiol.* **555**, 527-43 (2004).
- Salin PA, Malenka RC, Nicoll RA. Cyclic AMP mediates a presynaptic form of LTP at cerebellar parallel fiber synapses. *Neuron* **16**, 797-803 (1996).
- Sandkühler J. Models and mechanisms of hyperalgesia and allodynia. *Physiol Rev.* **89(2)**, 707-758 (2009).
- Sandkühler J. Translating synaptic plasticity into sensation. *Brain* **138**, 2463-2464 (2015).
- Schouenborg J. Functional and topographical properties of field potentials evoked in rat dorsal horn by cutaneous C-fiber stimulation. *J Physiol.* **356**, 169-192 (1984).
- Schouenborg J, Sjölund BH. Activity evoked by A- and C-afferent fibers in rat dorsal horn neurons and its relation to a flexion reflex. *J Neurophysiol.* **50(5)**, 1108-1121 (1983).
- Schulz PE, Cook EP, Johnston D. Using paired-pulse facilitation to probe the mechanisms for long-term potentiation (LTP). *J Physiol. Paris* **89**, 3-9 (1995).
- Shin RM, Tully K, Li Y, Cho JH, Higuchi M, Suhara T, Bolshakov VY. Hierarchical order of coexisting pre- and postsynaptic forms of long-term potentiation at synapses in amygdala. *Proc Natl Acad Sci U S A.* **107**, 19073-19078 (2010).
- Simonetti M, Hagenston AM, Vardeh D, Freitag HE, Mauceri D, Lu J, Satagopam VP, Schneider R, Costigan M, Bading H, Kuner R. Nuclear calcium signaling in spinal neurons drives a genomic program required for persistent inflammatory pain. *Neuron* **77(1)**, 43-57 (2013).
- Song Y, Li HM, Xie RG, Yue ZF, Song XJ, Hu SJ, Xing JL. Evoked bursting in

- injured Abeta dorsal root ganglion neurons: a mechanism underlying tactile allodynia. *Pain* **153**, 657e665 (2012).
- Spike RC, Puskár Z, Andrew D, Todd AJ. A quantitative and morphological study of projection neurons in lamina I of the rat lumbar spinal cord. *Eur J Neurosci.* **18(9)**, 2433-2448 (2003).
- Sun W, Miao B, Wang XC, Duan JH, Wang WT, Kuang F, Xie RG, Xing JL, Xu H, Song XJ, Luo C, Hu SJ. Reduced conduction failure of the main axon of polymodal nociceptive C-fibres contributes to painful diabetic neuropathy in rats. *Brain* **135**, 359-375 (2012).
- Sun W, Yang F, Wang Y, Fu H, Yang Y, Li CL, Wang XL, Lin Q, Chen J. Contribution of large-sized primary sensory neuronal sensitization to mechanical allodynia by upregulation of hyperpolarization-activated cyclic nucleotide gated channels via cyclooxygenase 1 cascade. *Neuropharmacology* **113**, 217-230 (2017).
- Xiao MM, Zhang YQ, Wang WT, Han WJ, Lin Z, Xie RG, Cao Z, Lu N, Hu SJ, Wu SX, Dong H, Luo C. Gastrodin protects against chronic inflammatory pain by inhibiting spinal synaptic potentiation. *Sci Rep.* **6**, 37251 (2016).
- Xiao WH, Bennett GJ. Persistent low-frequency spontaneous discharge in A-fiber and C-fiber primary afferent neurons during an inflammatory pain condition. *Anesthesiology* **107**, 813e821 (2007).
- Yang Y, Calakos N. Presynaptic long-term plasticity. *Front Synaptic Neurosci.* **5**, 8 (2013).
- Zhou HY, Chen SR, Chen H, Pan HL. Opioid-induced long-term potentiation in the spinal cord is a presynaptic event. *J Neurosci.* **30**, 4460-4466 (2010).

REVIEWERS' COMMENTS

Reviewer #1 (Remarks to the Author):

The authors have conducted additional experiments assessing protein changes in dorsal roots, the results of which support the previous conclusion on the effects of preNMDARs in the terminals. Thoughtful revisions were also made to clarify PPR analysis and other experimental approaches. I have no further comments. The revised work would be a good contribution to the pain field.

Reviewer #3 (Remarks to the Author):

The authors have satisfied my previous comments in the first review

I have also read the questions from Reviewer 2, the authors' responses etc. The rebuttal is very thorough and further studies have been performed as requested by Reviewer 2. As far as I can tell and to the best of my knowledge these address the issues raised and I would be willing to recommend acceptance of the MS.